# Chemo-mechanical diffusion waves explain collective dynamics of immune cell podosomes

Ze Gong [1,2], Koen van den Dries [3], Rodrigo A. Migueles-Ramírez [4,5,6], Paul W. Wiseman [4], Alessandra Cambi [3] & Vivek B. Shenoy [1,7] ✉

Immune cells, such as macrophages and dendritic cells, can utilize podosomes, mechanosensitive actin-rich protrusions, to generate forces, migrate, and patrol for foreign antigens. Individual podosomes probe their microenvironment through periodic protrusion and retraction cycles (height oscillations), while oscillations of multiple podosomes in a cluster are coordinated in a wave-like fashion. However, the mechanisms governing both the individual oscillations and the collective wave-like dynamics remain unclear. Here, by integrating actin polymerization, myosin contractility, actin diffusion, and mechanosensitive signaling, we develop a chemo-mechanical model for podosome dynamics in clusters. Our model reveals that podosomes show oscillatory growth when actin polymerization-driven protrusion and signaling-associated myosin contraction occur at similar rates, while the diffusion of actin monomers drives wave-like coordination of podosome oscillations. Our theoretical predictions are validated by different pharmacological treatments and the impact of microenvironment stiffness on chemo-mechanical waves. Our proposed framework can shed light on the role of podosomes in immune cell mechanosensing within the context of wound healing and cancer immunotherapy.

Dendritic cells (DCs) and macrophages, act as gatekeepers of the immune system by surveilling peripheral tissues for foreign antigens. During this process, these cells form dozens of specialized actin-rich protrusions called podosomes, which are typically organized into large clusters. By utilizing podosomes, DCs and macrophages can control adhesions, degrade the surrounding extracellular matrix (ECM), and remodel the extracellular environment to facilitate their migration[1–3]. Podosomes are characterized by an actin-based core surrounded by an adhesive ring consisting of integrins and adaptor proteins, such as vinculin and talin[4,5]. Podosomes generate protrusive forces to penetrate the underlying ECMs at the cores, while applying tensile forces to pull the ECMs at the rings[6,7]. Studies in DCs and macrophages have shown that individual podosomes exhibit periodic oscillations in the fluorescence intensity of core actin, ring components, and the protrusive forces exerted at their cores[7–9]. Collectively, podosomes in a cluster show spatially correlated behaviors, where the oscillations of podosome components are coordinated in a wave-like manner within the cluster[10,11]. However, the relationship which directs the core protrusive and ring tensile forces into oscillations of individual podosomes and subsequent spatiotemporal wave patterns in clusters remains unknown. Understanding these processes can provide insights on how podosomes in a cluster collectively probe and respond

[1]Center for Engineering Mechanobiology, University of Pennsylvania, Philadelphia, PA 19104, USA. [2]CAS Key Laboratory of Mechanical Behavior and Design of Materials, Department of Modern Mechanics, University of Science and Technology of China, Hefei, Anhui 230027, China. [3]Department of Medical Biosciences, Radboud University Medical Center, Nijmegen, The Netherlands. [4]Departments of Chemistry and Physics, McGill University, Montreal, QC H3A 0B8, Canada. [5]Quantitative Life Sciences, McGill University, Montreal, QC H3A 3R1, Canada. [6]Department of Biology, McGill University, Montreal, QC H3G 0B1, Canada. [7]Department of Materials Science and Engineering, University of Pennsylvania, Philadelphia, PA 19104, USA. ✉e-mail: vshenoy@seas.upenn.edu

to chemo-mechanical cues from their surroundings, which is essential for their function.

To obtain biophysical insights into the dynamics of podosome clusters, we adopted a bottom-up modeling approach. We first integrated actin polymerization, myosin contractility, and mechanosensitive signaling pathways into a chemo-mechanical model for oscillatory growth of individual podosomes. By considering diffusion of actin monomers within a cluster, our model demonstrates how mesoscale coordination of individual podosome height oscillations can arise and form wave-like propagation in clusters, which we call chemo-mechanical diffusion waves. This is the first theoretical model, to our knowledge, that systematically elaborates the mechanism for the spatiotemporal wave dynamics in podosome clusters. Our model predicts the influence of pharmacological treatments targeting force-generating processes, including myosin activity, actin polymerization, and Rho-associated kinase (ROCK) activity, on podosome dynamics. We validate these predictions experimentally in living primary human DCs, using fluorescence microscopy, image analysis and spatiotemporal image correlation spectroscopy (STICS) to characterize the wavelengths, periods, and speeds of chemo-mechanical waves. Furthermore, our model predicts the enhanced chemo-mechanical diffusion waves previously observed on stiffer substrates[12]. The mechanistic understanding of the oscillatory growth of individual podosomes as well as the wave-like propagation of podosome components and forces can provide means to modulate immune cell mechanosensing and migration, the key processes in wound healing and cancer immunotherapy.

## Results

### A chemo-mechanical model for dynamics of individual podosomes

To develop a biophysical description of podosome growth, we start by considering the molecular mechanisms of force generation in a podosome. Podosomes have a conical structure characterized by a protrusive actin-rich core, an adhesive integrin ring, and ventral actin filaments that connect the core with the ring[12,13] (Fig. 1A, B). During podosome growth, actin monomers (G-actin) continuously polymerize into actin filaments (F-actin), generating a protrusive or compressive force in the core to drive the growth of core F-actin. At the same time, myosin motors are dynamically recruited to the ventral actin filaments, generating active contractile forces to constrain core growth[3,9,12]. The core protrusive force generated by actin polymerization $F_p$ is balanced by the tensile force generated by the actomyosin contractility, $F_r$; balance of forces gives the relation $F_p = F_r \cos(\theta)$, where we assume $\theta$ to be a constant angle between the ventral actin filaments and the core F-actin (Fig. 1A). Next, we describe the dynamics of the two force-generating processes.

In the podosome core, actin polymerization generates protrusive forces to drive the core protrusion. The protrusive force $F_p$ generated by actin polymerization deforms the underlying substrate following $F_p = k_s l_1$, where $k_s$ and $l_1$ are the stiffness and the displacement of the substrate, respectively. The resistance force from the substrate reduces actin polymerization[14,15], which can be expressed as $V_p = V_{pm}(1 - F_p/F_{sp0})$, where $V_{pm}$ is the maximum polymerization speed (in the absence of protrusive forces) and $F_{sp0}$ is the characteristic protrusive force generated by the core. By assuming the F-actin depolymerizes at a constant speed $V_d$, the growth rate of the core height is determined by the difference between the polymerization and depolymerization speed, which can be written as:

$$\frac{d(l + l_1)}{dt} = V_{pm}\left(1 - \frac{F_p}{F_{sp0}}\right) - V_d + \chi_p(t). \tag{1}$$

Here, $l + l_1$ is the total podosome height, where $l$ is the core height above the undeformed substrate (Fig. 1A), and the Gaussian noise $\chi_p(t)$

accounts for fluctuations in the polymerization process. When the polymerization speed balances with the depolymerization speed (i.e., $V_p = V_d$) and the noise $\chi_p(t)$ is neglected, we obtain the steady-state protrusive force $F_{ps} = F_{sp0}\left(1 - V_d/V_{pm}\right)$. To account for larger protrusive forces on stiffer substrates revealed by protrusion force microscopy[7], we write the characteristic stall force as $F_{sp0} = F_{p0}k_s/(k_c + k_s)$, where $k_c$ is the core stiffness and $F_{p0}$ is the characteristic protrusive force on a rigid substrate; the protrusive force increases linearly with the substrate stiffness $k_s$ at low levels but saturates on very stiff substrates.

The other force-generating process is the mechanosensitive recruitment of myosin to the ventral actin filaments. As the F-actin network assembles in the podosome core and generates a protrusive force, myosin motors exert contractile forces on the ventral actin filaments to balance the core protrusive force and constrain core growth. To further model the contraction of actomyosin filaments, we adopt a two-element active contraction model[16], consisting of an active element (with contractile force $F_m$) in parallel with a passive elastic element (with stiffness $k_f$). The active element characterizes myosin contractility, while the passive (elastic) element represents the stiffness of the ventral actin filaments (Fig. 1B, inset i). Thus, the tensile force sustained by ring $F_r$ can be written as the sum of the active and passive forces: $F_r = F_m + k_f(x_r - x_{r0})$, where $x_r$ and $x_{r0}$ denote the current and initial length of the ventral actin filaments, respectively. Note here the ventral actin filament length $x_r$ is linearly correlated with the core height above the undeformed substrate $l$ based on the geometric constraint, i.e., $x_r \cos(\theta) = l$.

Myosin activity is known to be mediated by the Rho-ROCK signaling pathway[10,17,18]. Specifically, the tensile force $F_r$ transmitted to the substrate through integrin adhesions can positively feed back to the active force $F_m$ through this pathway[16] (Fig. 1B, inset ii). In analogy with muscle fibers, the contraction of actomyosin filaments, $x_r - x_{r0}$, reduces available binding sites for myosin, causing a decrease in bound myosin motors and active contractility[16,19,20]. Here, we use $\alpha$ to characterize the effects of positive feedback (proportional to the tensile force $F_r$) and $\gamma$ to characterize the effects of actomyosin filament length on active contractility, which leads to the following equation governing the dynamics of myosin force:

$$\tau_m \frac{dF_m}{dt} + F_m = F_{m0} - \gamma(x_r - x_{r0}) + \alpha F_r + \chi_m(t). \tag{2}$$

Here $\tau_m$ is the characteristic time for myosin turnover, $F_{m0}$ is the base level myosin force, and $\chi_m(t)$ is a Gaussian noise signal accounting for fluctuations in myosin dynamics. The equations in the simulations are summarized in Methods section, and more details on the derivation and parameter selection can be found in Supplementary Notes 1–3.

### Characteristic timescales and mechanosensitive feedback determine the distinct phases of podosome dynamics

The governing equations for the core protrusion and ring contraction dynamics (Eqs. (1) and (2)) are two coupled ordinary differential equations. Starting from arbitrary initial core height and myosin force, the podosome growth system can reach a steady state either monotonically or in an oscillatory manner (Supplementary Fig. 1). Here, we applied a linear perturbation analysis to this podosome growth system and analyzed the stability of its steady-state solutions (refer to Methods). By evaluating the eigenvalues for the perturbed podosome system (Eq. (8)), we found that podosome oscillations are governed by the signaling-associated feedback parameter $\Gamma = \frac{\alpha k_f - \gamma}{k_f - \gamma} \frac{k_s}{k_s + k_f}$ and the ratio $\tau_m/\tau_p$ between the timescales governing myosin turnover ($\tau_m$) and the core protrusion ($\tau_p$). Note here the protrusion timescale is $\tau_p = \frac{F_{sp0}}{V_{pms}}\left(\frac{1}{k_f} + \frac{1}{k_s}\right)\left(1 - \frac{\gamma}{k_f}\right) \sim \frac{F_{sp0}}{k_s V_{pms}}$, which characterizes the time it takes the podosome core with polymerization speed $V_{pms}$ to grow by a characteristic substrate displacement $F_{sp0}/k_s$. The magnitude of the

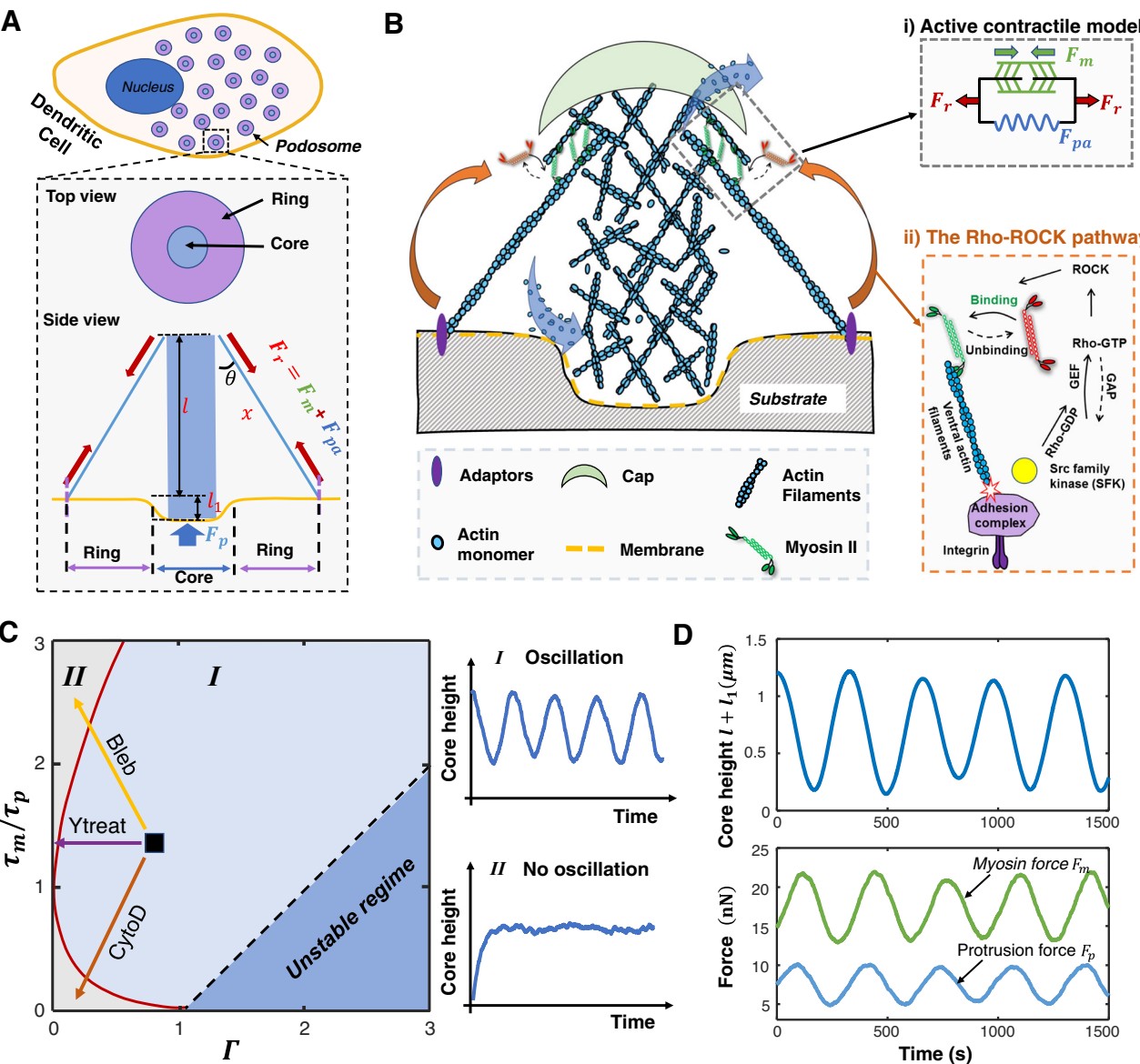

**Fig. 1 | A chemo-mechanical model for the oscillatory growth of individual podosomes. A** Schematic showing a dendritic cell with podosomes. Inset shows simplified geometry of the podosome with top and side views. **B** Schematic describing the chemo-mechanical model for podosome growth. The orange arrows indicate mechanosensitive signaling through the Rho-ROCK pathway. Inset i): the active contractile model for myosin-based force generation, where the active contractile forces $F_m$ (the green element with arrows), the passive actin filament elastic force $F_{pa}$ (blue spring), and total ring force $F_r$ (red arrows) are marked. Inset ii): tensile forces in the ring trigger a conformational change in vinculin, exposing binding sites of Src family of tyrosine kinases (SFKs); this change promotes Rho-

GTPases by controlling the activity of guanine nucleotide exchange factors (GEFs) and GTPase-activating proteins (GAPs), which eventually increases the contractile forces in the ventral F-actins. **C** Phase diagram showing two types of protrusion patterns: oscillatory protrusions (regime I) and monotonically growing protrusions (regime II), based on the ratio $\tau_m/\tau_p$ and the feedback parameter $\Gamma$. The arrows indicate the influence of cytochalasin D (CytoD, red), blebbistatin (Bleb, yellow), and Y27632 (Ytreat, purple) treatments on the dynamics, corresponding to Fig. 2A, B. **D** Representative curves showing the theoretically predicted dynamics for podosome core height $l + l_1$ (top panel), protrusion force $F_p$ (blue line in the bottom panel), and myosin force $F_m$ (green line in the bottom panel).

feedback parameter $\Gamma \sim \alpha$ characterizes the strength of the Rho-ROCK pathway when the stiffness for the passive component is much larger than the effective stiffness for the active element of actomyosin filaments, i.e., $k_f \gg \gamma$.

The stability analysis allows us to generate a phase diagram (Fig. 1C), which reveals distinct phases of oscillatory or monotonic dynamic behaviors based on the feedback parameter $\Gamma$ and the time-scale ratio $\tau_m/\tau_p$. Podosomes exhibit oscillatory growth when the two timescales are comparable ($\tau_m/\tau_p \sim 1$,) and the feedback parameter $\Gamma$ is at an intermediate value (region I in Fig. 1C). Although the spontaneous oscillations in region I are damped, the random fluctuations (i.e., Gaussian noise contributions $\chi_m(t)$ and $\chi_p(t)$ in Eqs. (1) and (2)) from

myosin recruitment or core-actin polymerization constantly push the podosome away from the steady state; this initiates excursions in the phase space of core height and myosin force, leading to persistent oscillations (Supplementary Fig. 1A, B). Our simulations also show that the presence of noise can sustain persistent oscillations with a random amplitude but an almost fixed oscillation period $\sim 2\pi\sqrt{\tau_m\tau_p}$ (Supplementary Fig. S1B, refer to Methods). On the other hand, when the two timescales substantially differ, the system relaxes monotonically to the steady-state (region II in Fig. 1C); this indicates that oscillatory growth requires that the two competing force-generating mechanisms, the polymerization-driven protrusion and myosin contraction, occur at similar rates. Importantly, the mechanosensitive Rho-ROCK

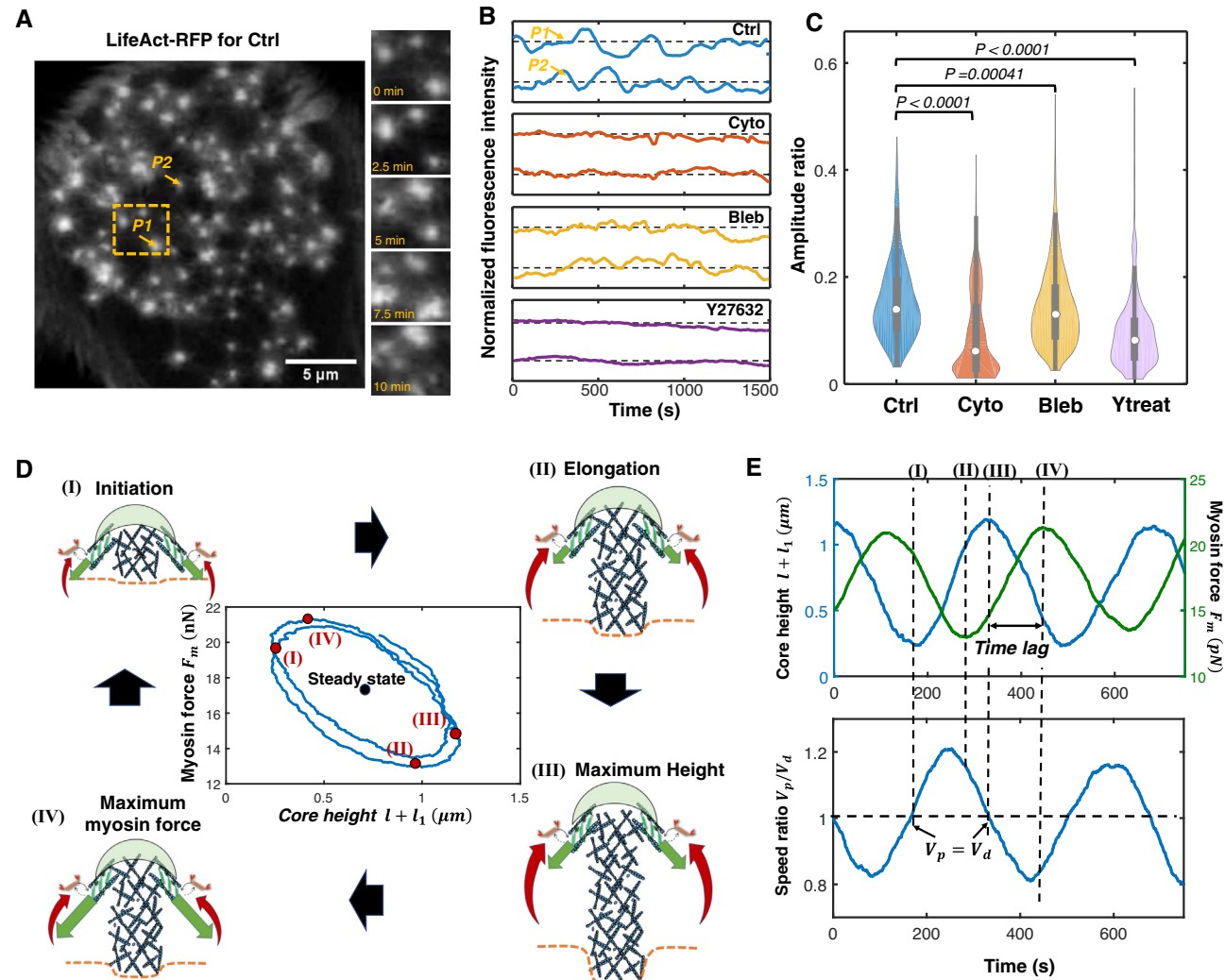

**Fig. 2 | Model predicts the effects of pharmacological treatments on podosome oscillations. A** A representative LifeAct-RFP transfected dendritic cell for the control case. Podosome dynamics at different times for the yellow-dashed region are shown in the right panels. Scale bar, 5 μm. Dynamics of LifeAct-RFP fluorescence intensity for the two representative podosomes (marked as P1 and P2) are shown in the top panel of **B**. **B** Representative fluorescence intensity profiles for control (Ctrl, blue), cytochalasin D (red), blebbistatin (yellow), and Y27632 (purple) treatments. **C** The amplitude ratio $r_a$ for the control and different pharmacological treatments. Statistically significant differences are indicated ($n$ = 906, 451, 984, 536 podosomes from left to right, ANOVA with Benjamini–Hochberg procedure was used). Boxplots are displayed using the Tukey method (center dot, median; box limits, upper and lower quartiles; whiskers, last point within a 1.5 × interquartile range). **D** Schematic showing the interplay of intracellular processes in the four stages of oscillatory podosome growth. The substrate provides only a small resistance force at the beginning, and actin monomers polymerize to drive podosome growth (stage I). As the podosome grows and deforms the substrate, the protrusion speed gradually decreases (stage II). The active myosin contractility (green arrows) continues to increase when the podosome core reaches its maximum height (stage III) because of the time delay associated with signaling feedback (red arrows). The podosome begins to retract in response to high levels of contractile forces (stage IV). Inset (middle panel): myosin force versus core height. **E** The simulated core length $l + l_1$ (blue line in top panel), myosin contractile force $F_m$ (green line in top panel), and the ratio between polymerization and depolymerization speed $V_p/V_d$ (blue line in bottom panel) plotted as a function of time.

---

pathway is critical for oscillations as well, since a very weak signaling feedback $\Gamma \to 0$ also leads to monotonic podosome growth. Note that the steady state can also be physically unstable when the signaling feedback $\Gamma$ is too large.

### Polymerization-driven core protrusion and signaling-associated myosin dynamics govern podosome oscillatory growth

By choosing the parameters in the oscillatory regime (region I in Fig. 1C), we can simulate the oscillatory behaviors of the core height, the protrusive force, and active myosin contractility (Fig. 1D; Supplementary Fig. 2A). We transfected human DCs with LifeAct-RFP and vinculin-GFP and found that both the F-actin and vinculin fluorescence intensities oscillate periodically with time (Fig. 2A, B; Supplementary Movie 1). As a larger podosome height corresponds to more F-actin in

the core and hence higher fluorescence intensity of LifeAct-RFP, we assume that the fluorescence intensity is directly proportional to the podosome height. The predicted oscillation period $2\pi\sqrt{\tau_m\tau_p}$ is on the order of several minutes in line with our experiments (Fig. 2A, B). It is worth noting that fluorescence traces of F-actin and vinculin are highly correlated (Supplementary Fig. 2B, C), which confirms the predicted in-phase relationship between the core height and the ring size dynamics (Supplementary Fig. 2A). Meanwhile, the time-averaged magnitudes of the simulated protrusive force (~10 nN, Supplementary Fig. 2A), ventral actin filament length (~1 μm, Supplementary Fig. 2A), and core height (~700 nm, Supplementary Fig. 4C) agree with previous experimental measurements[7,9,12,21]. To further understand the mechanism governing podosome oscillations, we identify the four stages of a typical protrusion-retraction oscillatory cycle (Fig. 2D, E):

(I)     In the early stage of growth, the podosome has a small core height and low levels of active myosin contractility (green arrows in Fig. 2D). The podosome height increases as the polymerization speed exceeds the depolymerization speed (Eq. (1)).

(II)    As the F-actin network within the core grows and pushes against the substrate with a characteristic time $\tau_p$, the protrusive force $F_p$ increases and resists actin polymerization. The protrusion speed gradually decreases.

(III)   When the protrusion speed decreases to zero, the core height, protrusive force, and ring force attain their maximum values, leading to a high level of feedback via the Rho-Rock pathway (red arrows in Fig. 2D). In response to this signaling response, more myosin are recruited and generate larger contractile forces (Eq. (2)). Since myosin force also requires a characteristic time $\tau_m$ (that is comparable to $\tau_p$) to accumulate, there is a time delay between the maximum core height and the maximum myosin force (Fig. 2E).

(IV)    As the protrusion speed continues to decrease and drops below zero, the podosome starts retracting. Subsequently, the myosin force decays from its maximum value, and polymerization speed increases. Once the polymerization speed exceeds the depolymerization speed, the next oscillation cycle begins.

The two force-generating processes, i.e., polymerization-driven core protrusion and signaling-associated myosin contraction, govern the dynamics of podosome oscillations. Ring contractile forces continue to increase when podosomes reach their maximum height, and podosomes begin to retract in response to high contractile forces in the ring. When contractile forces decrease and polymerization dominates, podosomes start to grow again. This back-and-forth interaction between the two force-generating processes causes podosome oscillations.

## Model predicts the effects of pharmacological treatments on podosome dynamics

Next, we validate our model by perturbing the podosome system using pharmacological treatments targeting the two force-generating processes and the mechanosensitive Rho-ROCK pathway. Based on the phase diagram (Fig. 1C), we predict that when we inhibit actin polymerization, core-actin growth becomes much slower than myosin dynamics, i.e., $\tau_m/\tau_p \ll 1$, causing non-oscillatory podosome behaviors (Fig. 1C, red arrow). When myosin contractility is inhibited, myosin turnover becomes very slow such that $\tau_m/\tau_p \gg 1$, leading to non-oscillatory growth in podosomes (Fig. 1C, yellow arrow). Podosomes can also show non-oscillatory growth when the mechanosensitive Rho-ROCK pathway is inhibited, i.e., $\Gamma \rightarrow 0$ (Fig. 1C, purple arrow). All these perturbations lead to non-oscillatory behaviors, which are summarized in the phase diagram (Fig. 1C) and can also be observed in our simulations (Supplementary Fig. 3, refer to Supplementary Note 4). To test these predictions, we treated DCs with cytochalasin D, blebbistatin, and Y27632 to inhibit actin polymerization, myosin contractility, and the Rho-ROCK signaling pathway, respectively. By extracting the LifeAct-RFP intensity traces for the control and treatment cases, we observed that core-actin oscillations become less prominent after these treatments (Fig. 2B). To further quantitatively characterize the oscillation of the fluorescence traces, we evaluated their oscillation amplitude ratio (i.e., the ratio between the oscillation amplitude and the time-averaged intensity[22], refer to Methods). We found that all these treatments significantly reduce the amplitude ratios of LifeAct-RFP intensity (Fig. 2C). This reduced ratio indicates that oscillations are inhibited after pharmacological treatments, in agreement with our predictions (Fig. 1C; Supplementary Fig. 3A–C). Overall, our experiments with pharmacological treatments validate our model predictions that podosomes oscillate when actin polymerization-driven core protrusion and myosin turnover occur at similar rates and when the signaling feedback is not too strong or weak.

## Actin diffusion drives coordinated wave-like patterns in podosome clusters

Next, we studied how the oscillations of individual podosomes give rise to wave-like patterns in a podosome cluster. To model the spatiotemporal collective dynamics, we consider the diffusion and exchange of actin between the podosomes within the cluster. The time evolution of the G-actin concentration $c_a(x, y, t)$ in the plane of the cluster at time $t$ is determined by both actin diffusion and actin polymerization (or depolymerization), which can be written as:

$$\frac{\partial c_a}{\partial t} = D_a \nabla^2 c_a - \mu \frac{d(l+l_1)}{dt}. \qquad (3)$$

Here $D_a$ is the G-actin diffusion constant, and parameter $\mu$ controls the consumption (release) of G-actin as actin monomers are assembled into (disassembled from) the core to increase (decrease) its height, $l + l_1$; this is proportional to the actin filament number $N_a$ of the podosome core, i.e., $\mu \propto N_a$. At the same time, a larger G-actin concentration at the podosome core increases the polymerization speed, that is $V_{pm} = V_{p0} + \beta c_a$, where a linear dependence of polymerization speed on actin concentration with sensitivity $\beta$ is assumed for simplicity. By integrating this diffusion process with the model for individual podosome dynamics, we simulate the spatiotemporal dynamics of the podosome cluster using both discrete and coarse-grained approaches. We first consider the dynamics of an array of podosomes individually with a uniform distance $d_0$ from each other in the discrete model (Fig. 3A and Supplementary Fig. 4A); we then further generalize the discrete model using a coarse-grained (or continuum) approach (Fig. 3B; Supplementary Fig. 4B), where only a spatial average of the podosome heights, treated as a continuous variable, is studied (refer to Supplementary Note 2). By assuming an initial distribution where only podosomes at the center have large heights, both the discrete and continuum models show that a height wave front originates from the center and gradually transits outward, forming a radial wave pattern (Supplementary Movies 2 and 3). These simulated radial patterns closely resemble the observed LifeAct-RFP intensity dynamics in the experiments (Fig. 3C).

To understand the mechanism for the formation and propagation of radial waves, we take a closer look at the dynamics of individual podosomes located at different distances from the center during wave propagation (Fig. 3D).

Phase 1: The podosomes at the center with large heights (above the steady-state core height, denoted with $a$) depolymerize, release G-actin, and increase the local G-actin concentration. Then, G-actin diffuses outward, altering the G-actin concentration near the podosome $b$ located away from the center, thus disrupting the previous balance between actin consumption and release. Podosome $b$ starts growing in turn.

Phase 2: As the height of podosome $a$ decreases and reaches its minimum value, podosome $b$ reaches its maximum height and begins to depolymerize, sending G-actin further outwards to activate the oscillation of podosome $c$. The radial wave pattern forms and propagate.

Overall, the oscillations of individual podosomes constantly change the local actin concentration, and the uneven spatial distribution of actin concentration drives diffusion in the podosome cluster, leading to the wave patterns in podosome core heights. As this phenomenon involves coordination of actin diffusion, mechanical forces, and biochemical signaling, we have coined the term chemo-mechanical diffusion waves to describe the collective dynamics of podosome clusters.

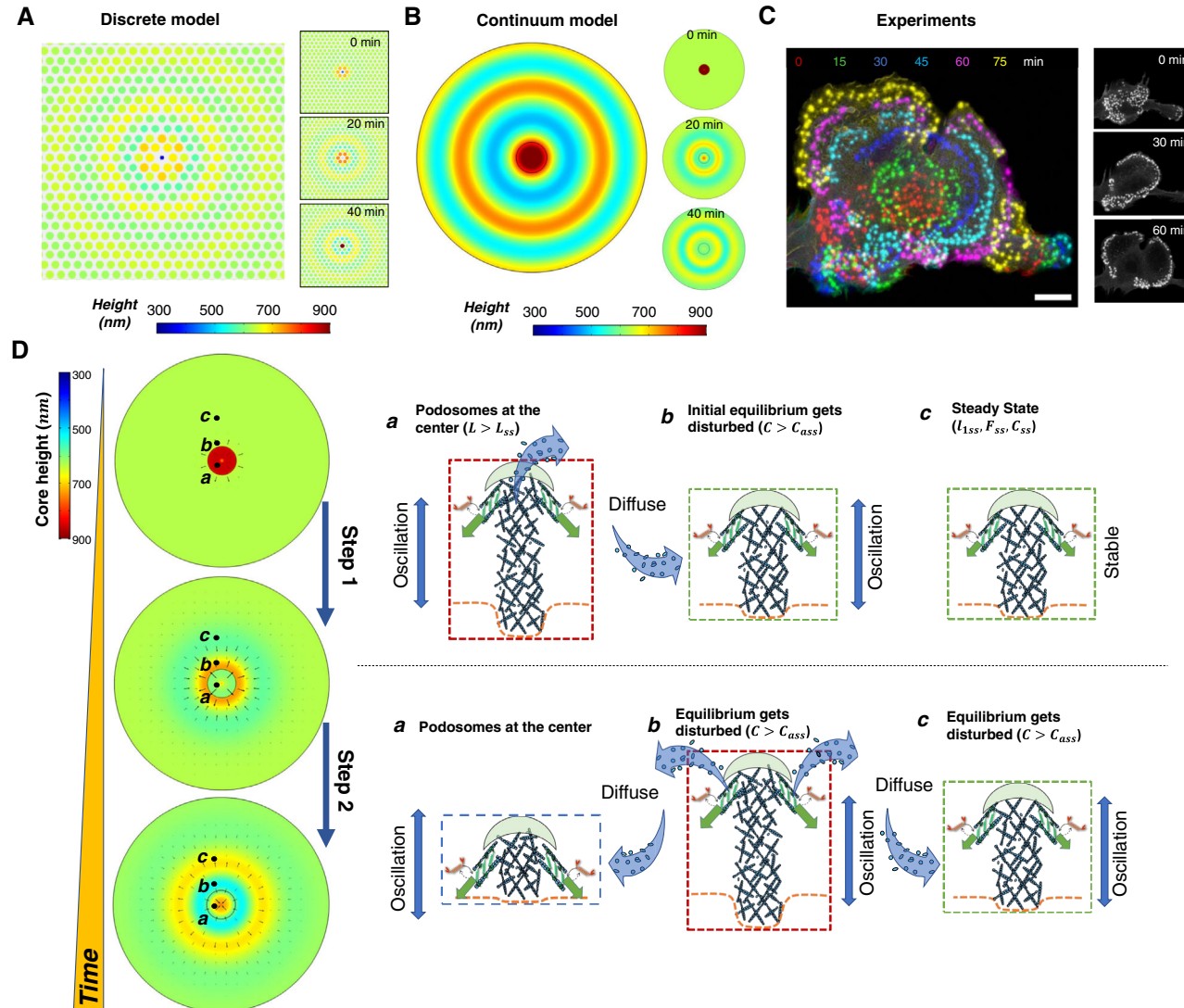

**Fig. 3 | Mechanism of radial wave formation in podosome clusters.** The simulated heights in the podosome cluster using (**A**) a discrete model of podosomes and (**B**) a continuum model showing radial wave patterns. Each circle in panel **A** represents an individual podosome with the size and color representing its height. Different colors in panel **B** represent the podosome heights. **C** DCs transfected with LifeAct-RFP showing the propagation of radial waves. Colors denote the fluorescence-labeled podosomes at different times (Figure adapted from our previous study[10]). The insets in **A**–**C** show the radial wave patterns at different times. Scale bar, 10 μm. **D** Schematic showing the mechanism for the propagation of radial waves. The black arrows in the left panels indicate the flow of G-actin.

## Theory quantitatively predicts the periods and wavelengths of chemo-mechanical waves

Most chemo-mechanical diffusion waves observed in experiments show some degree of randomness in the wave patterns. To simulate the random waves (Supplementary Movie 1), we assume a randomly distributed podosome heights in the beginning. Both the discrete and continuum approaches capture the propagation of random waves in the core heights of podosome clusters (Fig. 4A, D, and Supplementary Movies 2–3). Plots of the dynamics of podosome core heights from the discrete model show that the dynamics of neighboring podosomes are strongly correlated, while the dynamics of distant podosomes are generally uncorrelated (Fig. 4B). This prediction is further supported by our experiments based on LifeAct-RFP intensity traces of neighboring and distant podosome pairs in the cluster (Fig. 4C). Furthermore, by using kymograph analysis (which displays a series of images of the indicated rectangular area in Fig. 4D, E over time), we found that the simulated peaks of podosome heights travel within the cluster, forming wave patterns (Fig. 4D). The simulated patterns span regions with a typical length scale of ~3 μm and persist with a lifetime of ~5 min,

in agreement with the LifeAct fluorescence patterns in our experiments (Fig. 4E). Similar to the radial waves, these random wave patterns originate from the coupling of individual podosome oscillations and actin diffusion. By applying a small perturbation $\sim e^{iqr+i\omega t}$ to Eqs. (5)–(7) (refer to Methods section), we obtain the angular wavenumber of the damped chemo-mechanical diffusion waves:

$$q \approx \sqrt{-\frac{i\omega + \omega_p}{D_a}}. \tag{4}$$

Here $\omega = 2\pi/T$ is the radial frequency of podosome oscillations and $\omega_p = \mu\beta V_d/V_{pms}$ represents the effective G-actin consumption (or production) rate. The real part of the wavenumber yields the wavelength $\lambda = 2\pi/\mathscr{R}_e(q)$, which can be approximated as a diffusion length scale $\lambda \sim \sqrt{D_a T}$ for a small G-actin consumption rate ($\omega_p \ll \omega$). The imaginary part characterizes the damping of the waves (refer to Supplementary Note 5).

Next, to quantitatively compare our simulations with experiments, we extracted the wave periods and wavelengths from our

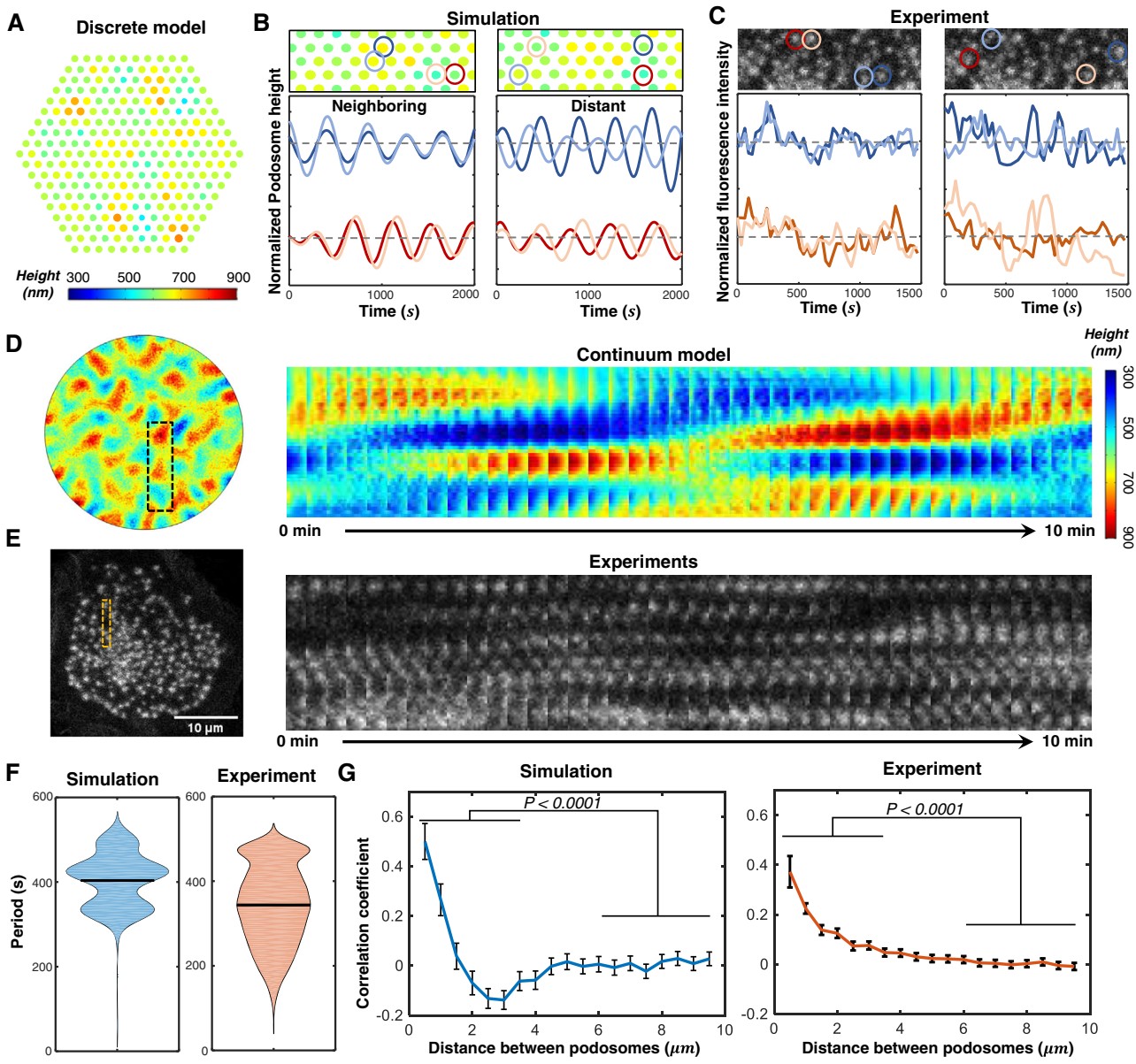

**Fig. 4 | Model quantitively predicts the wavelengths and periods of random waves. A** Simulated heights in the podosome cluster using the discrete approach showing random wave patterns. Each circle represents an individual podosome with its size and color representing height. Plots of (**B**) simulated podosome heights and (**C**) LifeAct-RFP intensity over time for (left panel) two representative neighboring and (right panel) distant podosome pairs. The representative podosome pairs are marked in the movie snapshots in the top panels. Data are normalized by (**B**) mean height or (**C**) mean RFP intensity. **D** Simulated heights of a podosome cluster using the continuum approach with the kymograph of the indicated rectangular area. The color legend is the same as the legend in **A**. **E** A representative LifeAct-RFP transfected DC with the kymograph of the indicated rectangular area. Scale bar, 10 μm. **F** The oscillation periods of individual podosomes extracted by fast Fourier transformation for (left panel) simulations and (right panel) experimental results. Black lines indicate the mean values. **G** (Left panel) Average correlation coefficient of simulated core heights and (right panel) experimentally measured actin intensity fluctuations as a function of podosome pair distance ($n = 29642$ podosome pairs for experiments, $n = 18489$ podosome pairs for the simulated podosome cluster). Data are presented as mean values +/− SEM, one-sided Student's $t$ test was used.

podosome simulations and LifeAct-RFP intensity dynamics. Using Fast Fourier Transformations (FFT) to process the experimentally measured LifeAct-RFP intensity fluctuations of individual podosomes, we extracted the oscillation periods of all podosomes in a cluster (Supplementary Fig. 5, refer to Methods). We found that the average oscillation period extracted from experiments is between five and eight minutes (~400 s), in line with our simulation results (Fig. 4F) and our analytically predicted period for individual podosome oscillation $T \approx 2\pi \sqrt{\tau_m \tau_p}$. To extract the wavelength from the simulations and experiments, we calculate the correlation coefficient, which defines

the correlation of intensity dynamics between two podosomes (refer to Methods). The correlation coefficients extracted from both the experiments and the simulations decay with the distance between the podosome pair (Fig. 4G), indicating that podosome dynamics become gradually uncorrelated as the distance increases. From these correlation coefficient plots, we obtain the characteristic length scale for chemo-mechanical diffusion waves, which agrees with our model predicted wavelength scale $\lambda \sim 3\,\mu m$. Note that the small anti-correlation region observed in the simulation (Fig. 4G) is due to the absence of noise in our simulations, which leads to periodic

(sinusoidal-like) oscillations. This anticorrelation does not appear in the experiments because the noise from myosin recruitment or polymerization can cause less periodic and more stochastic podosome patterns, eventually eliminating the anticorrelation when Gaussian noise is included in podosome dynamics. The effects of different noise patterns on the collective dynamics have been discussed in Supplementary Note 3 (Supplementary Fig. 6A–C).

## Oscillations of individual podosomes mediate the propagation of chemo-mechanical waves

Our model predicts that individual podosome oscillations cause spatial variations in G-actin concentration, which in turn drives chemomechanical diffusion waves. We next consider how wave propagation is impacted when oscillations of individual podosomes are inhibited. First, in our coarse-grained model, we disrupt the individual podosome oscillations by inhibiting myosin contractility, actin polymerization, or mechanosensitive signaling feedback (as we have shown in Fig. 2B, C). The in-silico inhibition of myosin contractility (increasing $\tau_m$), actin polymerization (increasing $\tau_p$), or mechanosensitive feedback (reducing $\Gamma$) predicts the disruption the chemomechanical diffusion waves (Fig. 5A, Supplementary Fig. S7A, B, See Supplementary Note 4), since individual podosome oscillations are inhibited and therefore cannot consume or release G-actin, resulting in a spatially homogenous G-actin concentration (Fig. 5A). To further quantify the changes of chemo-mechanical diffusion waves using our theory, we can approximate the speed of wavefront propagation using $v_c = \omega/q$ (by applying a linear perturbation $\sim e^{iqr + i\omega t}$, refer to Methods section). Note that podosome diffusion waves predicted by our theory are highly dissipative and display dispersive behavior. Different from

conventional traveling waves with well-defined and constant wave speed, the predicted wavefront propagation speed depends on the oscillation frequency (as shown in Eq. (10) and Fig. 5C, refer to Methods section). As a larger oscillation period $T$ corresponds to less pronounced oscillations ($T \to \infty$ means non-oscillatory growth), our model predicts that the wave propagation speed $v_c$ decreases after the inhibition of individual podosome oscillations (arrows in Fig. 5C).

To quantitatively characterize the podosome cluster dynamics experimentally, we applied Spatio-Temporal Image Correlation Spectroscopy (STICS)[10,23]. STICS has been previously applied to study podosome oscillatory dynamics in DCs using a variety of fluorescently-labeled podosome components[10,12]. The STICS algorithm has been previously described and recent improvements are detailed in Methods section. Using STICS, we quantified the propagation of podosome height waves by leveraging its ability to tune the targeted spatial and temporal regimes, thus minimizing the effect of individual podosome lateral displacements in our measurements. This allowed us to map the direction and magnitude of podosome wave propagation across the cell, showing that the waves propagate in a dissipative manner, which results in a qualitative assessment of the height wave propagation velocity (refer to Supplementary Note 6; Supplementary Figs. 8–10 and Supplementary Movies 4–5). Next, we examined the changes in the flow velocities of DCs transfected with LifeAct-RFP after treatment with blebbistatin, cytochalasin D, or Y27632 (which have been shown to reduce the podosome oscillations in Fig. 2D) using STICS. Importantly, our experiments show that the magnitudes of F-actin flow velocities are significantly reduced after treatments (Fig. 5D, E, Supplementary Fig. 7C–E, and Supplementary Movie 4). In addition to characterizing speeds of wave dynamics by STICS, we can estimate the wave speed

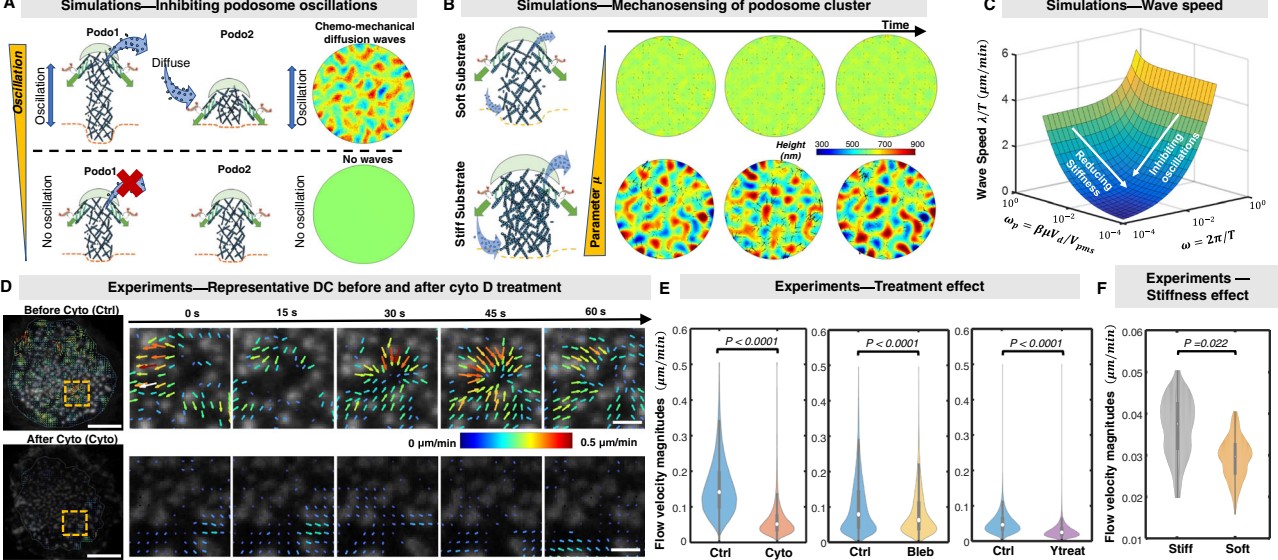

**Fig. 5 | Model predicts the inhibition of pharmacological treatments and the influence of substrate stiffness on wave propagation. A** (Left panels) Schematic and (right panels) simulations showing that inhibition of podosome oscillations disrupts chemo-mechanical diffusion waves. **B** The simulated podosome heights in a cluster plotted with time for (top panels) soft and (bottom panels) stiff substrates. Black arrows in the cluster indicate the magnitudes and directions of G-actin flow. **C** Wave propagation velocity $v_c$ plotted versus the oscillation radial frequency $\omega$ and effective G-actin consumption rate $\omega_p$. The arrow indicates the influence of pharmacological treatments that abrogate oscillations. **D** Representative dendritic cell (DC) before (top panels) and after (bottom panels) adding cytochalasin D. DCs are transfected with LifeAct-RFP and imaged using confocal microscopy with 15s frame intervals. Time series subjected to STICS analysis are plotted as vector maps. The arrows indicate flow directions, and both the size and color denote flow magnitudes. Note that the arrows with white color indicate the magnitudes up to

0.5 μm/min. Scale bar, 10 μm; Insets scale bar, 2 μm. **E** (Left panel) The velocity magnitudes measured by STICS for control (Ctrl, blue) and cytochalasin D (Cyto, red) treatment; (Middle panel) The velocity magnitudes for control (Ctrl, blue) and blebbistatin treatment (Bleb, yellow); (Right panel) The velocity magnitudes for control (Ctrl, blue) and Y27632 (Ytreat, purple) treatments. Statistically analysis was performed with a Mann–Whitney U-test. (Left panel) $n = 116043, 63188$ vectors; (Middle panel) $n = 75317, 72466$ vectors; (Right panel) $n = 195133, 199560$ vectors. (**F**) The velocity magnitudes for stiff and soft substrates measured by STICS (Data obtained from our previous work[12]). Statistically analysis was performed with a Student's t test. $n = 5$ cells for soft and stiff substrates pooled from three independent experiments. All statistical tests carried out were two-sided and boxplots are displayed using the Tukey method (center dot, median; box limits, upper and lower quartiles; whiskers, last point within a 1.5× interquartile range).

using the relation $v_c = \lambda/T$, where wavelength $\lambda$ can be obtained by fitting the correlation coefficient curves and period $T$ is extracted from the individual podosome oscillation (refer to Supplementary Note 6; Supplementary Fig. 8A, B). In agreement with our theoretical predictions (Fig. 5C), wave speed is reduced in treated cells as quantified by STICS vector magnitudes (Fig. 5E) and podosome intensity oscillation cross-correlation (Supplementary Fig. 8B). Together, these results indicate that the inhibition of podosome oscillations disrupts the chemo-mechanical diffusion waves, validating the predictions of our model.

### Podosome clusters probe substrate stiffness by modulating the G-actin consumption rate

Next, to understand mechanosensing of podosome clusters, we investigated how podosome clusters respond to substrates with varied stiffnesses. For individual podosomes, our simulations predict that higher substrate stiffness increases the steady-state protrusive force $F_{ps} = F_{p0}k_s/(k_s + k_c)\left(1 - V_d/V_{pms}\right)$, while reducing the substrate displacements $l_{1s} = F_{ps}/k_s$ (Supplementary Fig. 11A), in line with experimental evidence[7,12]. Microscopically, this increased protrusive force on stiffer substrates is due to the stronger and denser F-actin networks in the podosome core[3,8]. This mechanosensitive structural difference affects the cluster dynamics because a denser core actin (i.e., more actin filament number $N_a$) on stiffer substrates requires more G-actin to assemble and releases more G-actin upon disassembly (Fig. 5B). Since our model assumes that the consumption (release) of G-actin increases proportionally with core F-actin number (i.e., $\mu \propto N_a$), the parameter $\mu$ should also be proportional to substrate stiffness ($\mu \propto k_s$), such that the effective G-actin consumption rate increases with substrate stiffness, i.e., $\omega_p = \mu \beta V_d/V_{pms} \propto k_s$. Based on the analytical approximation for the propagation velocity $v_c$ (Eq. (10) in Methods), we found that a smaller consumption rate $\omega_p$ reduces the chemo-mechanical wave propagation speed $v_c$ (Fig. 5C). Reducing the parameter $\mu$ in our model leads to a lower propensity for wave dynamics in the podosome cluster. This is because a smaller G-actin consumption rate $\omega_p$ causes less variance (or fluctuation) in the spatial distribution of actin concentration, reducing the gradient of actin concentration and eventually decreasing the wave propagation speed $v_c$ (via Eq. (10), Fig. 5F). Recent studies have shown that the near-neighbor distance between individual podosomes can vary with stiffness[12,24]; however, this difference in the neighboring distance does not affect the wave speeds, as the podosome near-neighbor distances for substrates with different stiffnesses are below the diffusion length scale $d_0 \leq \sqrt{D_a T} \approx 3\,\mu m$ (Supplementary Fig. 6D–F, refer to Supplementary Note 3).

The above predictions are consistent with previously measured podosome cluster dynamics of live DCs seeded on polydimethylsiloxane (PDMS) substrates with different stiffness[12] showing reduced dynamics on soft substrates (Fig. 5F). To further confirm that the slow wave speeds are due to small G-actin consumption rates rather than changes in the podosome oscillation periods, we extracted the oscillation periods of individual podosomes and found that the oscillation periods for soft and stiff substrates are indeed not significantly changed, along with their amplitudes (Supplementary Fig. 11B). However, the correlation coefficient decays much faster with pair distance for podosome clusters on soft substrates, indicating a smaller chemo-mechanical wavelength for soft substrates (Supplementary Fig. 11C). Taken together, our model recapitulates the effect of mechano-sensitivity showing that stiffer substrates can enhance the propagation of chemo-mechanical diffusion waves, and that podosome clusters can serve as a mechano-sensing platform by modulating the G-actin consumption rate of individual podosomes.

## Discussion

In this study, by developing and experimentally validating a chemo-mechanical model, we elucidated the mechanisms of individual podosome oscillations and the wave-like dynamics of podosome clusters. Following a bottom-up approach, we first showed that two competing force generation processes of polymerization-driven protrusion at the core and myosin dynamics in the ring occur at similar rates, causing the vertical oscillations of individual podosomes. Next, our model reveals that the oscillatory growth of individual podosomes leads to release or consumption of G-actin locally, which in turn causes diffusion of G-actin and drives the wave-like coordination among the podosomes in a cluster. It is the first theoretical model, to our knowledge, that systematically illustrates how the vertical dynamics of individual podosomes are synchronized to form wave-like dynamics in podosome clusters of immune cells and how the podosomes in a cluster can collectively probe mechanical cues from surroundings.

For individual podosomes, the two force-generating processes—actin polymerization for core protrusive forces and myosin dynamics for ring contractile forces—are balanced in a dynamic manner. Our model shows that the core height, ring components, and protrusion forces of podosomes all oscillate with time, in agreement with the experimental results presented in this work and previous studies[7,9,10]. Our analysis reveals that the oscillatory growth of podosomes is determined by the timescale governing the core protrusion process $\tau_p$ and the timescale for myosin turnover $\tau_m$. Podosomes spontaneously oscillate when these timescales are comparable, and the oscillation period can be estimated as $2\pi\sqrt{\tau_m\tau_p} \sim 400s$. In addition to the oscillatory protrusion pattern, podosomes can exhibit monotonic growth when these timescales are not comparable, which has been validated by pharmacological treatments targeting polymerization and myosin activity. More importantly, our study demonstrates that the Rho-ROCK pathway is a critical mechanosensitive mediator of the vertical oscillations of podosomes. This mechanosensitive signaling feedback has not been emphasized in previous theoretical studies on dynamics of individual podosomes[6,7,25,26]. Here, our model predicts that the weak mechanosensitive feedback $\Gamma$ due to this pathway inhibits podosome oscillations, which has been validated by our experiments using Y27632 treatment.

Oscillatory behaviors, such as actin filaments and adaptors, have also been found in other protrusion types, such as invadopodia[22,27,28] and filopodia[29,30]. Different from invadopodia and filopodia, podosomes possess a complex conical structure in which ventral actin filaments branch from the core F-actin and connect with ring adhesion. Although these complex structural components are found in individual podosomes, the dynamics of the ring components and core actin are highly synchronized and both dynamics can be mediated by myosin contractility. Therefore, actin polymerization and myosin contractility remain the two most dominant processes governing podosome dynamics. Furthermore, previous models on the dynamics of invadopodia[22] and filopodia[29] have also shown that their oscillatory growths are caused by interplay between actin polymerization process and myosin dynamics. Taken together, these studies suggest that the ubiquitous nature of oscillations in such nonlinear cellular protrusion systems can originate from the competing force generating processes that produce protrusive (actin polymerization) and tensile forces (myosin contractility) in actin networks.

Our chemo-mechanical model shows that G-actin diffusion can synchronize the vertical height oscillations of individual podosomes in a cluster to form chemo-mechanical diffusion waves. Using discrete and continuum approaches, we simulated both the radial and random waves of the core F-actin observed in experiments. By evaluating the correlations between podosome dynamics in both our simulations and experiments, we found that individual podosome dynamics becomes gradually uncorrelated as their distances become larger than the characteristic wavelength, $\lambda$-3 μm. Our model also shows that the wave propagation speed $\lambda/T$ is regulated by the individual podosome

oscillation period $T$ and G-actin consumption rate $\omega_p$. This prediction has been further validated by our experimental results that the wave propagation becomes slower after using pharmacological treatments to inhibit the oscillations. More interestingly, our model shows that stiffer substrates enhance the wave speeds by forming stronger podosome cores to increase its G-actin consumption rate. This finding also reveals the mechanism of how podosome clusters act as a mechanosensitive platform to collectively probe the mechanical properties of surroundings. Although a regular pattern of podosomes in the cluster is assumed in our discrete model, our continuum model has generalized all the discrete cases with a near-neighbor distance below the diffusion length scale ($d_0 \leq \sqrt{D_a T}$). More complex or irregular cluster patterns, such as rosette patterns with large podosome near-neighbor distances ($d_0 > \sqrt{D_a T}$), can be easily adapted in our discrete model and discussed separately in a future study. It is also worth noting that we chose not to include the effects of dorsal filaments[3,12] that connect with adjacent podosomes in our model, because the dorsal filaments mainly apply horizontal forces on the podosome core; these horizontal forces are balanced at the core and, hence, are not considered to affect podosome dynamics.

Wave-like dynamics have emerged as a widely prevalent phenomenon in many biological systems, such as actin waves in cortex[31–33] and stress waves in epithelia expansion[34,35], and controlling these waves is found to regulate cell motility and division[33,36,37]. In view of this relationship, the wave-like dynamics in podosomes are considered essential for proper immune surveillance and migration of immune cells such as DCs. By capturing antigens and subsequently migrating to lymphoid tissues, DCs are the primary initiators of immune responses. This is not only a critical process in wound healing and inflammation[38,39], but also the first line of defense against the development and progression of cancer. As such, DCs loaded with tumor antigens have become a promising tool for therapeutic immunity against cancer[40,41]. By predicting podosome dynamics under different environmental conditions, our chemo-mechanical model can potentially be used to better anticipate the ability of DCs to migrate and probe their environment in various therapeutic settings. Future theoretical developments and computational models may build upon our chemo-mechanical model by incorporating the nonlinear properties of the microenvironment (e.g., matrix viscoelasticity, matrix heterogeneity, and topographical cues) and matrix degradation, leading to a comprehensive understanding of podosome dynamics in the context of wound healing and cancer immunotherapy.

## Methods
### Model formulation
The equations in Results section that govern the spatiotemporal dynamics of a podosome cluster can be expressed as:

$$\frac{F_{sp0}}{V_{pm}}\left(\frac{1}{k_f}+\frac{1}{k_s}\right)\frac{dF_p}{dt}+F_p=F_{sp0}\left(1-\frac{V_d}{V_{pm}}\right)+\frac{F_{sp0}}{V_{pm}}\frac{\cos(\theta)}{k_f}\frac{dF_m}{dt}, \quad (5)$$

$$\tau_m\frac{dF_m}{dt}+\left(1-\frac{\gamma}{k_f}\right)F_m=F_0+\left(\alpha-\frac{\gamma}{k_f}\right)\frac{F_p}{\cos(\theta)}, \quad (6)$$

$$\frac{\partial c_a}{\partial t}=D_a\nabla^2 c_a-\mu\bullet\left[V_{pm}\left(1-\frac{F_p}{F_{sp0}}\right)-V_d\right]. \quad (7)$$

There are three variables: the protrusive force $F_p$ ($x$, $y$, $t$), the active myosin force $F_m$ ($x$, $y$, $t$), and the G-actin concentration $c_a$ ($x$, $y$, $t$). Equations (5)–(7) describe polymerization-driven protrusion, signaling-associated myosin recruitment, and reaction-

diffusion of G-actin, respectively. Note that when the maximum polymerization speed $V_{pm} = V_{p0} + c_a\beta$ is constant, Eqs. (5) and (6) reduce to the model for individual podosome oscillations (i.e., Eqs. (1) and (2) without Gaussian noise). In steady state (i.e., when the time derivatives in Eqs. (5)–(7) vanish), we obtain the steady-state protrusive force $F_{ps} = F_{sp0}(1 - V_d/V_{pms})$, myosin force $F_{ms} = \frac{k_f}{k_f-\gamma}(F_0+(\alpha-\frac{\gamma}{k_f})\frac{F_{ps}}{\cos(\theta)})$, and actin concentration $c_{as}$, where $V_{pms} = V_{p0} + c_{as}\beta$ denotes the maximum polymerization speed in steady-state and $F_{sp0} = F_{p0}k_s/(k_c + k_s)$ is the characteristic stall force for core-actin polymerization. To solve Eqs. (5)–(7), we applied both discrete and continuum approaches using MATLAB and COMSOL packages. All the parameters used in the simulations are summarized in Supplementary Tables 1–2, and more details on the derivation and simulation can be found in Supplementary Notes 1–2.

### Linear stability analysis
To analytically obtain the wave periods and wavelength, we performed a linear stability analysis on our chemo-mechanical model. We applied small perturbations $\delta F_{ps}$, $\delta F_{ms} \sim e^{i\omega t}$ to the steady-state forces ($F_{ps}$, $F_{ms}$) in Eqs. (5) and (6), where the steady-state actin concentration $c_{as}$ and maximum polymerization speed $V_{pms}$ are assumed not to vary with time. Thus, we can write the eigenvalues that characterize the dynamics of the perturbed system:

$$\omega_{1,2}=\frac{\Gamma-1-\tau_m/\tau_p\pm\sqrt{(\Gamma-1-\tau_m/\tau_p)^2-4\tau_m/\tau_p}}{2\tau_m}. \quad (8)$$

Here $\tau_p = \frac{F_{sp0}}{V_{pms}}(\frac{1}{k_f}+\frac{1}{k_s})(1-\frac{\gamma}{k_f})$ is the timescale that characterizes polymerization-driven core protrusion, $\tau_m$ is the timescale for myosin turnover, and $\Gamma = \frac{\alpha k_f-\gamma}{k_f-\gamma}\frac{k_s}{k_s+k_f}$ is the parameter group for signaling-associated contraction of ventral actin filaments. The eigenvalues shown in Eq. (8) allow us to determine the dynamics of individual podosomes: Podosomes spontaneously oscillate only when the eigenvalues' imaginary part $\mathscr{I}_m(\omega_{1,2}) \neq 0$, i.e., $(\Gamma-1-\tau_m/\tau_p)^2 < 4\tau_m/\tau_p$. For oscillations with constant amplitudes, the real part $\mathscr{R}_e(\omega_{1,2}) = 0$, i.e., $\tau_m/\tau_p = \Gamma-1$, and the oscillation periods can be estimated as $T = 2\pi/\mathscr{I}_m(\omega_{1,2}) \approx 2\pi\sqrt{\tau_m\tau_p}$. The system becomes unstable when the real part $\mathscr{R}_e(\omega_{1,2}) > 0$, i.e., $\Gamma > 1 + \tau_m/\tau_p$.

Next, to obtain the wavelength for collective wave dynamics, we apply a small variation $\delta F_{ps}$, $\delta F_{ms}$, and $\delta c_{as}$ to the reaction-diffusion Eq. (7) when it is in steady state ($F_{ps}$, $F_{ms}$, $c_{as}$):

$$\frac{\partial\delta c_{as}}{\partial t}=\underbrace{D_a\nabla^2\delta c_{as}}_{diffusion}-\underbrace{\omega_p\delta c_{as}}_{consumption\,or\,production}+\underbrace{\mu V_{pms}\frac{\delta F_{ps}}{F_{ps0}}}_{driving\,force}, \quad (9)$$

where $\omega_p = \mu\beta(1-F_{ps}/F_{sp0}) = \beta\mu V_d/V_{pms}$ is the force-mediated effective G-actin consumption (production) rate. Equation (9) is a diffusion wave equation[42] with the oscillatory driving force term $\mu V_{pms}\frac{\delta F_{ps}}{F_{sp0}}$. By assuming the complementary solution for Eq. (9) (i.e., setting the driving force term as zero) in the form $\delta c_{as} \sim e^{i\omega t + iqr}$, we can obtain the relation between the angular wavenumber $q$ and angular frequency $\omega$, i.e., $q = \sqrt{-\frac{i\omega+\omega_p}{D_a}}$. Therefore, the wavelength can be written as $\lambda = \frac{2\pi}{|\mathscr{R}_e(q)|} = 2\pi(\frac{D_a^2}{\omega_p^2+\omega^2})^{\frac{1}{4}}\frac{1}{\cos(\varphi/2)}$, where the phase angle is $\varphi = \arctan(\omega/\omega_p) - \pi$ (refer to Supplementary Note 5). As the podosome diffusion waves predicted by our theory are highly dissipative and display dispersive behavior, there is no well-defined wave speed compared to the conventional traveling waves with a constant wave speed. Here we can approximate the wavefront

propagation speed $v_c$ for the chemo-mechanical diffusion wave speed as:

$$v_c = \frac{\lambda}{T} \approx \sqrt{D_a \omega} \left( \frac{\omega^2}{\omega_p^2 + \omega^2} \right)^{\frac{1}{4}} \frac{1}{\cos(\varphi/2)}. \tag{10}$$

Importantly the approximated equation has the ability to distinguish between the control and pharmacological treatment cases, as the treatments or substrate stiffness affect the oscillation frequency $\omega$ or effective G-actin consumption (production) rate $\omega_p$.

## Preparation of human DCs

DCs were generated from peripheral blood mononuclear cells (PBMCs)[43,44]. Monocytes were derived either from buffy coats or from a leukapheresis product, purchased at Sanquin blood bank, Nijmegen, the Netherlands. PBMCs were isolated by Ficoll density gradient centrifugation (GE Healthcare Biosciences, 30 min, 4 °C, 2100 rpm). PBMCs were washed in cold phosphate buffered saline (PBS) supplemented with 0.1% (w/v) bovine serum albumin (BSA, Roche Diagnostics) and 0.45% (w/v) sodium citrate (Sigma Aldrich). Washing steps were repeated until the supernatant was clear, which was usually after approximately five times. Next, PBMCs were seeded in plastic culture flasks for 1 h after which monocytes were isolated by washing the flasks at least four times with cold PBS. This procedure removes all non-adherent cells and leaves the monocytes adhered to the flask. After this isolation by plastic adherence, monocytes were cultured in RPMI 1640 medium (Life Technologies) supplemented with fetal bovine serum (FBS, Greiner Bio-one), 1 mM ultra-glutamine (BioWhittaker), antibiotics (100 U ml$^{-1}$ penicillin, 100 µg ml$^{-1}$ streptomycin, and 0.25 µg ml$^{-1}$ amphotericin B, Gibco) for six days, in a humidified, 5% CO2-containing atmosphere. During these six days, DC differentiation was induced by addition of IL-4 (500 U ml$^{-1}$) and GM-CSF (800 U ml$^{-1}$) to the culture medium. Medium, supplemented with cytokines, was refreshed at day three and at day five or day six, cells were collected and reseeded onto coverslips or imaging dishes.

To investigate the effect of substrate stiffness, cells were seeded on dishes spincoated with PDMS (Sylgard 184, Dow Chemicals; base-curing ratio 1:20 = stiff, -800 kPa; 1:78 = soft, -1 kPa)[12]. The PDMS mixtures were then used to coat WillCo-dishes (WillCo Wells B.V.) by spin coating 150 µl silicone mixture at 3100 rpm for 2 min, resulting in thin (10–20 µm), high-resolution microscopy-compatible layers of PDMS. Finally, the medium was replaced by a medium supplemented with 10% (v/v) FCS and antibiotics for up to 24 h. Prior to live-cell imaging, cells were washed with PBS and imaging was performed in RPMI without Phenol red. All live cell imaging was performed at 37 °C.

## DC transfection

Transient transfections were carried out with the Neon Transfection System (Life Technologies)[10]. Briefly, DCs were washed with PBS and resuspended in 115 µl Resuspension Buffer per $0.5 \times 10^6$ cells. Subsequently, cells were mixed with 6 µg DNA per $10^6$ cells per transfection and electroporated. Next, cells were quickly transferred to WillCo-dishes (WillCo Wells B.V.) with pre-warmed medium without antibiotics or serum and allowed to recover for 3 h at 37 °C. For the pharmacological treatments, the following inhibitors were used: cytochalasin D (2.5 µg/ml, Sigma-Aldrich), blebbistatin (20 µM, Sigma-Aldrich), and Y27632 (20 µM, Selleck). All live cell imaging was performed at 37 °C.

## Image analysis

All image analysis was performed using Fiji[45] and the data were subsequently processed using MATLAB (MathWorks). Prior to analysis, movies were registered, allowing translation only, using the StackReg plugin[46] and bleach correction was performed using the "Histogram Matching" option. For amplitude ratio analysis, podosome centers were detected in the first frame of the image series with a slightly adapted ImageJ algorithm that was developed previously[47]. Briefly, before median filter processing, we first performed an unsharp mask filter (radius: 4.5 pixels, weight: 0.8) and this sequence was repeated twice after which a background subtraction was performed. Podosome centers were subsequently determined using the ImageJ maxima finder (prominence: 1000). Fluorescence intensity was measured in a circular region of interest (ROI) with a radius (-0.5 µm) over the course of the movie.

## Extracting oscillation amplitudes, periods, and correlation coefficients

To extract the oscillation amplitude and frequency, the measured fluorescence intensity dynamics were converted from the time domain to frequency domain using Fast Fourier transformation (FFT). The largest four peaks in the spectrum were chosen to calculate the amplitudes and frequencies of the podosome dynamics (Supplementary Fig. 5). The amplitude ratio was calculated as the ratio of the averaged amplitudes and the mean intensity. Next, to estimate the spatial correlation of two podosomes in a cluster, Pearson correlation coefficients were calculated for the time series of fluorescently tagged LifeAct over a 25 min time window. By denoting the LifeAct intensity dynamics of two podosomes as vectors $A$ and $B$ of length $N$, the correlation coefficient is written as:

$$\rho(A,B) = \frac{1}{N-1} \sum_{i=1}^{N} \left( \frac{A_i - \mu_A}{\sigma_A} \right) \left( \frac{B_i - \mu_B}{\sigma_B} \right), \tag{11}$$

where $\mu_A$ and $\sigma_A$ are the mean and standard deviation of $A$, respectively, and $\mu_B$ and $\sigma_B$ are the mean and standard deviation of $B$, respectively. The correlation has a value between −1 and 1, which correspond to in phase and out-of-phase correlation, respectively. The *corrcoef* function in MATLAB (MathWorks) is used to calculate the correlation coefficient.

## STICS analysis

We performed Spatio-Temporal Image Correlation Spectroscopy (STICS)[23] on confocal laser scanning microscopy (CLSM) time series of DCs transiently expressing fluorescently tagged LifeAct acquired with a 15 s lag between frames and 0.14 µm/pixel. Movies were first registered using a Fourier-space image registration algorithm, reducing jitter to a sub-pixel scale. For F-actin intensity wave propagation quantification, a Gaussian filter (sigma = 7) was applied to the movies using Fiji/ImageJ. Movies were then analyzed using an improved and extended version based on the original STICS[23]; a detailed description of the new algorithm and a graphical user interface will be published separately. Briefly, we divide each image into small spatial windows or regions of interest (ROIs) of $32 \times 32$ pixels (4.48 µm × 4.48 µm) and overlap adjacent ROIs so that the distance between them is four pixels (0.56 µm) in the horizontal and vertical directions to map the entire field of view with oversampling in space. We then divide the time series into overlapping temporal windows or times of interest (TOIs) with a duration of 5 frames (1:15 min) with adjacent TOIs delayed by one frame (15 s) to cover the entire image series with oversampling in time. We then calculate spatiotemporal correlation functions (CFs) for each ROI × TOI. The central region (pixels within a radius of 8 pixels, 1.12 µm from the center) of each resulting CF is then fit with a symmetrical Gaussian function. The coordinates of the Gaussian peak center are extracted and the displacements in $x$ and $y$ of the Gaussian peak as a function of the time delay τ is fit with a linear model for time lags up to τ = 5. STICS relies on the assumption that, within a ROI and during a

given TOI, the dynamics in the time series behave uniformly and linearly. To ensure high-quality vectors, the linear fit is thus required to have a coefficient of determination ($R^2$) greater than 0.9, and the maximum number of time lags is adjusted to satisfy this condition. Vectors magnitudes and directions are thus obtained and subjected to a series of quality control filters (refer to Supplementary Note 6 for more details) to discard any spurious vector. For visualization purposes, only high-quality vectors that satisfy all quality control requirements and that were generated from more than 3/5 displacement data points are shown. Vectors are scaled to the physical displacement that would be expected in the TOI.

## Statistics and reproducibility

Statistical analyses were performed using MATLAB (MathWorks), and significance was determined using a 95% confidence interval. Data were compared using the one-way ANOVA with Benjamini–Hochberg Procedure, Mann–Whitney U-test, or Student test as indicated in figure legends, and the sample sizes $n$ and $p$ values are given in figure legends. Box plots indicate median (middle white dot), 25th, 75th percentile (box) and last point within a 1.5× interquartile range (whiskers). All experiments were repeated independently at least thrice, except where noted.

## Reporting summary

Further information on research design is available in the Nature Portfolio Reporting Summary linked to this article.

## Data availability

All the data supporting the findings described in this study are archived and publicly available via 4TU.Research Data (https://data.4tu.nl/datasets/b10471d8-8823-42d8-9768-a01dead6b25b/1). Source data are provided with this paper.

## Code availability

MATLAB and COMSOL files used in this work are openly available on Github (https://github.com/gongze/Podosome_wave).

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

## Acknowledgements

This work was supported by National Cancer Institute awards R01CA232256 and U54CA261694 (Z.G., V.B.S.); National Institute of Biomedical Imaging and Bioengineering awards R01EB017753 (V.B.S.) and R01EB030876 (V.B.S.); NSF Center for Engineering Mechanobiology Grant CMMI-154857 (Z.G., V.B.S.); NSF Grants MRSEC/DMR-1720530 and DMS-1953572 (V.B.S.); Natural Sciences and Engineering Research Council of Canada Discovery Grant RGPIN-2017-05005 (P.W.W.). Experimental work was supported by Intramural funding from Radboud University Medical Center (A.C. and K.D.). Quebec Fonds de Recherche Nature et Technologies B2X-318078 (R.M.R.) and McGill Quantitative Life Sciences Excellence Award Program (R.M.R.).

## Author contributions

Z.G. and V.B.S. conceived the chemo-mechanical model and carried out the theoretical and numerical analyses. A.C. and K.D. conceived the experiments. K.D. performed the experiments. Z.G., K.D. and R.M. analyzed the experimental data. Z.G., K.D., R. M., A.C., P.W.W., and V.B.S. wrote the paper.

## Competing interests

The authors declare no competing interests.
