## [Peer Review File · Nature Communications]

REVIEWER COMMENTS

Reviewer #1 (Remarks to the Author):

The article by Gong. et al. presents an interesting model for the collective dynamics of Immune cell podosomes.

The model is able to explain the oscillations of individual podosomes as well as the propagation of radial waves (already observed by the authors in a previous paper) or random waves.

If the model seems based on rather reasonable hypotheses and seems to have been robustly tested (even if I do not feel entitled to completely judge this part), I have however problems with the robustness of the experimental analysis to which the model is confronted.

Main comment/problem

In the introduction it is stated:

"we extracted the wavelengths, periods and speeds of chemo-mechanical waves"

If I could agree that the authors are measuring a frequency (even if certainly not really accurate (bias from discrete FFT and short timespan)

I do not agree on the fact that they are measuring a wavelength, this is appearing also in the title of Fig. 4:

"Model quantitatively predicts the wavelengths...", and the only thing that I see on the figure that could tell me

something on wavelength are the kymographs on D and E but these are not measurements....

In the same way, I really do not understand how the STICS measurements could be good measurements for the phase speed.

I have many problems with the STICS analysis.

From my understanding STICS could be a more accurate version of PIV for an object which would move at a constant speed

over several frames (but here watching movie S1 it appears it is far from being the case).

To apply these kind of analyses you have to assume that an object (pattern) could be tracked between two consecutive frames,

this object or pattern should not differ too much during the acquisition time.

This is not what I see on the movies, intensity is varying a lot of course while podosome are appearing/disappearing, so I don't

see how a speed can be measured. Moreover, let's say it could still be feasible on certain locations under the cells, then

the algorithm should give for example a quantity judging of the quality of the analysis (and in particular of the linear fit that is used to get the speed), so that we could discard the bad quality points and remove noise in the data (that appeared to be huge seeing sup movie 4 where we see plenty of white arrows, and also arrows outside of the cell of the same kind of amplitude than below the cells).

Finally, I have a conceptual problem with this approach : Factin in a single podosome should not be moving (contrary to G actin that we can not see).
Am I wrong? How these measurements done on F-actin can tell us something on the phase speed of the wave?

In any case, in regards to the weak experimental proofs on the accuracy of the model I would change the verb used in the title "orchestrate" with something less affirmative like "is compatible" or "may orchestrate"...

Intermediate comments

- * For testing the different rigidities, the authors used syldgard 184, while it is fine to use for high rigidity it is well known that it does not behave as an elastic solid at a ratio 1:20 but rather as a viscoelastic material, this is annoying if one wants to test the effect of rigidity only. For this purpose it is better to use DOW CORNING TORAY CY 52-276 A and B or polyacrylamide gels for example
- In the same way, the thickness of 10-20 μm may be a little too small if one wants the cell not to feel the hard coverslip beneath the gel.
- * Fig 2 B curves are supposed to be representative of the different conditions, and it clearly appears that the amplitude ratio is much smaller in the case of blebbistatin than in control (similar to cyto), and this is absolutely not apparent in Fig 2 C, where the statistics seem rather similar to control (the *** difference is even bothering). What is wrong B or C?
- * Fig 4 G Could you explain why on the simulation we observe a clear anticorrelation, which is absent from the experiment?
- * In the model the polymerization speed is taken linear with the force. Is it a reasonable assumption shouldn't the dependency be exponential?

Minor comments:

- * Fig 4 B and C, why are there four timecurves (and not two) below each images whereas only two podosomes are encircled?
- * line 605 the authors define the amplitude ratio as the ratio of averaged amplitudes... Are they talking here about the standard deviation of the signal?
- * There is no scale bar on Fig 5 and panel D is really small
- * the imaginary number i is missing in equation 8
- * line 605 I guess the authors are referring to Fig. S5 and not S4.

Reviewer #2 (Remarks to the Author):

In this paper an elaborate model is developed to describe the observed oscillation of the individual podosome and the wavelike coupling between podosomes in a cluster. The experimental characterization of the dynamics of the podosomes is the focus of the group of one of the authors (Koen van den Dries). In general, the model predictions are found to be in line with experimental observations.

The authors claim that they are the first to describe a theoretical model for the spatiotemporal dynamics in clusters of podosomes. This appears to be the case indeed.

This paper has a high potential but in its current state some concerns can be expressed.

1. The emphasis of this work is clearly on the development of the model. I do realize the challenge the authors are facing in finding a proper balance between conveying the message in an accessible way on one hand, and describing the underlying model to a sufficient detail, on the other hand. However, I found it difficult to read this paper as one has to go back and forth between the main text, the end of the main text and the supporting information to fully grasp this nontrivial model. Model parameters are defined in the main text and I had to make a list of symbols and notations for myself. Table S1 does contain several model parameters but not all of them are presented. The authors could help the reader by at least providing a list of symbols. Furthermore, some relations are within the text without equation number and also these have to be written down to keep track of all definitions and relations. I suggest that the authors provide a text in the SI with a logical sequence of definitions, relations and equations combined with the current notes in the SI. In doing so the model and the predictions

by the model will be conveyed in a clear and transparent way. This will be appreciated by the readers even when there are repeats between the main text and the full description of the model in the SI. Furthermore, when the full model is described in detail in the SI the initial and boundary conditions for the various differential equations can be mentioned.

2. Concerning the Table S1: in the “Reference” column there five different types of entries: just a reference number, “adjusted”, “fitting parameter”, “estimated” and “measured”. It is not clear how “adjusted” should be interpreted as this indication appears in combination with a reference number. The diffusion coefficient is a fitting parameter and the reader is referred to SI Note 2. It is not made clear how the value in Table S1 is obtained. Also for other entries the indication “fitting” is used; does this mean fitting a model to experimental data? For the very last entry “measured” is used. However no reference is given and I assume that the value of this parameter is determined in the current work. However, it is not clear how. I will be appreciated if the authors could be more specific wrt the entries in the last column of Table S1. The listed values are crucial in the prediction power of the model. The authors should give an indication of the uncertainty of the fitted parameters.

3. The details of the actual experiments are rather limited. The reader can assume that the details of the instrument and the actual settings are similar the approach described in the paper by Maddens et al. [ref. 10] where also dendritic cells are investigated and the twSTICS is used. However, this should be explicitly mentioned. The same oversampling in space and time are used as in the Maddens et al paper. So, one can assume that all other details are the same but it would good that this is explicitly stated in the current manuscript, otherwise the specific settings used in the current should be described to sufficient detail (in SI). The abbreviations twSTICS, TOI and ROI should be clarified in the current manuscript. The twSTICS experiments run over a substantial period of time. Are the data fully corrected for motions of the cell body as such by the Fourier filter?

4. In order to be able to repeat the experiments: what are the actual concentrations of

blebbistatin, cytochalasin D, and Y27632 used in the experiments? This should also be related to the number of cells and the area of the sample chamber.

5. In the model the angle theta between the ventral actin filaments and the core F-actin is assumed to be constant. No justification is given for this assumption. As theta is taken to be constant the high correlation between the length of the ventral actin filament and the ring diameter (Fig S2, panel A) in the simulations is to be expected.

The experimentally obtained high correlation between the actin and vinculin related fluorescence signals (Fig. S2, panel C) can be taken as evidence that the angle theta can be considered to be constant. If the authors agree with this reasoning then this can be explicitly mentioned as experimental evidence that it is justified that theta can be considered to be constant. Otherwise, they have to justify the assumption of a constant value for theta. In Table S1 it is mentioned that the value of theta is estimated in this work. I might be mistaken but where do the authors describe the determination of the actual value of the angle theta? The authors could be more specific about the determination procedure of the value of this angle.

6. In the main text one can read: “Although the spontaneous oscillations in region I are damped, the random fluctuations (i.e., Gaussian noise contributions $\chi_m(t)$ and $\chi_p(t)$ in Eqs.1-2) from myosin recruitment or core-actin polymerization constantly push the podosome away from the steady state; this initiates excursions in the phase space of core height and myosin force, leading to persistent oscillations (Fig. S1A and S1B)”.

How strong must the noise contributions be in order to maintain the oscillations? This question is also to be seen within the context of the following. In Fig. S1, panel B, myosin force versus core height, $R_{\text{sigma}} = 0.1$ seems to generate the same pattern as for $R_{\text{sigma}} = 0$. However, the myosin force versus core height graph in Fig. 2, panel D, rather refers to $R_{\text{sigma}} = 0.01$ in Fig. S1, panel B. Therefore one expect that $R_{\text{sigma}} = 0.01$ would be an appropriate choice. However, in SI Note 2 it is stated that: “The Gaussian noises are generated by MATLAB randn function $\chi_m(t)$, $\chi_p(t) \sim N(0, \sigma)$, where their standard deviation are estimated around 1%-10% of the characteristic protrusive force ($\sigma(\chi_m) = 0.05F_p0 \sim 5000 \text{ pN}$) and initial

polymerization speed ($\sigma(\chi p) = 0.05Vp0^{\sim}5 \text{ nm/s}$), respectively". This is 5%. This value is mentioned in the figures and in the Table S1 one finds 10%. I am puzzled by this. Please clarify.

7. In the main text it is stated that: "Our model predicts that individual podosome oscillations cause spatial variation in G-actin density, which in turn drives chemomechanical diffusion waves". What are the required minimal concentration changes in

G-actin to stimulate protrusion growth at a distant point? The changes in G-actin concentration are correlated with the distance between the podosomes in the cluster.

The distance between the podosomes changes with the stiffness of the substrate

[Collin O, Tracqui P, Stephanou A, Usson Y, Clément-Lacroix J, Planus E.

Spatiotemporal dynamics of actin-rich adhesion microdomains: influence of substrate flexibility. *J Cell Sci.* 2006 May 1;119(Pt 9):1914-25. doi: 10.1242/jcs.02838. PMID:

16636076]. In the current paper experiments are performed on substrates with a different stiffness and the wave propagation has been determined and compared with the predictions of the model. In Table S1 only a single value for the distance between the podosomes in the cluster is considered. Why did the authors not consider a different distance when changing the stiffness of the substrate? In general one can ask the question how the distance between the podosomes influences the velocity of the wave over the cluster. I would be happy to learn from the answer by the authors.

8. I wonder what the effect of the lipids are in the dynamics of the wave reaching the various points where the podosomes will undergo their changes. The growth of the podosomes in the cluster must be accompanied by a substantial growth in membrane area in the podosome cluster. Can the rate of lipid recruitment due to the change in lipid area have a modulatory effect on the spatiotemporal characteristics considered in this work? Please comment.

9. Fig. 4, panel G: a negative correlation at intermediate distance between the podosomes appears in the simulation that is absent in the experiment. The authors should comment on this.

10. Autocorrelation of the changes in fluorescence intensity (control situation) can reveal

the frequency of the oscillations. This can directly be compared with the predicted value. Is there a reason why the authors did prefer not to do so?

11. The combination of Eq. 10: $vc = \lambda T \approx \sqrt{Da\omega} (\omega^2 \omega_p^2 + \omega^2)^{-1/4} \cos(\theta/2)$ and the relation $\omega_p = \mu\beta Vd/Vpms \propto ks$ indicates that the phase velocity decreases with increasing stiffness. This seems to be in contradiction with Fig. 5-C and Fig.5-F. I am puzzled by this. Please clarify.

12. How does the directionality of the velocity vectors result from the equations?

Diffusion is in all directions. Can the authors give a plausible explanation for the observed directionality and indicate how this directionality will result from the coupled differential equations.

13. The podosome cluster radius R_0 is mentioned in Table S1 where it is also indicated that this parameter is measured in experiments. Where is this cluster radius used in the model and how is the value actually determined in the experiments?

14. In the discrete model a regular pattern of podosomes is assumed. I do understand that this is within a model but how close to the reality is this approximation as, in general, podosomes arrange in different patterns?

Minor comments

1. In the main text one can read: "Based on the phase diagram (Fig. 1C), we found that when we inhibit actin polymerization, core-actin growth becomes much slower compared to myosin dynamics, i.e., $\tau_p/\tau_m \ll 1$, causing non-oscillatory podosome behaviors (Fig. 1C, red arrow). When the myosin contractility is inhibited, myosin turnover becomes very slow such that $\tau_m/\tau_p \gg 1$, leading to non-oscillatory growth in podosomes (Fig. 1C, yellow arrow)". The marked inequalities express the same conditions wrt the characteristic times. Please correct.

2. Fig S2. Oscillatory behaviours in podosome core and ring. The labels of the panels are not correctly indicated in the legend to the figure. Please correct.

3. In the main text one can read: "The largest four peaks in the spectrum were chosen to calculate the amplitudes and frequencies of the podosome dynamics (Fig. S4)". This should be Fig. S5.

4. Table S1: FM0 should be Fm0.

5. Fig. S1, panel B: the orange and yellow colour are used to denote the same trace.

Please adjust the colours.

In this paper an elaborate model is developed to describe the observed oscillation of the individual podosome and the wavelike coupling between podosomes in a cluster. The experimental characterization of the dynamics of the podosomes is the focus of the group of one of the authors (Koen van den Dries). In general, the model predictions are found to be in line with experimental observations.

The authors claim that they are the first to describe a theoretical model for the spatiotemporal dynamics in clusters of podosomes. This appears to be the case indeed.

This paper has a high potential but in its current state some concerns can be expressed.

1. The emphasis of this work is clearly on the development of the model. I do realize the challenge the authors are facing in finding a proper balance between conveying the message in an accessible way on one hand, and describing the underlying model to a sufficient detail, on the other hand. However, I found it difficult to read this paper as one has to go back and forth between the main text, the end of the main text and the supporting information to fully grasp this nontrivial model. Model parameters are defined in the main text and I had to make a list of symbols and notations for myself.

Table S1 does contain several model parameters but not all of them are presented. The authors could help the reader by at least providing a list of symbols. Furthermore, some relations are within the text without equation number and also these have to be written down to keep track of all definitions and relations. I suggest that the authors provide a text in the SI with a logical sequence of definitions, relations and equations combined with the current notes in the SI. In doing so the model and the predictions by the model will be conveyed in a clear and transparent way. This will be appreciated by the readers even when there are repeats between the main text and the full description of the model in the SI. Furthermore, when the full model is described in detail in the SI the initial and boundary conditions for the various differential equations can be mentioned.

2. Concerning the Table S1: in the "Reference" column there five different types of entries: just a reference number, "adjusted", "fitting parameter", "estimated" and "measured". It is not clear how "adjusted" should be interpreted as this indication appears in combination with a reference number. The diffusion coefficient is a fitting parameter and the reader is referred to SI Note 2. It is not made clear how the value in Table S1 is obtained. Also for other entries the indication "fitting" is used; does this mean fitting a model to experimental data? For the very last entry "measured" is used. However no reference is given and I assume that the value of this parameter is determined in the current work. However, it is not clear how. I will be appreciated if the authors could be more specific wrt the entries in the last column of Table S1. The listed values are crucial in the prediction power of the model. The authors should give an indication of the uncertainty of the fitted parameters.

3. The details of the actual experiments are rather limited. The reader can assume that the details of the

instrument and the actual settings are similar the approach described in the paper by Maddens et al. [ref. 10] where also dendritic cells are investigated and the twSTICS is used. However, this should be explicitly mentioned. The same oversampling in space and time are used as in the Maddens et al paper. So, one can assume that all other details are the same but it would be good that this is explicitly stated in the current manuscript, otherwise the specific settings used in the current should be described to sufficient detail (in SI). The abbreviations twSTICS, TOI and ROI should be clarified in the current manuscript. The twSTICS experiments run over a substantial period of time. Are the data fully corrected for motions of the cell body as such by the Fourier filter?

4. In order to be able to repeat the experiments: what are the actual concentrations of blebbistatin, cytochalasin D, and Y27632 used in the experiments? This should also be related to the number of cells and the area of the sample chamber.

5. In the model the angle θ between the ventral actin filaments and the core F-actin is assumed to be constant. No justification is given for this assumption. As θ is taken to be constant the high correlation between the length of the ventral actin filament and the ring diameter (Fig S2, panel A) in the simulations is to be expected. The experimentally obtained high correlation between the actin and vinculin related fluorescence signals (Fig. S2, panel C) can be taken as evidence that the angle θ can be considered to be constant. If the authors agree with this reasoning then this can be explicitly mentioned as experimental evidence that it is justified that θ can be considered to be constant. Otherwise, they have to justify the assumption of a constant value for θ . In Table S1 it is mentioned that the value of θ is estimated in this work. I might be mistaken but where do the authors describe the determination of the actual value of the angle θ ? The authors could be more specific about the determination procedure of the value of this angle.

6. In the main text one can read: "Although the spontaneous oscillations in region I are damped, the random fluctuations (i.e., Gaussian noise contributions $\chi_m(t)$ and $\chi_p(t)$ in Eqs.1-2) from myosin recruitment or core-actin polymerization constantly push the podosome away from the steady state; this initiates excursions in the phase space of core height and myosin force, leading to persistent oscillations (Fig. S1A and S1B)".

How strong must the noise contributions be in order to maintain the oscillations? This question is also to be seen within the context of the following.

In Fig. S1, panel B, myosin force versus core height, $R_{\sigma} = 0.1$ seems to generate the same pattern as for $R_{\sigma} = 0$. However, the myosin force versus core height graph in Fig. 2, panel D, rather refers to $R_{\sigma} = 0.01$ in Fig. S1, panel B. Therefore one expects that $R_{\sigma} = 0.01$ would be an appropriate choice. However, in SI Note 2 it is stated that: "The Gaussian noises are generated by MATLAB randn function $\chi_m(t), \chi_p(t) \sim N(0, \sigma)$, where their standard deviation are estimated around 1%-10% of the characteristic protrusive force ($\sigma(\chi_m) = 0.05F_p \sim 5000 \text{ pN}$) and initial polymerization speed ($\sigma(\chi_p) = 0.05V_p \sim 5 \text{ nm/s}$), respectively". This is 5%. This value is mentioned in the figures and in the Table S1

one finds 10%. I am puzzled by this. Please clarify.

7. In the main text it is stated that: "Our model predicts that individual podosome oscillations cause spatial variation in G-actin density, which in turn drives chemomechanical diffusion waves". What are the required minimal concentration changes in G-actin to stimulate protrusion growth at a distant point? The changes in G-actin concentration are correlated with the distance between the podosomes in the cluster.

The distance between the podosomes changes with the stiffness of the substrate [Collin O, Tracqui P, Stephanou A, Usson Y, Clément-Lacroix J, Planus E. 3 Spatiotemporal dynamics of actin-rich adhesion microdomains: influence of substrate flexibility. J Cell Sci. 2006 May 1;119(Pt 9):1914-25. doi: 10.1242/jcs.02838. PMID: 16636076].

In the current paper experiments are performed on substrates with a different stiffness and the wave propagation has been determined and compared with the predictions of the model. In Table S1 only a single value for the distance between the podosomes in the cluster is considered. Why did the authors not consider a different distance when changing the stiffness of the substrate? In general one can ask the question how the distance between the podosomes influences the velocity of the wave over the cluster. I would be happy to learn from the answer by the authors.

8. I wonder what the effect of the lipids are in the dynamics of the wave reaching the various points where the podosomes will undergo their changes. The growth of the podosomes in the cluster must be accompanied by a substantial growth in membrane area in the podosome cluster. Can the rate of lipid recruitment due to the change in lipid area have a modulatory effect on the spatiotemporal characteristics considered in this work? Please comment.

9. Fig. 4, panel G: a negative correlation at intermediate distance between the podosomes appears in the simulation that is absent in the experiment. The authors should comment on this.

10. Autocorrelation of the changes in fluorescence intensity (control situation) can reveal the frequency of the oscillations. This can directly be compared with the predicted value. Is there a reason why the authors did prefer not to do so?

11. The combination of Eq. 10: $vc = \lambda T \approx \sqrt{Da\omega} (\omega^2 \omega p^2 + \omega^2)^{-1/4} \cos(\theta/2)$ and the relation $\omega p = \mu \beta V d / V p m s \propto ks$ indicates that the phase velocity decreases with increasing stiffness.

This seems to be in contradiction with Fig. 5-C and Fig.5-F. I am puzzled by this.

Please clarify.

12. How does the directionality of the velocity vectors result from the equations? Diffusion is in all directions. Can the authors give a plausible explanation for the observed directionality and indicate how this directionality will result from the coupled differential equations.

13. The podosome cluster radius R_0 is mentioned in Table S1 where it is also indicated that this parameter is measured in experiments. Where is this cluster radius used in the model and how is the value actually determined in the experiments?

14. In the discrete model a regular pattern of podosomes is assumed. I do understand that this is within a model but how close to the reality is this approximation as, in general, podosomes arrange in different patterns?

Minor comments

1. In the main text one can read: “Based on the phase diagram (Fig. 1C), we found that when we inhibit actin polymerization, core-actin growth becomes much slower compared to myosin dynamics, i.e., $\tau_p/\tau_m \ll 1$, causing non-oscillatory podosome behaviors (Fig. 1C, red arrow). When the myosin contractility is inhibited, myosin turnover becomes very slow such that $\tau_m/\tau_p \gg 1$, leading to non-oscillatory growth in podosomes (Fig. 1C, yellow arrow)”. The marked inequalities express the same conditions wrt the characteristic times. Please correct.

2. Fig S2. Oscillatory behaviours in podosome core and ring. The labels of the panels are not correctly indicated in the legend to the figure. Please correct.

3. In the main text one can read: “The largest four peaks in the spectrum were chosen to calculate the amplitudes and frequencies of the podosome dynamics (Fig. S4)”. This should be Fig. S5.4. Table S1: FM0 should be F_{m0} .

5. Fig. S1, panel B: the orange and yellow colour are used to denote the same trace. Please adjust the colours.

RESPONSE TO REVIEWERS

We would like to thank the reviewers for providing a detailed and insightful review of our manuscript, which we believe has significantly improved the quality of our study. We have addressed each comment raised by the reviewers through significant revisions to our manuscript as described below. We hope that it is now acceptable for publication in *Nature Communications*.

(Reviewer comments in blue, our responses in plain text and edits in the revised manuscript are highlighted)

Reviewer #1 (Remarks to the Author):

The article by Gong, et al. presents an interesting model for the collective dynamics of Immune cell podosomes. The model is able to explain the oscillations of individual podosomes as well as the propagation of radial waves (already observed by the authors in a previous paper) or random waves. If the model seems based on rather reasonable hypotheses and seems to have been robustly tested (even if I do not feel entitled to completely judge this part), I have however problems with the robustness of the experimental analysis to which the model is confronted.

Response: We sincerely thank the reviewer for the thorough review of our manuscript and excellent comments/suggestions. To better answer the major questions on STICS, we added Dr. Paul W. Wiseman from McGill University, an expert in STICS and image analysis, to our team and reviewed the validity of our methods in detail. The key improvements made to the revised manuscript in response to the reviewer's comments are listed below:

- A more detailed explanation of the method for the wavelength quantification
- Elaboration of the STICS method and how it differs from PIV
- Addition of Fig. S8 for phase speed and its correlation with the measured speeds
- Additional simulations and an explanation for the anticorrelation region

Our detailed responses to specific comments follow below:

Main comment/problem:

R1.1 In the introduction it is stated: "we extracted the wavelengths, periods and speeds of chemo-mechanical waves". If I could agree that the authors are measuring a frequency (even if certainly not really accurate (bias from discrete FFT and short timespan), I do not agree on the fact that they are measuring a wavelength, this is appearing also in the title of Fig. 4: "Model quantitatively predicts the wavelengths...", and the only thing that I see on the figure that could tell me something on wavelength are the kymographs on D and E but these are not measurements....

Response: We would like to thank the reviewer for the opportunity to clarify how the frequencies and wavelengths are deduced from the experimental data. We agree that the

discrete FFT can have a bias due to sampling and noise. However, we want to point out that the discrete FFT approach has been widely used for extracting the frequency in the dynamics of fluorescence intensity (Liu et al. 2020; Thoke et al. 2017; Collin et al. 2006; Gong et al. 2021). In our study, the extracted oscillation period (300-400 s) is much larger than the time interval of 15 s, and the total timespan of 1500 s is several times larger than the oscillation period. These values indicate that the extracted frequency is unlikely to be biased from the sampling frequency and the total time span. Importantly, the measured oscillation periods are consistent with the observation in previous studies (Meddens et al. 2016; Collin et al. 2006).

With regards to the wavelength, we did extract the wavelength from the experiments, simulations, and theoretical analysis. In Fig. 4G, we evaluated the correlation coefficient of individual podosomes as a function of their separation from others. Both our simulations and experiments show that the correlation coefficient decays with the distance between the pairs, and the characteristic lengths are around $3 \mu m$. This indicates that, within this characteristic length, the dynamics of individual podosomes are highly-correlated. This characteristic length is indeed the wavelength for correlated dynamics. Since the podosomes are distributed discretely in the cluster, this is the best method of quantifying the wavelength of collective podosome dynamics. In addition, theoretically, we show a linear stability analysis (solution in the form $\sim e^{i\omega t + iqr}$) in the *Models and Methods* section. We obtained the wavelength $\lambda = 2\pi/\text{Re}(q) \sim \sqrt{D_a T}$, which predicts $\sim 3 \mu m$, which is in agreement with our measurement. Overall, we believe that the wavelength has been well quantified from both the experiments and simulations.

R1.2 In the same way, I really do not understand how the STICS measurements could be good measurements for the phase speed. I have many problems with the STICS analysis. From my understanding STICS could be a more accurate version of PIV for an object which would move at a constant speed over several frames (but here watching movie S1 it appears it is far from being the case). To apply these kind of analyses you have to assume that an object (pattern) could be tracked between two consecutive frames, this object or pattern should not differ too much during the acquisition time. This is not what I see on the movies, intensity is varying a lot of course while podosome are appearing/disappearing, so I don't see how a speed can be measured. Moreover, let's say it could still be feasible on certain locations under the cells, then the algorithm should give for example a quantity judging of the quality of the analysis (and in particular of the linear fit that is used to get the speed), so that we could discard the bad quality points and remove noise in the data (that appeared to be huge seeing sup movie 4 where we see plenty of white arrows, and also arrows outside of the cell of the same kind of amplitude than below the cells).

Response: We thank the reviewer for raising these important points. In the following text, we first elaborate on the differences between STICS and PIV, then we explain how the wave speeds are measured, and finally we discuss how STICS is applied to the podosome system.

First, we consider the reviewer's assertion that "STICS could be a more accurate version of PIV for an object which would move at a constant speed over several frames". To clarify the differences between STICS and PIV, it is important to note that PIV obtains "snapshots" of velocities of tracer particles based on single frame pair cross-correlations

between sequential images in the image time series. The PIV flow measurement depends on the presence of multiple tracer particles in the flow field that allow the flow to be measured with single frame pair correlations as a function of time. By contrast, STICS does not depend on the presence of a dense particle field. Instead, STICS measures spatial correlations of the optically resolved fluorescence signals within a region of interest (ROI) for sequential pairs of images separated by a variable time lag. This variable time lag is a discrete frame shift between pairs of image frames. The calculated spatial correlation function for a given time lag is the average of the spatial correlations between the time lag separated image pairs iterated over the total time window (tw) of the STICS measurement. Hence, if there is a spatio-temporal periodicity in the fluorescence signals, STICS can be used to detect it whereas PIV cannot.

Next, although we agree with the reviewer's comment that “intensity is varying a lot of course while podosome are appearing/disappearing”, this varying intensity does not affect wave speed measurement via STICS. This is because STICS calculates the spatio-temporal correlations of fluorescence signals within the chosen ROI and time window, and the velocity is measured by the translation of spatial correlation peaks as a function of the time lag. Therefore, STICS does not need to track the individual particles or objects, which differs from conventional PIV that requires that “this object or pattern should not differ too much during the acquisition time”. Specifically, in the podosome system, STICS does not measure the lateral velocity of a given podosome. Instead, it quantifies the velocity of the positions with maximal F-actin density. This is because the F-actin constantly disassembles at one position while re-assembles at others. This transition velocity of positions with maximal F-actin density is indeed the phase speed of waves, and we have provided a detailed reasoning and explanation with Fig. S8A-S8C in comment **R1.3**.

Finally, regarding the “quality of the analysis”, we want to point out that the STICS method and related image correlation methods have successfully been applied to the podosome system in dendritic cells before, and the propagation of fluorescence intensity for a variety of fluorescently labeled podosome components has been measured (Meddens et al. 2016; van den Dries et al. 2019). In practice, the STICS correlation function calculation has statistical parameters set in the algorithm to accept or reject a fit correlation peak based on its magnitude relative to background correlation. In addition, peak tracking is terminated when the correlation function maximum peak decays below the set threshold (explained in detail in Meddens et al. 2016). Prior to twSTICS analysis, we have also applied an immobile filtering procedure (in the Fourier time frequency domain) to every pixel in the entire image time series to specifically remove the lowest frequency components and effectively filter out the global immobile population. The large arrows outside of the cell have not been counted for wave velocities, as we have selected the cell boundaries as ROIs for postprocessing. The following text has been added to the revised manuscript (see page 16) to clarify the twSTICS method and the measured velocity.

“To quantify the wave speeds in the podosome cluster, we applied a recently developed technique—sliding time window spatiotemporal image correlation spectroscopy (twSTICS). This twSTICS method has been successfully applied to the podosome system in DCs and the propagation of fluorescence intensity in space was measured for a variety of fluorescently labelled podosome components¹². The details of the instrument and the actual settings are

described in the *Models and Methods* section. It is important to note that, by using twSTICS, we are not measuring the lateral velocity of a podosome; instead, we are quantifying the phase velocity of positions with maximal F-actin density. Hence, the measured velocity magnitude can be considered as phase speed $v_c = \omega/\mathcal{R}_e(q)$ for waves with the linear dispersion $\omega \propto \mathcal{R}_e(q)$. We have explained how the phase velocity is extracted using the example of radial waves in *SI Note 6*".

R1.3 Finally, I have a conceptual problem with this approach: F-actin in a single podosome should not be moving (contrary to G-actin that we cannot see). Am I wrong? How these measurements done on F-actin can tell us something on the phase speed of the wave?

Response: The point that F-actin in a single podosome does not move is indeed correct. In our study, we are not using STICS to measure the lateral velocity of a given podosome. Instead, we are quantifying the velocity of positions with a maximal F-actin density, since the F-actin constantly disassembles at one position while re-assembling at others. The position with the maximum F-actin density travels like a "wave". For example, for the radial waves in a podosome cluster (Fig. S8 or Fig. 3), we assume an initial distribution in which only the podosomes at the center of the cluster have a large height. Subsequently, our model predicts the following scenarios for the dynamics of individual podosomes at three time points (Fig. S8):

- T1: Podosome *a* has a larger height (above the steady-state height). Podosome *a* is going to depolymerize leading to a decrease in its height, while podosome *b* and *c* remain close to steady states (Fig. S8A).
- T2: As podosome *a* depolymerizes, the newly-released G-actin diffuse outward, altering the G-actin concentration near podosome *b* and disrupting the previous balance. Podosome *b* starts growing and reaches its maximum height. The wavefront moves outward where podosome *b* is located (Fig. S8B and S8C)
- T3: Podosome *b* begins to depolymerize, sending G-actin further outward to activate the assembly of podosome *c*. When podosome *c* reaches its maximum height, the position of podosome *c* becomes the wavefront (Fig. S8A).

As a result, although the podosomes (*a, b, c*) are not laterally moving, the position with the largest podosome height travels in a wave-like fashion. Therefore, using the STICS method, one can obtain the speed that the wavefront travels (i.e., the distance travelled by the front divided by the time difference). The above discussion on the rationale of wave speed measurement has been added to *SI Note 6*. With this, we hope we have sufficiently explained that these measurements on maximal F-actin intensity do provide the necessary parameters for the phase speed of the wave.

Fig.S8A-S8C. (A) The simulated heights in the podosome cluster using (left panel) a discrete model of podosomes and (right panel) a continuum model showing radial wave patterns. (B) The core height plotted with the distance from the center for three different instances in time. (C) The core height plotted versus time for three representative podosomes marked in (A). Note that curves in panels B and C are extracted from the continuum model, as the discrete and continuum model provides similar results.

R1.4 In any case, in regards to the weak experimental proofs on the accuracy of the model I would change the verb used in the title "orchestrate" with something less affirmative like "is compatible" or "may orchestrate"...

Response: Based on our above response, we believe that the wavelength has been well quantified in the experiments, and STICS is a reliable tool for measuring the wave speed. However, we do feel there can be a more accurate verb in the title; in accordance with the reviewer's suggestion, we have changed the title to "Chemo-mechanical Diffusion Waves Explain Collective Dynamics of Immune Cell Podosomes".

Intermediate comments:

R1.5 For testing the different rigidities, the authors used sylgard 184, while it is fine to use for high rigidity it is well known that it does not behave as an elastic solid at a ratio 1:20 but rather as a viscoelastic material, this is annoying if one wants to test the effect of rigidity only. For this purpose it is better to use DOW CORNING TORAY CY 52-276 A and B or polyacrylamide gels for example. In the same way, the thickness of 10-20 μm may be a little too small if one wants the cell not to feel the hard coverslip beneath the gel.

Response: We agree that sylgard 184 can show viscoelastic behaviour, however, a previous study (Petet et al. 2021) has shown that the storage modulus of sylgard 184 is much higher than its loss modulus, indicating that the viscous contribution is negligible. More importantly, when

subjected to nanoNewton forces and nanometer displacements (which is the case in our study), sylgard 184 has a very short relaxation time of ~ 0.1 s (Leong et al. 2015). This indicates that sylgard 184 relaxes much faster compared to the podosome oscillation period (in several minutes). Thus, we are justified in treating it as an elastic substrate in our study.

With regards to the thickness of the substrates, we believe that a PDMS layer with a 10-20 μm thickness is sufficiently large for a cell not to feel the underlying coverslip. The typical podosome indentation depth in a soft PDMS substrate is measured to be ~ 100 nm (van den Dries et al. 2019) and the 10-20 μm PDMS layer is thus 100-200 times thicker than the podosome indentation depth. In addition, since we consistently detect differences in podosome cluster dynamics between soft and stiff PDMS. Therefore, we believe that our substrate preparation is a valid procedure for studying the effect of substrate stiffness on podosome (cluster) dynamics.

R1.6 Fig 2 B curves are supposed to be representative of the different conditions, and it clearly appears that the amplitude ratio is much smaller in the case of blebbistatin than in control (similar to cyto), and this is absolutely not apparent in Fig 2 C, where the statistics seem rather similar to control (the *** difference is even bothering). What is wrong B or C?

Response: We apologize that the previous curves for the treatments in Fig. 2B are not representative. In the revised manuscript, we have revised Fig. 2B by choosing those podosome data or curves that are more representative for each condition. We conducted the one-way ANOVA with the Benjamini-Hochberg procedure for the statistical analysis in Fig. 2C to obtain the statistical significance between the control and different treatment cases. The reason for the significant difference (***) is the high statistical power of the experiments since more than 400 podosomes were evaluated for each case.

Fig. 2B-2C. (B) Representative fluorescence intensity profiles for the control (Ctrl, blue), cytochalasin D (red), Blebbistatin (yellow), and Y27632 (purple) treatments. (C) The amplitude ratio, τ_a for control and different pharmacological treatments. At least 400 podosomes were measured for each condition. Statistically significant differences are indicated (***) $p < 0.001$, (****) $p < 0.0001$, ANOVA with Benjamini-Hochberg procedure).

R1.7 Fig 4 G Could you explain why on the simulation we observe a clear anticorrelation, which is absent from the experiment?

Response: We thank the reviewer for highlighting this important point. The absence of an anticorrelation observed in the experiments is due to noise, which is typically present in biological systems and was not included in our simulations for cluster dynamics. In our simulation, we obtained the periodic sinusoidal-like podosome oscillation (with period $\sim 2\pi\sqrt{\tau_m\tau_p}$) (Fig.1D). Therefore, the anticorrelation appears when the phase shift between two podosomes becomes larger than π . This phase shift is determined by the time taken for G-actin to diffuse between the podosomes. However, in our experiments, noise from myosin recruitment, polymerization dynamics, or even thermal fluctuation in the microenvironment can make the podosome dynamics less periodic and more stochastic. It can be anticipated that stochasticity in podosome dynamics can potentially remove the anticorrelation between the dynamics of neighbouring podosomes.

To further validate this argument, we applied a Gaussian noise signal to individual podosomes (i.e., χ_p and χ_m in Eqs. 1&2 applied onto the nodes in the continuum model) in our simulations and studied the changes in the correlation coefficient. We found that a higher level of noise could decrease the degree of anti-correlation (Fig. S6A–S6B), which was consistent with the arguments we presented. It is important to note that the noise also slightly decreases the positive correlation; this is because we applied spatially uncorrelated noise to each node, whereas noise may be spatially correlated *in vitro*. However, even in the presence of noise, the characteristic decay distance is consistent with the theory we have presented ($\lambda \sim \sqrt{D_a T} \approx 3 \mu m$). In order to address this point, we have added the following text to the manuscript:

"Note that the small anti-correlation region observed in the simulation (Fig. 4G) is due to the absence of noise in the theory, which leads to periodic (sinusoidal-like) oscillations. This anticorrelation does not appear in the experiments because the noise from myosin recruitment or polymerization can cause less periodic and more stochastic podosome patterns, eventually eliminating the anticorrelation when Gaussian noise is included in podosome dynamics (Fig. S6A-S6B, refer to *SI Note 3*)."

Fig. S6A-S6B. (A) The correlation coefficient plotted with the distance between podosomes for different levels of Gaussian noise. (B) The podosome height (left panel) without noise $R_\sigma = 0$ and (right panel) with noise $R_\sigma = 0.1$.

R1.8 In the model the polymerization speed is taken linear with the force. Is it a reasonable assumption shouldn't the dependency be exponential?

Response: Both the linear and exponential assumptions are reasonable. The linear relation used in our model can be considered as a linearization of the exponential relation between the polymerization speed and force

$$V_p = V_{pm} e^{-\frac{F_p}{F_{sp0}}} \approx V_{pm} \left(1 - \frac{F_p}{F_{sp0}}\right).$$

This linearization has also been used for different force-velocity relations in previous studies (Motahari and Carlsson 2019; Gong et al. 2021; Chan and Odde 2008). Additionally, the linearization allows us to obtain an analytical solution. To further verify that linearization can capture the essential physics, we also tested our simulations for podosome dynamics with the exponential polymerization-force relation. As shown in Fig. R1, we also observed the periodic oscillations in podosomes with similar height and force magnitudes. We have added the related discussion to *SI Note 1* in the revised manuscript.

Fig. R1. (A-C) The simulated (A) core height and protrusive force (red line) are plotted versus time. (C) Phase plot showing simulated myosin force versus core height. The exponential force-speed relation is used, and noise is neglected. The characteristic protrusive force, $F_{p0} = 13 \text{ pN}$, unloading polymerization speed, $V_{p0} = 70 \text{ nm/s}$, and an initial length of the ventral actin filament, $x_0 = 2.4 \mu\text{m}$ were chosen.

Minor comments:

R1.9 Fig 4 B and C, why are there four timecurves (and not two) below each images whereas only two podosomes are encircled?

Response: We apologize for the confusion regarding this figure. We have now revised and marked the time curves in Fig.4B and 4C.

R1.10 line 605 the authors define the amplitude ratio as the ratio of averaged amplitudes... Are they talking here about the standard deviation of the signal?

Response: The amplitude ratio is defined as the ratio between the oscillation amplitudes and the time-averaged value of fluorescence intensity dynamics. The amplitude ratio reflects the fluctuation intensity of a podosome time-curve relative to its mean value and hence has no units. Importantly, the amplitudes extracted in the frequency domain can filter out noise from the signals in the undesired frequency range. In contrast, the standard deviation has the unit same as the data (i.e., fluorescence intensity), and it quantifies the general variation of signals

in the entire frequency range. Therefore, we believe that the amplitude ratio defined here is a better index to indicate the relative oscillation intensity of podosome dynamics.

R1.11 There is no scale bar on Fig 5 and panel D is really small

Response: We apologize for these issues and have added the scale bar to Fig.5 and increased the size of panel D accordingly. .

R1.12 the imaginary number i is missing in equation 8

Response: There is no need to add the imaginary number i , as the square root term yields the imaginary number if we have $(\Gamma - 1 - \tau_m/\tau_p)^2 - 4\tau_m/\tau_p < 0$. On the other hand, whether the eigenvalues have imaginary parts can tell us about the two different podosome growth patterns—oscillatory or monotonic growth (Fig. 1C).

R1.13 line 605 I guess the authors are referring to Fig. S5 and not S4.

Response: We have revised this issue.

Reviewer #2 (Remarks to the Author):

In this paper an elaborate model is developed to describe the observed oscillation of the individual podosome and the wavelike coupling between podosomes in a cluster. The experimental characterization of the dynamics of the podosomes is the focus of the group of one of the authors (Koen van den Dries). In general, the model predictions are found to be in line with experimental observations. The authors claim that they are the first to describe a theoretical model for the spatiotemporal dynamics in clusters of podosomes. This appears to be the case indeed. This paper has a high potential but in its current state some concerns can be expressed.

Response: We sincerely thank the reviewer for the thorough review of our manuscript and excellent comments/suggestions. We would like to highlight the following key improvements made to the revised manuscript in response to the reviewer's comments:

- Addition of a detailed model section in SI Note 1
- Justification of the parameter values in SI Note 3 and an update of Tables S1 and S2
- New simulations to explain the anticorrelation in Fig. 4G
- Correction of figures related to noise and additional discussion on noise effects
- Additional discussion of the effects of lipid recruitment in podosome dynamics
- New simulations for the effects of podosome near-neighbour distance on wave dynamics
- Additional discussion of the frequency extraction using FFT *versus* autocorrelation

Our detailed responses to specific comments are as follows:

R2.1 The emphasis of this work is clearly on the development of the model. I do realize the challenge the authors are facing in finding a proper balance between conveying the message in an accessible way on one hand, and describing the underlying model to a sufficient detail, on the other hand. However, I found it difficult to read this paper as one has to go back and forth between the main text, the end of the main text and the supporting information to fully grasp this nontrivial model. Model parameters are defined in the main text and I had to make a list of symbols and notations for myself.

Table S1 does contain several model parameters but not all of them are presented. The authors could help the reader by at least providing a list of symbols. Furthermore, some relations are within the text without equation number and also these have to be written down to keep track of all definitions and relations. I suggest that the authors provide a text in the SI with a logical sequence of definitions, relations and equations combined with the current notes in the SI. In doing so the model and the predictions by the model will be conveyed in a clear and transparent way. This will be appreciated by the readers even when there are repeats between the main text and the full description of the model in the SI. Furthermore, when the full model is described in detail in the SI the initial and boundary conditions for the various differential equations can be mentioned.

Response: We apologize for the inconvenience caused to the reviewer and are thankful for the suggestion to present the details of the model in the SI. In accordance with the reviewer's

suggestion, we have provided a detailed model development section (SI Note 1) in the revised manuscript, in which we have stated a clear logical sequence of definitions and provided detailed derivations of all equations and relations. The equations are marked with numbers, and a list of symbols can be found in Tables S1 and S2. Additionally, we have included a section (SI Note 2) with all the initial and boundary conditions, as well as the discrete and continuum approaches for solving the equations.

R2.2 Concerning the Table S1: in the “Reference” column there five different types of entries: just a reference number, “adjusted”, “fitting parameter”, “estimated” and “measured”. It is not clear how “adjusted” should be interpreted as this indication appears in combination with a reference number. The diffusion coefficient is a fitting parameter and the reader is referred to SI Note 2. It is not made clear how the value in Table S1 is obtained. Also for other entries the indication “fitting” is used; does this mean fitting a model to experimental data? For the very last entry “measured” is used. However, no reference is given and I assume that the value of this parameter is determined in the current work. However, it is not clear how. I will be appreciated if the authors could be more specific wrt the entries in the last column of Table S1. The listed values are crucial in the prediction power of the model. The authors should give an indication of the uncertainty of the fitted parameters.

Response: We appreciate the reviewer’s suggestion regarding the “Reference” column in Table S1. In the revised manuscript, we have reorganized the entries in the column into two types—“this article, refer to SI Note 3” and reference number. In addition, in the SI Note S3 *Parameter justification for the model*, we have provided a detailed justification of how we selected the values for model parameters.

To illustrate the uncertainty of the fitted parameters, we have plotted the podosome steady-state height and oscillation period as a function of the key parameters (Fig. S3). The analysis of those key parameters can be also found in SI Note 3.

Fig. S3D-S3F. (D-F) The simulated (top panels) core height and (bottom panels) oscillation period plotted for (D) myosin turnover timescale and maximum protrusion force, (E) polymerization speed and Rho-associated feedback, and (F) ventral filament stiffness and effective stiffness for myosin contraction. The non-oscillatory regimes are marked in grey.

R2.3 The details of the actual experiments are rather limited. The reader can assume that the details of the instrument and the actual settings are similar the approach described in the paper by Maddens et al. [ref. 10] where also dendritic cells are investigated and the twSTICS is used. However, this should be explicitly mentioned. The same oversampling in space and time are used as in the Maddens et al paper. So, one can assume that all other details are the same but it would good that this is explicitly stated in the current manuscript, otherwise the specific settings used in the current should be described to sufficient detail (in SI). The abbreviations twSTICS, TOI and ROI should be clarified in the current manuscript. The twSTICS experiments run over a substantial period of time. Are the data fully corrected for motions of the cell body as such by the Fourier filter?

Response: In accordance with the reviewer's suggestion, we have now added the description with references in the revised manuscript (see page 16) as follows:

“To quantify the wave speeds in the podosome cluster, we applied a recently developed technique—sliding time window spatiotemporal image correlation spectroscopy (twSTICS)¹⁰. This twSTICS method has been successfully applied to the podosome system in DCs and the propagation of fluorescence intensity in space for a variety of fluorescently labelled podosome components was measured^{10,12}. The details of the instrument and the actual settings are described in the *Models and Methods* section.”

We have clarified the abbreviations twSTICS, TOI, and ROI in the revised manuscript. The STICS data has been fully corrected for cell body motions, of which the associated method has been described in our previous work (Meddens et al. 2016). We have also clarified this point in the revised *Model and Methods* section of the manuscript (see page 25).

R2.4 In order to be able to repeat the experiments: what are the actual concentrations of blebbistatin, cytochalasin D, and Y27632 used in the experiments? This should also be related to the number of cells and the area of the sample chamber.

Response: We thank the reviewer for pointing this out. In the experiments, we used 20 μM Blebbistatin, 2.5 $\mu\text{g}/\text{mL}$ cytochalasin D, and 20 μM Y27632 for the pharmacological treatments. As reported in our previous works (Meddens et al. 2016; van den Dries et al. 2013), we consistently plated at 1×10^6 DCs per 35 mm dish, and as these cells do not divide, this number does not change during the experiment. We have added the related text to the *Model and Methods* section in the revised manuscript (see page 24).

“For the pharmacological treatments, the following inhibitors were used: cytochalasin D (2.5 $\mu\text{g}/\text{ml}$, Sigma-Aldrich), blebbistatin (20 μM , Sigma-Aldrich), and Y27632 (20 μM , Selleck).”

R2.5 In the model the angle theta between the ventral actin filaments and the core F-actin is assumed to be constant. No justification is given for this assumption. As theta is taken to be constant the high correlation between the length of the ventral actin filament and the ring

diameter (Fig S2, panel A) in the simulations is to be expected. The experimentally obtained high correlation between the actin and vinculin related fluorescence signals (Fig. S2, panel C) can be taken as evidence that the angle theta can be considered to be constant. If the authors agree with this reasoning then this can be explicitly mentioned as experimental evidence that it is justified that theta can be considered to be constant. Otherwise, they have to justify the assumption of a constant value for theta. In Table S1 it is mentioned that the value of theta is estimated in this work. I might be mistaken but where do the authors describe the determination of the actual value of the angle theta? The authors could be more specific about the determination procedure of the value of this angle.

Response: We thank the reviewer for this insightful question and we indeed agree with this reasoning. In our model, we simplified a single podosome as a conical shape with a constant half apex angle of $\theta = \pi/4$. We selected this setting for two reasons: 1) previous experiments (Labernadie et al. 2010) have quantified the podosome core height as 400~900 nm, while the ring radius is measured around 500~900 nm (Labernadie et al. 2014; van den Dries et al. 2019). Hence, we chose the half apex angle to be $\theta = \pi/4$, such that the podosome core height h and ring radius r are equal, i.e., $h = r \cdot \tan(\pi/4)$; 2) as the reviewer has pointed it out, the high correlation between the actin and vinculin fluorescence intensity (Fig. S2, panel C) can be used as evidence for the constant apex angle. We have now added the above discussion and reasoning to SI Note 3.

R2.6 In the main text one can read: “Although the spontaneous oscillations in region I are damped, the random fluctuations (i.e., Gaussian noise contributions $\chi m(t)$ and $\chi p(t)$ in Eqs. 1-2) from myosin recruitment or core-actin polymerization constantly push the podosome away from the steady state; this initiates excursions in the phase space of core height and myosin force, leading to persistent oscillations (Fig. S1A and S1B)”.

How strong must the noise contributions be in order to maintain the oscillations? This question is also to be seen within the context of the following.

In Fig. S1, panel B, myosin force versus core height, $R_sigma = 0.1$ seems to generate the same pattern as for $R_sigma = 0$. However, the myosin force versus core height graph in Fig. 2, panel D, rather refers to $R_sigma = 0.01$ in Fig. S1, panel B. Therefore one expects that $R_sigma = 0.01$ would be an appropriate choice. However, in SI Note 2 it is stated that: “The Gaussian noises are generated by MATLAB randn function $\chi m(t)$, $\chi p(t) \sim N(0, \sigma)$, where their standard deviation are estimated around 1%-10% of the characteristic protrusive force ($\sigma(\chi m) = 0.05Fp_0 \sim 5000$ pN) and initial polymerization speed ($\sigma(\chi p) = 0.05Vp_0 \sim 5$ nm/s), respectively”. This is 5%. This value is mentioned in the figures and in the Table S1 one finds 10%. I am puzzled by this. Please clarify.

Response: We appreciate the reviewer’s question regarding the effects of noise. Gaussian noise (with any non-zero variance) can sustain oscillations for podosomes in the oscillatory regime (Region I in Fig. 1C), as the noise pushes the system from the steady state and initiates the excursions in the phase space (Fig. S1B). Gaussian noise with a larger variance σ only yields larger oscillation amplitudes as shown in Fig. S1B with the same frequency. This is in

contrast to the noise effects in the monotonic/non-oscillatory regime (Region II in Fig.1C), where the podosome system quickly relaxes to a steady state without initiating excursions in the phase space for any level of noise (Fig. S1C).

Fig S1. (B-C) The simulated (top panels) phase space of myosin force and core height plotted for (B) the oscillatory region and (C) the monotonic region. The simulated (middle panels) podosome core height and (bottom panels) myosin force plotted with time for (B) the oscillatory region and (C) the monotonic region. Lines with different colours correspond to the levels of Gaussian noise with different variance ratios, R_σ . The variance ratio was defined as $R_\sigma = \sigma(\chi_p)/V_{p0} = \sigma(\chi_m)/F_{m0}$, where $\sigma(\chi_p)$ and $\sigma(\chi_m)$ denote the variances for actin polymerization and myosin dynamics, respectively. The oscillation period $T \approx 2\pi\sqrt{\tau_m\tau_p}$ is marked in (B). The myosin turnover time was set as $\tau_m = 50$ s to obtain heavily damped oscillations for (B), while it was set as $\tau_m = 500$ s to obtain monotonic growth for (C). All the other parameters can be found in Table S2.

The variance ratio $R_\sigma = \sigma(\chi_p)/V_{p0} = \sigma(\chi_m)/F_{m0}$ for the Gaussian noise applied to myosin dynamics and actin polymerization is 5%. We have corrected the mis-labelling of the curves in Fig. S1B, and the parameter R_σ in Table S1 has also been updated. In Fig. 2D and 2E, the curves were obtained from the simulations without any noise for simplicity and clarity. However, we agree that this can cause confusion, and so we have replotted Fig. 2D and 2E with the effects of noise in the revised manuscript.

R2.7 In the main text it is stated that: “Our model predicts that individual podosome oscillations cause spatial variation in G-actin density, which in turn drives chemomechanical diffusion waves”. What are the required minimal concentration changes in G-actin to stimulate

protrusion growth at a distant point? The changes in G-actin concentration are correlated with the distance between the podosomes in the cluster.

The distance between the podosomes changes with the stiffness of the substrate [Collin O, Tracqui P, Stephanou A, Usson Y, Clément-Lacroix J, Planus E. 3 Spatiotemporal dynamics of actin-rich adhesion microdomains: influence of substrate flexibility. J Cell Sci. 2006 May 1;119(Pt 9):1914-25. doi: 10.1242/jcs.02838. PMID:16636076].

In the current paper experiments are performed on substrates with a different stiffness and the wave propagation has been determined and compared with the predictions of the model. In Table S1 only a single value for the distance between the podosomes in the cluster is considered. Why did the authors not consider a different distance when changing the stiffness of the substrate? In general one can ask the question how the distance between the podosomes influences the velocity of the wave over the cluster. I would be happy to learn from the answer by the authors.

Response: We agree with the reviewer that the distance between the neighbouring podosomes can affect podosome dynamics. To see the effect of the near-neighbour distance on podosome dynamics, we varied this distance d_0 in our discrete model:

Fig S6C-S6E. (A) The oscillation period T plotted as a function of podosome near-neighbour distance d_0 in our discrete model. Blue dots represent the mean oscillation periods extracted from the dynamics of podosomes simulated in the discrete model, whereas the red dashed line is the fitting line. Note that the oscillation period obtained from the continuum model is marked as T_0 . (B-C) The discrete model of the podosome cluster with the near-neighbour distance (left panel) $d_0 = 1125$ nm, (middle panel) $d_0 = 2250$ nm, and (right panel) $d_0 = 4500$ nm for (B) radial waves and (C) random waves.

Our simulations show that both the oscillation periods of individual podosomes (Fig. S6C) and the wave patterns (Fig. S6D-S6E) obtained from our discrete model are in accordance with the results from the continuum model (Fig. 4D and 4F) when the neighbouring distance d_0 is below $3 \mu\text{m}$. Beyond this critical distance, the podosome oscillation periods increase and wave patterns become obscure (Fig. S6). It is important to note that this critical near-neighbour distance is the diffusion length scale $\sqrt{D_a T_0} \approx 3 \mu\text{m}$, where T_0 is the oscillation period of podosomes when their near-neighbour distance is small (discrete model with $d_0 \leq 3 \mu\text{m}$ or continuum model). As the near-neighbour distance exceeds the diffusion length scale, the time

taken for G-actin to diffuse between the neighbouring podosomes becomes larger than the oscillation period T_0 (i.e., $d_0^2/D_a > T_0$). In addition, the oscillation of one podosome can only affect its neighbours through G-actin diffusion only after a long-time delay d_0^2/D_a , which slows down the podosome height growth/shrinking and leads to extended oscillation periods. When $d_0^2/D_a < T_0$, the time delay due to the diffusion is relatively small, and podosome oscillations are not affected. Hence, we can conclude that our continuum model, which yields a spatial average of the podosome heights, can be considered as a generalization for all the discrete cases with near-neighbour distance below the diffusion length scale ($d_0 \leq \sqrt{D_a T_0}$).

We would also like to thank the reviewer for pointing out this important study (Collin et al. 2006); many of the experimental results in this paper provide strong validation of many predictions of our theory, including the oscillation periods (several minutes) and the near-neighbour podosome distance ($\sim 1.5 \mu\text{m}$ on glass). Although a significant difference is observed for substrates with different stiffnesses in Collin's work (Collin et al. 2006), the podosome neighbouring distance is within the diffusion length scale $d_0 \leq \sqrt{D_a T_0} \approx 3 \mu\text{m}$. As previously mentioned, the podosome cluster with the near-neighbour distance below the diffusion length scale ($d_0 \leq \sqrt{D_a T_0}$) can be simulated by the continuum model. Therefore, the wave speeds are not affected by the near-neighbour distance. We agree that this is a valuable point, and we have added the above discussion to SI Note 3 and the following text in the Results section:

“Recent study has shown that the near-neighbour distance between individual podosomes can vary with stiffness²²; however, this difference in the neighbouring distance does not affect the wave speeds, as the podosome near-neighbour distances for substrates with different stiffnesses are below the diffusion length scale $d_0 \leq \sqrt{D_a T} \approx 3 \mu\text{m}$ (Fig. S6C-S6E, refer to *SI Note 3*).”

In addition, with regards to the minimal concentration change in G-actin mentioned by the reviewer, we focused on the steady state of the non-equilibrium system in our study, which is how the podosome cluster (system) responds when adding a perturbation. There is no such required minimal concentration change in G-actin to stimulate the growth of a distant point, as any small perturbation in the G-actin concentration, myosin force, or core height can induce the oscillations of podosomes at a distant position.

R2.8 I wonder what the effect of the lipids are in the dynamics of the wave reaching the various points where the podosomes will undergo their changes. The growth of the podosomes in the cluster must be accompanied by a substantial growth in membrane area in the podosome cluster. Can the rate of lipid recruitment due to the change in lipid area have a modulatory effect on the spatiotemporal characteristics considered in this work? Please comment.

Response: We agree that lipid dynamics could have some modulatory effects on podosome dynamics. We, however, reason that these effects are minimal based on our previous experimental evidence on the indentation depth of podosomes on stiff and soft substrates obtained by electron microscopy (see Supplementary Figure 7 of Van den Dries et al. 2019). On stiff substrates, almost no indentation is observed ($\sim 5\text{-}10\text{nm}$) and on soft substrates, the indentation range is between 30-200 nm. Thus, the assumption that the growth of podosomes

is accompanied by a substantial growth in membrane area in the podosome cluster seems unlikely. Nonetheless, many different lipids (PIP2, PIP3) are indeed known to modulate podosome spatiotemporal dynamics through biochemical interactions with signalling proteins. In the future, this topic deserves further investigation and possibly integration with theoretical models.

R2.9 Fig. 4, panel G: a negative correlation at intermediate distance between the podosomes appears in the simulation that is absent in the experiment. The authors should comment on this.

Response: We thank the reviewer for highlighting this important point for the improvement of our manuscript. The absence of an anticorrelation observed in experiments is due to noise, typically present in biological systems, which we did not include in our simulations for cluster dynamics. In our simulation, we obtained a periodic sinusoidal-like podosome oscillation (with period $\sim 2\pi\sqrt{\tau_m\tau_p}$) (Fig.1D); therefore, the anticorrelation appears when the phase shift between two podosomes becomes larger than π . This phase shift is determined by the time taken for G-actin to diffuse between the podosomes. However, in the experiments, noise from myosin recruitment, polymerization dynamics, or even thermal fluctuation in the microenvironment can make the podosome dynamics less periodic and more stochastic. It can then be anticipated that stochasticity in podosome dynamics can potentially remove the anticorrelation between the dynamics of neighbouring podosomes.

Fig. S6A-S6B. (A) The correlation coefficient plotted with the distance between podosomes for different levels of Gaussian noise. (B) The podosome height (left panel) without noise $R_\sigma = 0$ and (right panel) with noise $R_\sigma = 0.1$.

To further validate this argument, we applied a Gaussian noise signal to individual podosomes (i.e., χ_p and χ_m in Eqs. 1&2 applied onto the nodes in the continuum model) in our simulations and studied the changes in the correlation coefficient. We found that a higher level of noise could decrease the degree of anti-correlation (Fig. S6A-S6B), consistent with the arguments we presented. It should also be noted that the noise also slightly decreases the positive correlation; this is because we applied spatially uncorrelated noise to each node, while noise may be spatially correlated *in vitro*. However, even in the presence of noise, the characteristic decay distance is consistent with the theory we have presented ($\lambda \sim \sqrt{D_a T} \approx 3 \mu\text{m}$). Accordingly, we have added the following text to the manuscript:

"It is important to note that the small anti-correlation region observed in the simulation (Fig. 4G) is due to the absence of noise in the theory, which leads to periodic (sinusoidal-like)

oscillations. This anticorrelation does not appear in the experiments, because the noise from myosin recruitment or polymerization can cause less periodic and more stochastic podosome patterns, eventually eliminating the anticorrelation when Gaussian noise is included in the podosome dynamics (Fig. S6A-S6B, refer to *SI Note 3*).”

R2.10 Autocorrelation of the changes in fluorescence intensity (control situation) can reveal the frequency of the oscillations. This can directly be compared with the predicted value. Is there a reason why the authors did prefer not to do so?

Response: We agree that the autocorrelation of changes in fluorescence intensity can be used to extract the frequency of the oscillations; however, this autocorrelation method does not work well for signals with a short time span and multiple oscillation frequencies. This is because it only captures the overall trend of the signal and can be affected by the external low-frequency noise or perturbation. By contrast, Fast Fourier Transformation (FFT) yields all the individual frequency components of the signal. As most fluorescence signals in experiments have multiple oscillation frequencies, FFT is a better choice to analyse the oscillation pattern.

We have used a specific podosome profile (blue dots and line in Fig. R2A) extracted from our experiments to illustrate this point. Three peaks were captured in the frequency domain after FFT processing (red markers in Fig. R2B), which can be used to recover the podosome dynamics in the time domain (red lines in Fig. R2A). In contrast, the autocorrelation decays monotonically and no information on the oscillations can be readily gained (Fig. R2C). In addition, we should point out that FFT has been widely used for extracting frequencies in dynamics of fluorescence intensity in many previous studies (Liu et al. 2020; Thoke et al. 2017; Collin et al. 2006; Gong et al. 2021). Therefore, we selected the FFT approach to extract the oscillation frequency in this study.

Fig. R2. (A) The fluorescence intensity of a representative podosome plotted with time. (B) The frequency spectrum showing the amplitudes of intensity extracted using the Fast Fourier Transform (FFT) as a function of the frequency. Blue dots and the line represent experimentally measured dynamics, and the red line in the top panel indicates the time-domain dynamics transformed from the three largest peaks in the spectrum (i.e., red markers in the bottom panel) using FFT. (C) Temporal autocorrelation of the podosome dynamics.

R2.11 The combination of Eq. 10: $v_c = \lambda T \approx \sqrt{D_a \omega} / (\omega^2 \omega_p^2 + \omega^2)^{1/4} \cos(\theta/2)$ and the relation $\omega_p = \mu \beta V d / V_{pms} \propto k s$ indicates that the phase velocity decreases with increasing stiffness. This seems to be in contradiction with Fig. 5-C and Fig. 5-F. I am puzzled by this. Please clarify.

Response: This is because of the term $1/\cos(\theta/2)$, where $\theta = \arctan(\omega/\omega_p) - \pi$ defines the augment of the squared wave number, $q^2 = -\frac{i\omega + \omega_p}{D_a}$. A higher substrate stiffness corresponds to a larger ω_p , which decreases ω/ω_p , the augment θ , and the factor $\cos(\theta/2)$, as $\theta/2$ lies in the 4th Quadrant and $\cos(\theta/2)$ is a monotonic increasing function. Hence, the term $1/\cos(\theta/2)$ increases with stiffness, leading to an increasing wave speed with increasing stiffness. However, we realized that the Greek letter θ here can be misleading, as it has already been used for the core-ring angle. In the revised manuscript, we have replaced this augment θ with φ . In addition, we have added Fig. S8D-S8E and related discussion to SI Note 5 to show how the dispersion relation $q(\omega)$, velocity v_c , and wavelength λ vary with ω_p .

Fig. S8D-S8E. (D) (Top panel) The real part $R_e(q)$ and (bottom panel) imaginary part $I_m(q)$ of wave number plotted with different frequency ω with respect to different ω_p . (E) (Top panel) normalized wave speed v_c/V_{pms} plotted with frequency ω . (Bottom panel) normalized wavelength $\lambda/\sqrt{D_a T}$ plotted with normalized frequency ω/ω_p with respect to different ω_p . Note that the curves for normalized wavelength (normalized by diffusion length) versus frequency are overlapping for different ω_p .

R2.12 How does the directionality of the velocity vectors result from the equations? Diffusion is in all directions. Can the authors give a plausible explanation for the observed directionality and indicate how this directionality will result from the coupled differential equations.

Response: When analysing the wavelength and speed using the linear perturbation method, we consider a 1D problem ($\delta h \sim e^{i\omega t + iqx}$) with scalar wave numbers and speeds. For podosome waves in a 2D space ($\delta h \sim e^{i\omega t + i\vec{q} \cdot \vec{r}}$), the velocity measured from twSTICS is the phase speed $v = \omega/|\vec{q}|$, where the direction follows the wave vector direction $\vec{q}/|\vec{q}|$. Therefore, if we could approximate our simulated wave dynamics (i.e., the solution of coupled differential equations) of podosome height as a function $h(\omega t + \vec{q} \cdot \vec{r})$, the directionality of wave velocity comes from the gradient of core height $\vec{\nabla}h$, while its magnitude can be estimated as $(dh/dt)/|\vec{\nabla}h|$. We have added the above discussion to SI Note 5.

R2.13 The podosome cluster radius R_0 is mentioned in Table S1 where it is also indicated that this parameter is measured in experiments. Where is this cluster radius used in the model and how is the value actually determined in the experiments?

Response: In our continuum model, we have used a circular plate to represent the podosome cluster, and the cluster radius R_0 is the radius of the circular plate (Fig. 3B). In the experiments, we approximate the cluster radius to be the cell radius as the podosomes are all over the basal surface of the cell, which can be measured directly from the experimental images (Fig. 2A and 3C). In addition, previous experiments have quantified the distance between individual podosomes as $d_0 \approx 1.5 \mu m$ for cells cultured on glass (Collin et al. 2006). Considering $N = 100$ podosomes in a cluster, we can estimate the podosome cluster size to be around $R \sim d_0 \sqrt{N} \approx 15 \mu m$. To clarify this point, we have added the above discussion to SI Note 3 in our revised manuscript.

R2.14 In the discrete model a regular pattern of podosomes is assumed. I do understand that this is within a model but how close to the reality is this approximation as, in general, podosomes arrange in different patterns?

Response: We agree that the podosome clusters can have irregular patterns, such as rosette patterns. Although our discrete model assumes a regular pattern with a fixed near-neighbour distance for the podosome cluster, we have further generalized the discrete model to the continuum model. In principle, one can select random points in the circular plate (Fig. 3B) as individual podosome positions for the discrete model; the same wave pattern will be obtained as long as the podosomes have a near-neighbour distance below the diffusion length scale ($d_0 \leq \sqrt{D_a T}$), which can be seen in our extra simulations (Fig. S6C-S6D). However, we do agree that the cases with complex or irregular cluster patterns with large podosome neighboring distance ($d_0 > \sqrt{D_a T}$) can exist, which is worth considering separately using our discrete model in a future study. To address this important point, we have added the following sentences to the discussion section of our revised manuscript:

“Although a regular pattern of podosomes in the cluster is assumed in our discrete model, our continuum model has generalized all the discrete cases with a near-neighbour distance below the diffusion length scale ($d_0 \leq \sqrt{D_a T}$). More complex or irregular cluster patterns, such as rosette patterns with large podosome near-neighbour distances ($d_0 > \sqrt{D_a T}$), can be easily adapted in our discrete model and discussed separately in a future study.”

Minor comments:

R2.15 In the main text one can read: “Based on the phase diagram (Fig. 1C), we found that when we inhibit actin polymerization, core-actin growth becomes much slower compared to myosin dynamics, i.e., $\tau_p/\tau_m \ll 1$, causing non-oscillatory podosome behaviors (Fig. 1C, red arrow). When the myosin contractility is inhibited, myosin turnover becomes very slow such that $\tau_m/\tau_p \gg 1$, leading to non-oscillatory growth in podosomes (Fig. 1C, yellow arrow)”. The marked inequalities express the same conditions wrt the characteristic times. Please correct.

Response: Thank you for pointing this out. We have corrected this issue in line with your suggestion.

R2.16 Fig S2. Oscillatory behaviours in podosome core and ring. The labels of the panels are not correctly indicated in the legend to the figure. Please correct.

Response: We apologize for this omission. We have corrected the labels in the figure.

R2.17 In the main text one can read: “The largest four peaks in the spectrum were chosen to calculate the amplitudes and frequencies of the podosome dynamics (Fig. S4)”. This should be Fig. S5.4. Table S1: FM0 should be Fm0.

Response: Thank you for highlighting this issue. We have corrected both of these errors in the revised manuscript.

R2.18 Fig. S1, panel B: the orange and yellow colour are used to denote the same trace. Please adjust the colours.

Response: Thank you for this suggestion, we have adjusted the colours accordingly.

References:

- Chan, Clarence E., and David J. Odde. 2008. 'Traction dynamics of filopodia on compliant substrates', *Science*, 322: 1687-91.
- Collin, Olivier, Philippe Tracqui, Angélique Stephanou, Yves Usson, Jocelyne Clément-Lacroix, and Emmanuelle Planus. 2006. 'Spatiotemporal dynamics of actin-rich adhesion microdomains: influence of substrate flexibility', *Journal of Cell Science*, 119: 1914-25.
- Derenyi, I., F. Julicher, and J. Prost. 2002. 'Formation and interaction of membrane tubes', *Physical Review Letters*, 88: 238101.
- Fäßler, Florian, Georgi Dimchev, Victor-Valentin Hodirnau, William Wan, and Florian K. M. Schur. 2020. 'Cryo-electron tomography structure of Arp2/3 complex in cells reveals new insights into the branch junction', *Nature Communications*, 11.
- Gong, Ze, Katrina M Wisdom, Eóin McEvoy, Julie Chang, Kolade Adebawale, Christopher C Price, Ovijit Chaudhuri, and Vivek B Shenoy. 2021. 'Recursive Feedback between Matrix Dissipation and Chemo-Mechanical Signaling Drives Oscillatory Growth of Cancer Cell Invadopodia', *Cell Reports*, 35: 109047.

- Keren, K., Z. Pincus, G. M. Allen, E. L. Barnhart, G. Marriott, A. Mogilner, and J. A. Theriot. 2008. 'Mechanism of shape determination in motile cells', *Nature*, 453: 475-80.
- Labernadie, A., A. Bouissou, P. Delobelle, S. Balor, R. Voituriez, A. Proag, I. Fourquaux, C. Thibault, C. Vieu, R. Poincloux, G. M. Charriere, and I. Maridonneau-Parini. 2014. 'Protrusion force microscopy reveals oscillatory force generation and mechanosensing activity of human macrophage podosomes', *Nature Communications*, 5: 5343.
- Labernadie, Anna, Christophe Thibault, Christophe Vieu, Isabelle Maridonneau-Parini, and Guillaume M Charrière. 2010. 'Dynamics of podosome stiffness revealed by atomic force microscopy', *Proceedings of the National Academy of Sciences*, 107: 21016-21.
- Leong, Man Chun, Mui Hoon Nai, Fook Chiong Cheong, and Chwee Teck Lim. 2015. 'Viscoelastic effects of silicone gels at the micro-and nanoscale', *Procedia IUTAM*, 12: 20-30.
- Liu, Xili, Seungeun Oh, Leonid Peshkin, and Marc W. Kirschner. 2020. 'Computationally enhanced quantitative phase microscopy reveals autonomous oscillations in mammalian cell growth', *Proceedings of the National Academy of Sciences*, 117: 27388-99.
- Meddens, M. B., E. Pandzic, J. A. Slotman, D. Guillet, B. Joosten, S. Mennens, L. M. Paardekooper, A. B. Houtsmuller, K. van den Dries, P. W. Wiseman, and A. Cambi. 2016. 'Actomyosin-dependent dynamic spatial patterns of cytoskeletal components drive mesoscale podosome organization', *Nature Communications*, 7: 13127.
- Motahari, F., and A. E. Carlsson. 2019. 'Thermodynamically consistent treatment of the growth of a biopolymer in the presence of a smooth obstacle interaction potential', *Physical Review E*, 100.
- Mullins, R Dyche, John A Heuser, and Thomas D Pollard. 1998. 'The interaction of Arp2/3 complex with actin: nucleation, high affinity pointed end capping, and formation of branching networks of filaments', *Proceedings of the National Academy of Sciences*, 95: 6181-86.
- Petet, Thomas J., Halston E. Deal, Hanhsen S. Zhao, Amanda Y. He, Christina Tang, and Christopher A. Lemmon. 2021. 'Rheological characterization of poly-dimethyl siloxane formulations with tunable viscoelastic properties', *RSC Advances*, 11: 35910-17.
- Shi, Zheng, Zachary T. Graber, Tobias Baumgart, Howard A. Stone, and Adam E. Cohen. 2018. 'Cell Membranes Resist Flow', *Cell*, 175: 1769-79.e13.
- Thoke, Henrik S., Sigmundur Thorsteinsson, Roberto P. Stock, Luis A. Bagatolli, and Lars F. Olsen. 2017. 'The dynamics of intracellular water constrains glycolytic oscillations in *Saccharomyces cerevisiae*', *Scientific Reports*, 7.
- van den Dries, K., MBM Meddens, S De Keijzer, Shashank Shekhar, Vinod Subramaniam, Carl G Figdor, and A Cambi. 2013. 'Interplay between myosin IIA-mediated contractility and actin network integrity orchestrates podosome composition and oscillations', *Nature Communications*, 4: 1412.
- van den Dries, K., L. Nahidiazar, J. A. Slotman, M. B. M. Meddens, E. Pandzic, B. Joosten, M. Ansems, J. Schouwstra, A. Meijer, R. Steen, M. Wijers, J. Fransen, A. B. Houtsmuller, P. W. Wiseman, K. Jalink, and A. Cambi. 2019. 'Modular actin nano-architecture enables podosome protrusion and mechanosensing', *Nature Communications*, 10: 5171.

REVIEWER COMMENTS

Reviewer #1 (Remarks to the Author):

I want to thank the authors for the real efforts they made to try to convince me that the methodology used is valid to measure the wave speed. I have seen that one of the main author of the original technique STICS has now been included in the authors' list.

First, I want to say that I really like STICs and all its derivatives, I have downloaded a while ago the original packages developed by the main authors and used them for my teaching as I really think it helps understanding time-space cross correlation.

I have been doing image correlation for years now (PIV, RICS, optical flow...), and one thing I learned is that there is no point trying to measure something on an image or a pair of images that you cannot see with your eyes (of course the algorithms can give you subpixel accuracy that you cannot really get by eyes, but it cannot detect movement when your eyes can not). The real danger with these sophisticated techniques, is that whatever the conditions, it will give you an arrow field, representative... or not at all of the underlying movement.

I have spent quite a long time running the movies provided by the authors zooming on them, and honestly, for me it would be very very difficult to draw a wave velocity field by hand (which could constitute a ground truth for the analysis). I must admit that on some parts of the image I could draw some arrows, especially on the borders of the cell, but in many locations, I would certainly not be able to do so, as there is so many fusions dissociations... Maybe, in their previous paper, as there was radial waves it was a little easier to detect, but here...

The authors state now very clearly that you cannot detect podosome movements but that the algorithm correlate high intensity F actin area through time. But as the analysis is performed on a regular grid and with a very small correlation window, the algorithm is also trying to match in between two images areas with not so high F-actin intensity.

Again all these algorithms have been developed to measure real pattern/object speed and not fluctuating intensity patterns maybe this would be ok if there was just one or a few intense patches moving but it is not the case here, where you have many of them moving at the same time in many different directions, and there is no way of stating: this high intensity signal has moved on the left and not on the right...

As a conclusion, I would only believe in the experimental measurement of wave speed in this system, if the authors would be able to draw some few small ground truth by hands (representative of the different movements) and check that the algorithm is retrieving the result (as it is currently done to test deep learning algorithms.) Otherwise, I do not think the measurements are sufficiently robust to be quantitatively compared to the model.

Minor comments:

Anticorrelation part:

The author claim that the absence of anticorrelation in the data is due to noise. I appreciate the additional simulation they did including some Gaussian noise. However, the results of their analysis is worrying me a bit as it is also decreasing (significantly, and not slightly) the positive correlation.

Viscoelasticity of the PDMS

The authors are citing some articles (but I was not able to consult them as a name and a year are not enough to retrieve the papers)

Imaginary number.

I always learned that the square root function was defined on $[0, +\infty[$ and should not be written for a negative number (I agree it is purely a question of convention, but..) (equation 8)

Reviewer #2 (Remarks to the Author):

I appreciate the efforts by the authors in response to my questions and comments. I am satisfied by the detailed answers, the changes within the manuscript and the adapted sections in the supporting information. The manuscript can be recommended for publication.

I suggest the authors check the supporting information for some typing errors, such as:

- near-neighbour: (near)-neighbour
- behavior: behaviour

RESPONSE TO REVIEWERS

We thank the reviewers for another careful review of our manuscript. We are pleased that both reviewers positively appraised our efforts in addressing the previous comments.

Here, we address the remaining comments raised by Reviewer #1 on the STICS analysis. As detailed below, we have re-processed our data using an improved STICS approach (for which we added R. Migueles Ramirez to the author list) and performed additional simulations to validate our theoretical model independent of the STICS method as requested by Referee 2. Given the significant additions following the reviewer suggestions, we trust that our manuscript is now acceptable for publication in *Nature Communications*.

(Reviewer comments in blue, our responses in plain text and edits in the revised manuscript are highlighted)

Reviewer #1 (Remarks to the Author):

I want to thank the authors for the real efforts they made to try to convince me that the methodology used is valid to measure the wave speed. I have seen that one of the main author of the original technique STICS has now been included in the authors' list.

Main comment/problem:

R1.1 First, I want to say that I really like STICs and all its derivatives, I have downloaded a while ago the original packages developed by the main authors and used them for my teaching as I really think it helps understanding time-space cross correlation. I have been doing image correlation for years now (PIV, RICS, optical flow...), and one thing I learned is that there is no point trying to measure something on an image or a pair of images that you cannot see with your eyes (of course the algorithms can give you subpixel accuracy that you cannot really get by eyes, but it cannot detect movement when your eyes can not). The real danger with these sophisticated techniques, is that whatever the conditions, it will give you an arrow field, representative... or not at all of the underlying movement.

I have spent quite a long time running the movies provided by the authors zooming on them, and honestly, for me it would be very very difficult to draw a wave velocity field by hand (which could constitute a ground truth for the analysis). I must admit that on some parts of the image I could draw some arrows, especially on the borders of the cell, but in many locations, I would certainly not be able to do so, as there is so many fusions dissociations... Maybe, in their previous paper, as there was radial waves it was a little easier to detect, but here...

The authors state now very clearly that you cannot detect podosome movements but that the algorithm correlate high intensity F actin area through time. But as the analysis is performed on a regular grid and with a very small correlation window, the algorithm is also trying to match in between two images areas with not so high F-actin intensity.

Again all these algorithms have been developed to measure real pattern/object speed and not fluctuating intensity patterns maybe this would be ok if there was just one or a few intense

patches moving but it is not the case here, where you have many of them moving at the same time in many different directions, and there is no way of stating: this high intensity signal has moved on the left and not on the right....

As a conclusion, I would only believe in the experimental measurement of wave speed in this system, if the authors would be able to draw some few small ground truth by hands (representative of the different movements) and check that the algorithm is retrieving the result (as it is currently done to test deep learning algorithms.) Otherwise, I do not think the measurements are sufficiently robust to be quantitatively compared to the model.

Response: We would like to thank the reviewer for the careful review of our previous response and the questions on the STICS method, which helped us to better assess the merits and limitations of the STICS method and improve the manuscript. To address the reviewer's comments, we have now included:

- 1) a visual summary of how the STICS output relates to the fluorescence intensity fluctuations in the movies
- 2) a STICS analysis of the output of our theoretical model to cross-validate the predictions of STICS analysis of the experimental data.
- 3) a discussion on the capability of STICS in characterizing wave dynamics.
- 4) additional analysis to extract the wave speeds using the expression $v_c = \lambda/T$ to validate the predictions of our model independently of the STICS method.

1) *Visual summary of the STICS output and Lifeact-GFP movies*

First, we would like to apologize about the visual representation of the F-actin flows in the original Fig. 5D and Supplementary Movie S4. After careful reassessment, we indeed agree with the reviewer that the F-actin flows were hard to detect visually, something we should have realized earlier. To better appreciate the presence of waves, we have now improved our STICS analysis with improved algorithms with enhanced vector quality control. Briefly, we have removed vectors outside of the cluster and increased the colour bar range to eliminate the white vectors (Revised Fig. 5D); We also applied quality control tests to filter out spurious vectors and to ensure that the generated vectors are not due to noise (refer to *Methods* section in Page 25 of manuscript). While briefly summarized in the current manuscript, a more detailed description of the improved algorithms and a newly developed graphical user interface will be submitted as a separate manuscript for the benefit of the community. After applying these new algorithms to our data, we now believe that the flows (or patterns) of wave propagation can be detected easily, resulting in an improved version of Fig. 5D (see below). To further demonstrate that STICS is able to pick up fluorescence intensity waves, we selected representative regions in which the calculated flow vectors clearly correlate with fluorescence intensity that propagates from one podosome to its neighbouring podosomes (New Fig. S8C). For example, in Example Flow 1, when the arrows are pointing towards left, podosome #2 becomes brighter while podosome #1 becomes less bright; in Example Flow 2, when the STICS arrows are pointing towards the centre, podosome #1 in the centre position appears.

Fig. 5D. (D) Representative dendritic cell (DC) before (top panels) and after (bottom panels) adding cytochalasin D. DCs are transfected with LifeAct-RFP using confocal microscopy with 15 s frame intervals. Time series subjected to twSTICS analysis are plotted as vector maps. The arrows indicate flow directions, and both the size and colour denote flow magnitudes. Note that the arrows with white colour indicate the magnitudes exceed $0.5 \mu\text{m}/\text{min}$. Scale bar, $10 \mu\text{m}$; Insets scale bar, $2 \mu\text{m}$.

Fig. S8C. (C) Four representative examples of F-actin intensity flow regions with measured STICS vectors for Lifeact-RFP transfected dendritic cells. Top panels in each example: time series showing the dynamics of individual podosomes in the cluster, where the podosomes of interest are numbered. Bottom panels show STICS output vector maps for these regions in time. The arrows indicate flow directions, and both the size and colour denote flow magnitudes. Example Flow 1: when the arrows are pointing towards left, podosome #2 becomes brighter while podosome #1 becomes less bright. Example Flow 2: when the STICS arrows are pointing towards the centre, podosome #1 in the centre position appears. Example Flow 3: when the STICS arrows are pointing left upward, podosome #2 becomes brighter while podosome #1 becomes less bright. Example Flow 4: when the STICS arrows are pointing right downward, podosome #2 becomes brighter while podosome #1 becomes less bright. Scale bar, $1 \mu\text{m}$.

2) STICS analysis on the output of the theoretical model

We also performed STICS analysis on the data from our simulations based on the theoretical model (Eqs. 5-7) to show capability of STICS in characterizing wave dynamics. Our analysis shows that the STICS vectors can characterize the transition of the positions with maximal podosome heights despite the fast fusion and disassociation of podosomes (Movie S5). For example, as the STICS arrows in the region circled with white dashed line in Fig. S9 are pointing left downwards, the positions with large podosome heights (i.e. the wave fronts) propagates in the same direction. Interestingly, by plotting the histogram of velocity magnitudes measured by STICS (New Fig. S9B, see below), we found the magnitudes of flow velocities are around $0.1 \mu\text{m}/\text{min}$ which is the same magnitude with our theoretically-predicted wave speed $\frac{\lambda}{T} \sim \frac{3 \mu\text{m}}{7 \text{min}}$. The STICS analysis on experiments (New Fig. S8C) and simulations (New Fig. S9) shows that measured velocity vectors can characterize the dynamics of fluorescence intensity waves that are qualitatively observed.

Fig. S9 (A) The simulated heights for a representative podosome cluster. Scale bar, $5 \mu\text{m}$. (B) The occurrence probability of flow velocity magnitudes is measured by STICS. (C) Time series of the indicated yellow rectangular area in panel A illustrating wave-like dynamics in the podosome cluster.

Time series subjected to twSTICS analysis are plotted as vector maps. The arrows indicate flow directions, and both the size and colour denote flow magnitudes. The regions circled with dashed lines indicate the evolution(flow) of the positions with large podosome heights (i.e., the propagation of the wave fronts). Scale bar, $5 \mu\text{m}$.

3) Discussion on the capability of STICS to pick up waves

We agree with the reviewer that our STICS analysis is “performed on a regular grid”, which can introduce a bias in the STICS measurement of wave speeds. Applying the improved STICS analysis for the control and treatment cases, we have used ROIs with 32×32 pixels ($4.48 \mu\text{m} \times 4.48 \mu\text{m}$) and shifted adjacent ROIs four pixels ($0.56 \mu\text{m}$) in the horizontal and vertical directions, such that we analysed each podosome in the cluster while applying a large enough

ROI size to cover single and nearest neighbour podosomes. We agree that this method has sampling limitations in characterizing wave speeds due to the high mobility and randomness of podosomes. Yet, even with this limitation, STICS analysis is able to provide an estimate of the quantity (flow velocity of F-actin fluctuations) that characterizes the wave dynamics (new Fig. S8&S9 shown above), which can then be used to validate our model predictions. For consistency throughout the manuscript, we have now removed the word “quantitatively”, replaced “extract” with “characterize”, and changed “wave speed” measured by STICS to “flow velocity”. Additionally, the following text has also been added to the revised manuscript (page 17):

“Although the velocity vectors measured by STICS may not be precisely the phase velocities of waves, the measured magnitudes of flow can be used to qualitatively characterize the speed of wave propagation (refer to *SI Note 6*; Fig. S8, S9, and S10).”

Fig. S8A-S8B. (A) The correlation coefficient plotted with the distance between podosomes for a representative dendritic cell before (Blue line and dots) and after cytochalasin D treatment (Red line and dots). Solid lines represent the mean and shade areas represent 95% confidence intervals. Inset: Data was fitted by the exponential decay function ($y = y_0 + e^{x/\lambda_0}$; dashed lines) for a dendritic cell before and after cytochalasin D treatment. (B) The propagation speeds of waves λ/T for control (Ctrl, blue), cytochalasin D (CytoD, red), blebbistatin (Bleb, yellow), and Y27632 (Ytreat, purple) treatments. Statistically significant differences are indicated (** $p < 0.01$, ANOVA with Benjamini-Hochberg procedure).

4) Validation of the model through calculating wave speed using $v_c = \lambda/T$

To validate our model predictions for wave speeds independent of the STICS analysis, we have now also estimated the wave speeds by calculating the ratios between wavelength λ and oscillation period T , that is $v_c = \lambda/T$. We obtained the wavelength λ by fitting the correlation coefficient plot using an exponential decay function as shown in Fig. S8A. For example, we extracted and plotted the correlation coefficient of fluorescence dynamics of two podosomes as a function of their distance for a representative cell before and after cytochalasin D treatment (Fig. S8A), and then we fit the curves using an exponential decay function to obtain the characteristic decay length λ (New Fig. S8B inset). The oscillation period T can be obtained by using the FFT method (refer to the *Methods* section, page 26). Similar to our STICS results (Fig. 5D), the new analysis also shows that different pharmacological treatments decrease the propagation speeds of waves (New Fig. S8B). This reduced wave speeds after pharmacological

treatments further supports the validity of our model. The above discussion has been added in the revised manuscript (see Page 17) and highlighted in yellow:

In addition to characterizing speeds of wave dynamics by twSTICS, we can estimate the wave speed using the relation $v_c = \lambda/T$, where wavelength λ can be obtained by fitting the correlation coefficient curves and period T is extracted from the individual podosome oscillations (refer to *SI Note 6*; Fig. S8A-S8B). In agreement with our theoretical predictions (Fig. 5C), wave speed is reduced in treated cells as quantified vector magnitudes by STICS (Fig. 5E) and podosome intensity oscillation cross-correlation (Fig. S8B).

Overall, we agree with the reviewer that there are limitations in using STICS to extract the wave speeds accurately, but we hope that the newly added visualizations and analyses provide sufficient evidence that the STICS analysis can be used to characterize wave dynamics and provide the necessary validation of our theoretical predictions. Importantly, we would like to emphasize that the central novelty and innovative aspect of this work is the development of a new theoretical model which explains for the first time how wave-like dynamics emerges from the synergistic coupling between actin polymerization and diffusion. All the experimental data, including the impact of pharmacological treatments, presented in the paper are qualitatively consistent with the predictions of our theory.

Minor comments:

R1.2 The anticorrelation in the theoretical model:

The author claim that the absence of anticorrelation in the data is due to noise. I appreciate the additional simulation they did including some Gaussian noise. However, the results of their analysis is worrying me a bit as it is also decreasing (significantly, and not slightly) the positive correlation.

Response: We thank the reviewer for raising the question on the effect of noise on correlations in podosome dynamics, which led us to think more deeply about the role of noise in wave dynamics. In our previous simulations, at each point we apply Gaussian noise that is spatially uncorrelated (or independent) with the level of noise at other points, which leads to a decrease in positive correlation. However, in experiments, it is reasonable to expect that noise at different positions should be strongly correlated at short distances and becomes uncorrelated over large distances (Fig. S6A). To generate this spatially-correlated noise, we applied a k -dimensional random vector $\mathbf{X} = (X_1, \dots, X_k)$ with a multivariate Gaussian distribution $\mathbf{X} \sim \mathcal{N}_k(\boldsymbol{\mu}, \boldsymbol{\Sigma})$, where its covariance matrix $\Sigma_{i,j} = \text{Cov}[X_i, X_j]$ specifies the spatial correlation (refer to SI Note 3). When incorporating this spatially-correlated noise into our continuum model, we found that it eliminates anti-correlation and elevates the positive correlation in the correlation coefficient plot (Fig. S6B bottom panel). This is different from the previous case of spatially-uncorrelated noise where the positive correlation coefficients decrease (Fig. S6B top panel). Clearly, the spatially-correlated noise is more reasonable in our podosome system, as the correlation coefficient plot (Fig. S6B bottom panel) more closely resembles the plot from experiments (Fig. 4G). In addition, previous studies have also used the spatially-correlated noise in many biological systems, including DNA charge transfer (Liu, Beratan et al., J. Phys.

Chem. B, 2016) and neuronal firing (Wang, Liu et al., Phys. Rev. E, 2004; Lindner, Doiron et al., Phys. Rev. E, 2005). The above discussion has been added to our revised manuscript (Page 15) and SI Note 3.

Fig S6A-S6C. (A) (Top panel) Independent Gaussian noise and (bottom panel) spatially-correlated noise depicted in the plane of the podosome cluster. (B) The correlation coefficient plotted with the distance between podosomes for (top panel) independent Gaussian noise and (bottom panel) spatially-correlated noise. Lines with different colours correspond to different levels of noise, and the variance ratio was defined as $R_\sigma = \sigma/F_{m0}$, where σ denotes the variances for the Gaussian noise of myosin force. (C) The podosome height (left panel) without noise and (right panel) with noise for (top panel) the independent Gaussian noise and (bottom panel) spatially-correlated noise cases.

Reference:

- Lindner, B., B. Doiron and A. Longtin (2005). "Theory of oscillatory firing induced by spatially correlated noise and delayed inhibitory feedback." *Physical Review E* 72(6): 061919.
- Liu, C., D. N. Beratan and P. Zhang (2016). "Coarse-grained theory of biological charge transfer with spatially and temporally correlated noise." *The Journal of Physical Chemistry B* 120(15): 3624-3633.
- Wang, S., F. Liu, W. Wang and Y. Yu (2004). "Impact of spatially correlated noise on neuronal firing." *Physical Review E* 69(1): 011909.

R1.3 Viscoelasticity of PDMS

The authors are citing some articles (but I was not able to consult them as a name and a year are not enough to retrieve the papers)

Response: We apologize that the information we provided was not sufficient to find the papers. The two articles regarding the viscoelasticity of PDMS in our previous response letter are:

Leong, M. C., M. H. Nai, F. C. Cheong and C. T. Lim (2015). "Viscoelastic effects of silicone gels at the micro-and nanoscale." *Procedia IUTAM* 12: 20-30.

Petet, T. J., H. E. Deal, H. S. Zhao, A. Y. He, C. Tang and C. A. Lemmon (2021). "Rheological characterization of poly-dimethyl siloxane formulations with tunable viscoelastic properties." *RSC Advances* 11(57): 35910-35917.

We have also attached these two articles as a supplementary data file, and the main conclusions on the viscoelasticity of gels that we referred to in our previous response have been highlighted.

R1.4 Imaginary number.

I always learned that the square root function was defined on $[0, +\infty[$ and should not be written for a negative number (I agree it is purely a question of convention, but..) (equation 8)

Response: Thank you for highlighting this point, and we apologize that the reason of not adding the imaginary number i has not been state clearly in our last response letter. The main reason is that $(\Gamma - 1 - \tau_m/\tau_p)^2 - 4\tau_m/\tau_p$ in Eq.8 can also be positive. If $(\Gamma - 1 - \tau_m/\tau_p)^2 - 4\tau_m/\tau_p < 0$ is true for all conditions, we would also tend to add the imaginary number i in front of the square root for clarity. However, $(\Gamma - 1 - \tau_m/\tau_p)^2 - 4\tau_m/\tau_p$ can be positive depending on the parameter choices in our model. The positive or negative values correspond to different physical phenomena: if it is negative, the eigenvalues $\omega_{1,2}$ have imaginary parts (i.e., $\mathcal{I}_m(\omega_{1,2}) \neq 0$), leading to the oscillatory behaviours in simulated podosome dynamics; By contrast, the positive values yield real eigenvalues $\omega_{1,2}$ (without imaginary parts, i.e., $\mathcal{I}_m(\omega_{1,2}) = 0$) and hence the monotonic behaviours in simulated podosome dynamics. This is how we plot the phase diagram in Fig. 1C. Besides, we agree that it is a matter of convention, and the square root function of negative values is well-defined in the complex number system. It is also worth noting that this convention/notion has been widely used in stability analysis, including Steven Strogatz's classical textbook of Nonlinear Dynamics and Chaos (Strogatz, Steven H, 2018) and previous studies (Dierkes, Sumi et al. PRL, 2014, Banerjee, Utuje et al. PRL, 2015).

Reference:

Strogatz, Steven H. *Nonlinear dynamics and chaos: with applications to physics, biology, chemistry, and engineering*. CRC press, 2018.

Banerjee, S., K. J. Utuje and M. C. Marchetti. *Propagating Stress Waves During Epithelial Expansion*. Phys Rev Lett 114(22): 228101, 2015

Dierkes, K., A. Sumi, J. Solon and G. Salbreux. *Spontaneous oscillations of elastic contractile materials with turnover*. Phys Rev Lett 113(14): 148102, 2014

Reviewer #2 (Remarks to the Author):

I appreciate the efforts by the authors in response to my questions and comments. I am satisfied by the detailed answers, the changes within the manuscript and the adapted sections in the supporting information. The manuscript can be recommended for publication.

I suggest the authors check the supporting information for some typing errors, such as:

- near-neighbour: (near)-neighbour
- behavior: behaviour

Response: We thank the reviewer for the review of our manuscript, and we are very grateful for his/her recommendation for publication. We have also carefully proofread the supporting information and corrected the typing errors.

REVIEWER COMMENTS

Reviewer #1 (Remarks to the author) :

First I really would like to thank the authors for their careful work to try to answer my questions.

The authors really pushed me to search for a mathematical foundation to my doubts on the real capacity of STICS to give access to the wave velocity.

From the new data and analysis presented, two things really perturbed me :

- The fact that the velocity field was varying on a very short length scale (can we really think in terms of waves in this case?) and that a lot of arrows have now been discarded (why is the algorithm so often failing to recover a displacement?)
- The fact that they did not recover analyzing the simulating data the correct wave velocity (0.1 μm versus 0.43 μm)

I got an idea of what could be improved reading a paper by Heitz et al. (that I am now joining).

To measure the velocity of the actin field (\mathbf{w}) using the image intensity, one would have to solve the following equation:

$$\partial_t I + \mathbf{w} \cdot \nabla I + I \operatorname{div} \mathbf{w} = 0. \quad (2)$$

Where the $\operatorname{div}(\mathbf{w})$ would account for out of plane intensity gain or loss.

Which reduces to

$$\partial_t I + \mathbf{w} \cdot \nabla I = 0. \quad (6)$$

when we do not consider this variation (which can be a problem).

The paper explains pretty well the pitfalls of the correlation algorithm when used to solve the problem (paragraph 2.2 highlighted in yellow in the paper).

Hence, maybe the window size used here was too small to allow the following hypothesis to be right.

$$\forall d, \sum_{\mathbf{r} \in \mathcal{W}(x)} I_2(\mathbf{r} + d)^2 \approx \text{constant}. \quad (14)$$

To further test that, I could not help trying to use a KLT algorithm on the authors' movies which will solve (eq6) with not needing the hypothesis (14).

I am joining the results on the experimental movie and on the simulations I used a window of 64x64 pixels.

Maybe this is not completely correct, as I did not use the raw data, and was not sure about the scale of the simulation (from my understanding the disk we see is roughly $20\mu\text{m}$ in radius which on the image I used make a pixel size of $0.1\mu\text{m}/\text{pixel}$, and I took a 10 second difference in between two frames), this leads to the following histogram.

I did not use the beginning of the movie, as I saw that there seems to be a transient regime, and that the steady state takes time to be reached (is it true? Normal?)

The average value obtained is 0.46 and the median is 0.37.

I am wondering what value should be taken for comparing with the theoretical one...

In conclusion, the algorithm used and certainly the **window size** may influence greatly the results.

If the authors do not want to measure a podosome structure movement but rather a global fluctuation of the intensity, then the window should be relatively large to average on many of such structures, but the problem is that it cannot be too large compared to the wavelength, which is predicted, in final to be rather small...

But as the authors are now saying that their flow field measurement give mainly a 'qualitative' idea of the wavelength, I guess this may not be that important.

However, if the authors are interested in testing my KLT codes, I will be happy to provide them for this study or for future ones if they judge it appropriate.

For now, the code does not include the $\text{Idiv}(w)$ term of equation of equation 2, but I guess this would not be too complicated to add, and that may give interesting results, for future studies.

In conclusion, I am still not completely convinced of the validity of the flow field measurements.

The calculus on the simulation would anyway not convince me either, as the intensity, there, seems rather continuous, with no apparent podosome structures, which is for me the main problem of the experimental data.

But I guess it is not at all the main point of the paper, which I find anyway overall very interesting,

Concerning noise, I like the idea that noise can be locally correlated, and it indeed seems to be well in agreement with the experiment, however, I would have liked to visualize the exponential fit, it seems to me that the data would be somehow better fitted by a double exponential decay rather than by one.

And I am not sure that this correlation procedure is anyway giving a good estimation of the wavelength, as the flow field gives a direction, and you expect a better correlation in this direction (the one of the flow), whereas here all directions are averaged. So in a perfect world, I would have for each podosome center look at the maximum correlation length as a function of the angle, but this would require of course a lot of data, and I have no clue if it is possible. As a conclusion, I would expect anyway a larger correlation length.

Concerning the PDMS loss and elastic modulus, I am kind of surprised by the articles that have been selected by the authors to answer my concerns:

Petet 2021 is dealing with mixtures of two kinds of PDMS which is to my understanding not relevant here, and leang 2015 is addressing the problem of viscoelastic solid behavior at short time scale, whereas I was talking of liquid viscous elastic behavior at long time scales.

I am joining for example the results of indentation experiments performed by Trappmann et al. in their supplementary information paper in 2014, it is very instructive (Fig S1).

I was also surprised that the two lines describing the protocols have now disappeared from the paper, which I really find annoying.

Response to Reviewer #1

We appreciate the genuine interest of Reviewer #1 in our study and would like to thank them for the effort they have put to thoroughly read through our work, read related literature, and test different algorithms to provide thoughtful comments and even suggestions for future work.

To respond to the concerns of Reviewer #1, we discussed the key aspects of their comments/suggestions with all coauthors. Based on these discussions, we provide the following clarifications and revisions that are directly related to the core messages of our current manuscript:

- 1) We have better clarified our definition of waves (more precisely height waves) and thoroughly went over the manuscript to rephrase remaining inconsistent wording;
- 2) We re-emphasized the qualitative nature of our STICS validation and comment on the differences between the theoretical value (0.1) and the experimental value (0.43);
- 3) We discussed the fitting for the correlation data with single and double exponential function;
- 4) We have added further support for the use of PDMS for our experiments.

A detailed explanation of these aspects can be found as a point-to-point reply below (***Section A, Rebuttal for the manuscript***).

In addition to these aspects, the comments of Reviewer #1 really pushed us to critically reflect upon the possibilities and limitations of our STICS analysis and compare it to alternatives including PIV and KLT (as put forward by the Reviewer). We have therefore now also included a discussion on the following topics to this response letter

- 1) Considerations regarding the use of STICS in the context of our work
- 2) The ability of STICS to accurately measure height waves
- 3) Improving the quantification of height waves
- 4) Variations observed in the flow field
- 5) Ensuring STICS vector quality
- 6) Selecting appropriate parameters: sensibility to the window size
- 7) Sensibility to the algorithm used

An extensive and detailed discussion on these topics, including newly generated preliminary data that demonstrate the possibilities of future STICS improvements, can be found as a point-to-point reply below (***Section B, Possibilities and limitations of STICS***). We have specifically chosen to share these discussions and preliminary data confidentially with the Reviewer rather than including them in the current manuscript. The reason is that the main contribution and focus of our current work is on the development of a theoretical and mechanistic model that explains how height waves can arise in collective podosome dynamics. While the STICS analysis is part of the qualitative validation of our theoretical model, improving the methods to measure wave propagation is outside the scope of this manuscript. We do want to emphasize that the additional methodological improvements suggested by the Reviewer are of great interest to us and we would be delighted to further interact and collaborate with the reviewer to improve our STICS analysis. Yet, we hope that the Reviewer agrees that such improvements would be best addressed in future work rather than in the current manuscript as they would not add to its conceptual or scientific value.

Section A, Rebuttal for the manuscript

1. Clarification to the definition of waves

“The fact that ... (can we really think in terms of waves in this case?)”

Response: We would like to clarify our definition of *waves*, as this term can be applied in different contexts and may lead to different interpretations.

In our introduction, we define waves as follows: “[...] our model demonstrates how mesoscale coordination of individual podosome oscillations can arise and form wave-like propagation in clusters, which we call chemo-mechanical diffusion waves”. To clarify, we consider that podosomes display wave-like dynamics in two distinct spatiotemporal scales: lateral displacement of individual podosomes and height waves throughout the entire cluster.

1. Individual podosomes experience reorganization events due to actin polymerization and depolymerization processes. This leads to changes in their shape and to the apparent ***lateral displacement*** of the podosomes, as well as to splitting and merging events as the architecture of the podosomes is reorganized. These displacements can appear to be directional when a group of podosomes seem to be pushed or carried away together. The dynamics at this level occur at relatively small spatial and temporal scales, and both individual and collective displacement dynamics take place in the XY plane. As the reviewer points out, displacement dynamics at this scale are very eventful and eye-catching, making it hard to visually extract the second-type collective dynamics (stated in next paragraph). *Importantly, while our work does not primarily focus on this first type of collective behaviour, the effects of lateral displacements are accounted for by the Gaussian noise added in our theoretical model.*
2. Podosome clusters also collectively exhibit coordinated height fluctuations. Since an increased height is associated with more F-actin in the core, it is generally accepted that the fluorescence intensity of an F-actin probe (in our work LifeAct) reflects the height of podosomes. The coordination of podosome ***height*** oscillations therefore leads to the emergence of F-actin *intensity* waves that propagate within podosome clusters and are the subject of the model proposed in this manuscript. These dynamics can span significantly larger spatial scales than individual podosome dynamics and sometimes even the entire cell. *In our work, we provide a theoretical model for the molecular mechanisms that explain the emergence of these height waves, which is the key contribution of this work.*

We envision that future theoretical developments and models can further build upon our framework on height waves by incorporating directional displacement or biased formation of podosomes to interpret the complex podosome cluster dynamics in dendritic cells.

Specific revision: To make it clearer that we specifically focus on podosome height waves in our work, we have now added the word “height” to our definition of waves in the introduction, which now reads:

“[...] our model demonstrates how mesoscale coordination of individual podosome ***height*** oscillations can arise and form wave-like propagation in clusters, which we call chemo-mechanical diffusion waves”.

2. Recovery of the correct wave velocity

“The fact that they did not recover analyzing the simulating data the correct wave velocity (0.1 μm versus 0.43 μm).”

Response: We would like to thank the reviewer for this comment on the matching of wave speeds. First, we want to point out that **1)** podosome waves are highly dissipative and dispersive (the wave speed depends on the wavelength) and wave velocity is not uniquely defined, and there is no such a constant value for “correct wave velocity”; Secondly, we have discussed **2)** the potential causes for the wave speed difference between experiments and simulations, which includes the irregular distribution of podosomes; Lastly, we have **3)** estimated the propagation speed of radial waves without using STICS to further validate our theoretical modeling.

1) Podosome waves are highly dissipative and a wave velocity is not uniquely defined

We apologize if the definition of height wave speeds for podosome dynamics was ambiguous in our manuscript. We should note that the podosome “diffusion” waves predicted by our theory are highly dissipative and display dispersive behaviour, where the wave speed depends on frequency (as shown in Eq. 10). Podosome waves do not have a unique wave speed, which are different from the conventional traveling waves $u(\vec{r}, t) = F(\vec{r} - \vec{v}t)$ with constant and well-defined frequency-independent wave speed $|\vec{v}|$. It should also be noted that these waves are damped and have a decaying (imaginary) contribution unlike the conventional traveling waves.

To obtain the dispersion relation of the unconventional waves from the three governing equations for podosome height, myosin force and G-actin distribution (Eq. 5-7), we applied the linear perturbation $\delta h \sim e^{i\omega t + i\vec{q}\cdot\vec{r}}$ to the steady state, where \vec{q} is the wave vector and ω is the angular frequency. We then obtain the wave speed $v = \omega/|\vec{q}|$ that is a function of frequency ω (Eq.10, refer to Fig. S10D-E). To physically interpret this, recalling that oscillations of individual podosomes cause G-actin concentration change and drive the wave dynamics, each podosome oscillation dynamics yield a radial wave with a wave length and speed corresponding to its oscillation frequency. As the podosomes at different positions oscillate with varied frequency (due to noises and nonlinear coupling), the wave length and speed also vary with frequency. While a unique wave velocity is not defined, we can get an estimate of characteristic wave speeds by choosing $\omega = \omega_p$, which is the characteristic oscillation frequency of the podosome. Given this calculated wave speed is in the range of 0.1-1 $\mu\text{m}/\text{min}$ for the upper and lower limits of ω_p in Table S2, this is indeed an independent analytical estimate of wave speed compared to the decay of correlations.

The simple estimate above has the ability to distinguish between the control and pharmacological treatment cases, as the mechanism of the diffusion waves is disrupted after treatments. Here we consider how the characteristics of the diffusion waves change with different pharmacological treatments. As inhibition of myosin motors and actin polymerization disrupt the oscillations, we reduce the oscillation frequency $\omega \sim \omega_p$ in our model, leading to a decrease of the wave speeds. This observation has been summarized in the 3D surface plot in Fig. 5C, which shows how the wave speeds depends on the oscillation frequency ω and ω_p . Importantly, the flow velocity from STICS measurement also decreases as shown in Fig. 5D, in agreement with our theoretical predictions.

Above all, there is no a constant “correct wave velocity” as the reviewer #1 mentioned; The so-called “correct wave velocity 0.43 $\mu\text{m}/\text{min}$ ” is also an estimation obtained from the correlation plot by using the wavelength $\lambda \approx 3 \mu\text{m}$ and oscillation period $T \approx 7 \text{ min}$. **Since the theory predicts dispersive waves where the wave speed is not uniquely defined, the consistency between the order of magnitude obtained from STICS analysis (0.1 $\mu\text{m}/\text{min}$) and the theory using three different methods (analytical analysis presented above $\sim 0.1\text{-}1 \mu\text{m}/\text{min}$, estimation from the correlations and direct numerical simulations $\lambda/T \sim 0.43 \mu\text{m}/\text{min}$) is satisfactory.** Furthermore, our theory correctly predicts the trends in wave speed in the case of pharmacological inhibition experiments, lending strong support for the veracity of the theory; this is the main point of evaluating wave speeds in our work.

2) The irregular distribution of podosomes contribute to the difference in extracted wave speeds

We have used simulations to demonstrate that the proposed model would indeed result in the formation of propagating podosome height waves. We do not expect neither our simulations to reconstitute all the complexity of the cell and to reproduce what is observed in live cells, neither the measurements of experimental podosome dynamics using STICS to match the simulated height propagation speed. On that note, it is worth mentioning that the grayscale values displayed in our simulations do not represent podosome F-actin intensity. Instead, they represent the height that a podosome would have if there was a podosome in that position. Our simulations with continuum model could be interpreted as the podosome height field or potential, which our model predicts to propagate spatially, making discontinuously located podosomes' heights to oscillate in a wave-like manner. In addition, the irregular distribution of individual podosome in a cluster can also lead to the apparent *turbulence* in the waves observed experimentally, as opposed to the steady state behaviour observed in our podosome height potential simulations. Therefore, **it is not our intention to recover or match the exact same wave velocity obtained from STICS measurements on our simulations in those obtained from experiments, and the consistency between magnitudes of extracted wave speeds is what we initially aimed for.**

3) Estimating the propagation speed of radial waves without STICS

Lastly, we consider yet another approach to validate the predictions of our theory by estimating propagation speed of radial wavefronts in our experiments without using STICS or correlation plot. As shown in the **Fig. R1**, the wavefront propagates from the cell center to the peripheral in a radial manner, and average distance traveled during the 15 min time interval can be measured as around 5.53 μm . Thus, we can get an estimation of wavefront propagation speed $5.53/15 \mu\text{m}/\text{min} \approx 0.37 \mu\text{m}/\text{min}$, which again is fully consistent with the speeds observed for random waves from our theory and STICS.

Fig. R 1 (A) Dendritic cell transfected with LifeAct-RFP showing the propagation of radial waves. Colours denote the fluorescence-labelled podosomes at different times. The double-arrow lines indicate the wavefront traveling distance for 15 min time interval. Scale bar, 10 μm . Figure adapted from our previous study (M. Meddens, et al., Nat. Commun., 2016). (B) The wavefront propagation speed measured for the radial wave as shown in panel A. Error bar indicates the standard deviation.

3. Double exponential fit

“Concerning noise, I like the idea that noise can be locally correlated, and it indeed seems to be well in agreement with the experiment, however, I would have liked to visualize the exponential fit, it seems to me that the data would be somehow better fitted by a double exponential decay rather than by one.”

Response: We would like to thank the reviewer for this comment. We have now provided fitting of the data using a double exponential decay function $y = y_0 + ae^{-x/\lambda_1} + (1 - a)e^{-x/\lambda_2}$ for one typical correlation curve (**Fig. R2**).

Fig. R 2. The correlation coefficient plot of the distance between podosomes. Blue dots: experimental data; Red line: fitting curve with a single-exponential decay function; Yellow line: fitting curve with double-exponential decay function.

The reviewer is correct that fitting with double-exponential decay function yields a higher coefficient of determination $R^2 = 0.99$ compared to $R^2 = 0.93$ with single exponential decay function $y = y_0 + e^{-x/\lambda_0}$. While a double-exponential decay function leads to better fitting results, we do not think it provides further physical insight than a single exponential decay function.

Regarding the correlation procedure for wavelength estimation, we understand that the reviewer has concerns on this correlation procedure due to the direction of wave propagation. We appreciate the reviewer's suggestion for the improved procedure by taking account of angles, but currently the data does not allow us to do it, as the podosomes are discretely distributed in space and wave flows exhibit highly random propagation angles. What we have done gives an average over different directions. We agree that the current procedure may be not the best or most precise approach for the estimation of wavelengths, and that this correlation procedure could underestimate the correlation length and hence wave speed due to averaging among all directions. However, the key point is that the estimated wave propagation speeds are reduced after pharmacological treatments, which gives the same trend as the STICS measurements and further confirm our model predictions. We believe this is sufficient for the current study.

4. PDMS as a substrate choice

“Concerning the PDMS loss and elastic modulus, I am kind of surprised by the articles that have been selected by the authors to answer my concerns:

Petet 2021 is dealing with mixtures of two kinds of PDMS which is to my understanding not relevant here, and leang 2015 is addressing the problem of viscoelastic solid behavior at short time scale, whereas I was talking of liquid viscous elastic behavior at long time scales.

I am joining for example the results of indentation experiments performed by Trappmann et al. in their supplementary information paper in 2014, it is very instructive (Fig S1).

I was also surprised that the two lines describing the protocols have now disappeared from the paper, which I really find annoying.”

Response: We are familiar with the Trappmann *et. al.* paper about PDMS viscoelasticity. In Fig. S1 of their work, PDMS substrates show the quick relaxation of the stress in the first 10 s, and they eventually reach to steady state after ~ 30 s. As the oscillation period of the podosome (~ 400 s) is much larger than this relaxation time scale, the PDMS substrates relax to an “elastic” steady state at the very beginning of podosome protrusion. Hence, we believe that the viscoelasticity of PDMS will have negligible influence on the dynamics. However, we do agree that how the ECM viscoelasticity and other nonlinear mechanical properties affect the podosome dynamics can be an interesting topic and worth further investigation in future studies.

In Leang 2015 paper, the mechanical response of PDMS was measured using AFM probe whose dimensions are in the range of $1 \mu\text{m}$; This is different from the work of Trappmann *et. al.* 2014, which uses a 20 mm indenter and the mechanical properties correspond to a much larger scale. For a single podosome with its core diameter $\sim 1 \mu\text{m}$, we consider the measurements of Leang 2015 paper to be more relevant.

Specific revision: We have added the following sentences to the *Discussion* section of manuscript:

“Future theoretical developments and computational models may build upon our chemo-mechanical model by incorporating the nonlinear properties of the microenvironment (e.g., matrix viscoelasticity, matrix heterogeneity, and topographical cues) and matrix degradation, leading to a comprehensive understanding of podosome dynamics in the context of wound healing and cancer immunotherapy.”

Regarding to the descriptions to of protocols, we apologize for the deletion by mistake. We have provided the information for the protocols on fabricating the gels with different stiffness in the *Methods* section.

5. Conclusion

Altogether, we hope that these specific clarifications and revisions to the manuscript sufficiently address the reviewer's concerns. We believe our work represents a significant contribution to the field in providing a novel, sound and robust theoretical model to explain the emergence of collective podosome height waves and therefore hope it is now considered acceptable for publication in *Nature Communications*.

Section B, Possibilities and limitations of STICS

Considerations regarding the use of STICS in the context of our work

“...not completely convinced of the validity of the flow field measurements.”

Our ability to capture wave-like podosome dynamics differs considerably in each of the two wave cases presented above (*Section A, point 1*). For instance, we can measure *displacement* dynamics (wave type #1) using image correlation techniques such as PIV or STICS, since these techniques have been designed to extract flow from consecutive images of particles in motion (in which their position changes as a function of time). Yet, the collective *height* waves (wave type #2) that our model predicts are much harder to capture as they do not involve a change in position but a change in podosome height (which is picked up as F-actin intensity fluctuations by confocal microscopy). We realize that, since STICS was initially designed to capture particle displacements in the XY plane and not fluorescence intensity fluctuations, applying it for such task may raise concerns regarding the appropriateness of its use. We think that this is the core of the issue that continues to be raised by Reviewer #1 and that has led to confusion regarding what type of waves we are trying to measure. Nonetheless, the fluorescence intensity fluctuations can be detected by STICS with advantages compared to other methods. STICS can be considered as the spatiotemporal extension of Fluorescence Correlation Spectroscopy (FCS), or as the temporal extension of Image Correlation Spectroscopy (ICS). As such, STICS is particularly useful at extracting similarities of fluorescence intensity profiles in space and at analyzing the way they evolve as a function of time. As the reviewer is clearly aware, STICS follows a very similar framework as PIV, in which image correlation is calculated between two subsequent frames, and the displacement is retrieved by maximization of a fit to the obtained correlation function. Yet, as will be detailed below, STICS takes advantage of temporal oversampling to extract displacements from more than two frames, which allows for vector quality control estimation that would not be available when using only two frames as in PIV (see “Ensuring STICS vectors quality”). Beyond the enhanced information extracted from more than two frames, STICS is particularly suitable for this application because tuning the spatial and temporal parameters allows us to select which spatiotemporal regimes to preferentially pick-up. By doing so, we can increase our capacity to detect podosome height waves by selecting a big enough ROI and a small enough TOI, while minimizing the contribution of podosome lateral displacements, which would be better captured using a smaller ROI and an even smaller TOI. Leveraging this tuning capacity has been the rationale behind our decision to use STICS in this and previous publications, although the reach, capacities and limitations of its applications may not have been as thoroughly discussed in our previous publications as we are doing here.

Ensuring STICS vectors quality

“The fact that the velocity field was varying on a very short length scale [...] and that a lot of arrows have now been discarded (why is the algorithm so often failing to recover a displacement?)”

We appreciate the concerns raised by the reviewer on the variations between subsequent flow fields and on the number of vectors missing. We understand that this could be misinterpreted as an indication of suboptimal results. To thoroughly explain the origin, relevance and significance of such variations and absences, we first need to provide further details on how we ensure the quality of STICS vectors.

The reviewer has previously mentioned that if we cannot detect the movement by eye, it is hard to measure it, and that image correlation techniques can nevertheless provide a vector field. As this is a

serious issue when working with image correlation techniques, developing a robust and thorough protocol for parameter selection and quality control is complicated but extremely important. To address this, we have defined quantitative quality metrics and used them to implement multiple filters or quality tests that are applied at different steps in the algorithm. Among others, exclusion criteria are applied on:

- The correlation function (CF):
 - Number of peaks found and their size with respect to the major peak.
- The Gaussian fit properties:
 - The goodness of fit
 - Half-width
 - Amplitude
 - Amplitude decay rate.
- The linear fit on the coordinates of the Gaussian peak (displacement in X or Y vs τ , see **Fig. R3E**):
 - R^2 : The user can specify a minimal coefficient of determination (R^2) that must be satisfied by the linear fit to be considered valid (set at 0.9 in our work). If the coefficient of determination (either in X vs τ or in Y vs τ) does not satisfy the threshold, then the last point is discarded iteratively until a satisfactory R^2 value is reached. This results in a variable number of correlation functions retained for the calculation of each vector.
 - Significant fit ratio: The number of retained CFs (tau limit) divided by the TOI size (for normalization).
 - The significant fit ratio multiplied by the coefficients of determination R^2_x and R^2_y yield an overall score or metric that can be used to estimate the quality of the vectors. These metrics provide a quantity judging and ensuring the quality of the analysis.
- Visualization filters
 - In addition, only those vectors that have passed all the quality control filters and that have a significant fit ratio greater than a user-set value, are shown.
 - To avoid contributions from the fast actin retrograde flow observed at the edges of the cells, vectors were only calculated inside a mask encompassing regions where podosomes were observed.
 - The user can set a velocity range for vector display, masking off vectors with magnitudes outside the range.

Importantly, if a region is devoid of vectors, it does not necessarily mean that the algorithm has failed to detect the movement. Other potential reasons for a void region include being outside of the mask, the magnitude of the vector being outside the display range, or that at least one of the vector properties did not satisfy the visualization criteria set by the user and was not displayed. Although we have always taken thorough measures to ensure that the STICS vectors we reported are valid, we have also been working on the development of an enhanced and extended version of STICS, that includes a graphical user interface to ensure a further improved quality control of the STICS vectors in the future. We hope to make this user-friendly software available in a separate paper soon (RMR and PW would be happy to hear the opinion of Reviewer #1 on the new STICS version in the future). A big part of the efforts put into this improved version have been dedicated to quality control and to a parameter selection guide, including an interactive vector

explorer (**Fig. R3**). Using this tool, one can directly inspect each individual vector and interrogate the correlations that gave rise to it, thus identifying common causes that result in spurious vectors and adjusting the quality requirement thresholds to avoid them.

Fig. R3. Screenshots of the interactive Vector Explorer. A) Dendritic cell expressing LifeAct-GFP overlaid with STICS vector field. Vectors are coloured and scaled according to their magnitude. The ROI used to calculate the inspected vector is shown in blue. B) Magnified image showing the ROI in A and the inspected vector in the centre. C) Colour-scaled vector magnitude map in $\mu\text{m}/\text{min}$. D) To inspect the vector shown in B, the 5 correlation functions as a function of time delay are plotted in two views. On the left, a 3D view shows the correlation function as a coloured surface and the Gaussian fit as a black and grey mesh. Green, red, and magenta panels facilitate the interactive 3D visualization. On the right, a top view of the correlation function shows the coordinates of the CF maxima (red + marker) and Gaussian (red ° marker) peaks as a function of τ . E) The displacements in x and in y of the Gaussian peak centre show a linear relationship and are then used to calculate the vector components. F) The CF maxima (+ marker) and the positions of the Gaussian peak (° marker) are plotted, coloured by their time delay (τ), along with the resulting vector.

Absent vectors and variations observed in the velocity field.

“The fact that the velocity field was varying on a very short length scale [...] and that a lot of arrows have now been discarded (why is the algorithm so often failing to recover a displacement?)”

Now that we have detailed how STICS vector quality is ensured, we can explain the absence of vectors and discuss what is the meaning of the variations observed in our previous video.

The reason why many vectors are missing is not necessarily because the algorithm failed, but mostly because we set the visualization thresholds too high to show all the vectors. The minimum Significance Fit Ratio threshold used in our previous video was set at $> 3/5$, meaning that only vectors that arose from linear fits that conserved 4 or 5 points out of 5 were shown (**Fig. R4**). Bear in mind that all these vectors have a coefficient of determination of over 90% by design, so even the vectors that arise from 3 points

are well aligned. However, vectors that arise from 2 points have, by definition, an R^2 value of 1, regardless of whether they capture the dynamics of interest or not. Since we cannot untangle which of these vectors capture podosome height waves and which are spurious, we reject them all (they are not shown). Needless to say, that if only one out of the five correlation functions available passes the quality controls before linear fitting (or if none of them do), the vector is discarded. In these cases, we could indeed say that, using the current parameters, the algorithm has failed. However, this is rather rare: in our previous video, only 20 vectors out of 191,999 vectors (0.01%) failed (plus an unknown fraction of vectors composed of only two points). Nevertheless, with several regions of interest yielding vectors with $\geq 3/5$ points and a coefficient of determination of $\geq 90\%$, and keeping in mind that the position of these vectors is determined by a regular grid so that both feature or wave-rich and noisy regions are covered, we consider that the algorithm can in most cases successfully detect fluorescence intensity propagation.

Fig. R4. Significance Fit Ratio Vector Field. Representative vector field obtained by applying STICS to the first five frames of a dendritic cell expressing LifeAct-GFP time series. The vectors are sized according to their magnitude (see reference arrow in the lower right corner) and coloured according to the number of points used in the linear fit. A 32 x 32 pixels box is shown in the top left corner as a reference for the ROI size.

The fact that the set of hidden vectors changes for each flow field contributes greatly to the apparent variations between subsequent flow fields. Since every flow field is calculated on a sliding window, if the subsequent vector conserved less than 4 points, the vector would disappear, whereas regions in which vectors make it above the threshold would have a vector appear at later frames. However, among vectors that remain in subsequent flow fields, most of the changes in angle and magnitude are not as striking upon closer inspection (**Fig. R5**). This is to be expected given the overlap in time between the TOI window used to calculate subsequent flow fields.

Fig. R5. Comparison between two subsequent vector fields calculated by STICS. Only vectors with 3 points or more that appear in both the first and second flow fields are shown. Vectors are sized according to their magnitude and coloured according to the vector field they correspond to. A 32 x 32 pixels box is shown in the top left corner as a reference for the ROI size.

In fact, not only do regions in which vectors persist across two or more vector fields present small variations within what is expected, they also demonstrate the ability of STICS to quantify podosome height waves. We realize that this may be very difficult to visualize in our previous video, in which each vector field was overlaid atop the average of the frames used to calculate it and followed by the next flow field

very rapidly. To better show this, it is best to first visualize what STICS “sees” and to observe five consecutive frames without any vectors in a loop (**Video 1**). We can appreciate the small displacements of some podosomes, as well as the appearance and disappearance of some others, along with coordinated changes in the intensity of some more. By doing this, the podosome height waves become clearer, even to the naked eye. We invite the reviewer to select regions of interest in which podosome intensity, and not their position, fluctuate in a wave-like manner, whether this is a region with podosome formation and disappearance events or only fluctuating intensity among persistent podosomes. We can now compare this by-eye wave detection with the results shown in our previous video including vectors with 4 or 5 out of 5 points (**Video 2**). As pointed out by the reviewer, the map is rather sparse, yet it is clear that the regions in which wave propagation can be detected by eye are indeed reported by STICS with coherent magnitude and angle. To increase the density of the vector field, we can include in the visualization vectors with 3 points (**Video 3**). We can note that the newly added vectors, particularly those found near vectors with more than 4 points, are coherent in angle and magnitude to their neighbours. If we wanted to obtain a maximal vector density, we could then include vectors with 2 points (**Video 4**). Doing so will include both spurious and valid vectors, which for the moment cannot be untangled as previously discussed. However, while we see the appearance of spurious vectors, particularly at the edge of the cell mask, we can qualitatively speculate that some of the 2-point vectors found among 3 or more vector clusters are valid, as they also show consistent angles and magnitudes as their neighbours (**Video 5**).

We hope this section addresses the reviewer’s concerns regarding the variations between subsequent vector fields and explains why many of the vectors were not shown in our previous video. These points do not necessarily mean that STICS cannot detect podosome height waves, but rather that 1) noisy or wave-lacking regions in the image are not suitable for accurate STICS calculation and that 2) the way we represent our data and visualize it plays an important role in our capacity to deliver a clear message. We will come back to discussing how both variations and absent vectors fractions can be reduced in the section “Improving the quantification of podosome height waves”.

As show here, the TOI size can play a crucial role in determining the number of high-quality vectors. To be able to leverage the coefficient of determination for quality control, we need at least 3 points. Therefore, temporal oversampling by imaging at high frame rates allows us to obtain more data to fit our linear model to, while preserving the assumption of linear displacement over that TOI. For this reason, we have also paid meticulous attention to the selection of appropriate parameters.

Selecting appropriate parameters

Sensitivity to the window size

“...the window size may influence greatly the results.”

Indeed, the selection of an appropriate ROI and TOI sizes determines which spatiotemporal regimes are being picked-up by STICS and how well. In general, the size of the region of interest (ROI) must be adjusted to the characteristic size and velocity of the dynamics of interest and will indeed influence the results. To our knowledge, the only situation in which this could be minimized is if the data presents scale-free dynamics in only one spatiotemporal regime or if this one dominates over the rest. In this case changing the ROI size would mostly affect our *ability* to pick it up, but the overall values obtained (provided they are of high enough quality and with high signal-to-noise) should in principle follow the same distribution,

even if more sparsely sampled. Otherwise, varying the ROI size will determine at what scale are the dynamics being detected.

This allows us to tune between fine and coarse grain dynamics. When measuring particle displacements, STICS assumes that during a given time of interest (TOI), the particles contained within a given ROI experience displacements that have spatially homogeneous and temporally constant speed and direction. For this reason, it is best to oversample in both space (high magnification) and in time (high frame rate). To obtain optimal STICS results, the ROI size must be small enough to make sure all the particles are moving in the same direction at the same speed, but large enough to encompass and track patterns over a TOI (if the particle moves too fast, it will leave the ROI too soon and the amplitude of the correlation functions will decay quickly down to noise-dominated regimes). Same applies to the TOI size: it must be short enough to guarantee that the displacement was linear during that period of time, but long enough to make sure sufficient data points are collected for an appropriate linear fit.

To ensure that the ROI size was appropriate to capture the collective F-actin intensity wave propagation, we performed a screen using a single vector field, spanning 16, 32, 64 and 128 pixels per ROI side (since we use fast Fourier transforms (FFT), we need to scale the ROI size by 2^n). Guiding ourselves with the quantitative quality metrics described in the section above on vector quality as well as the ability to track patterns in podosome clusters, we selected a ROI size of 32 pixels.

The ability of STICS to accurately measure height waves.

“...doubts on the real capacity of STICS to give access to the wave velocity.”

As discussed previously (see “Clarification to the definition of waves”), our efforts to capture collective height waves using STICS are indeed conflated with picking up podosome displacement dynamics. Selectively picking up collective podosome height waves is very challenging, no matter which of the available image motion detection techniques we are familiar with we use. We, however, have always been aware of this limitation. In our previous publication (Meddens et al. 2016), we have already made efforts to estimate the contribution of lateral displacements to our STICS analysis of podosomes. For this, we applied STICS to image series of WASP-GFP. WASP is a component of the podosome core and is responsible for the nucleation of actin at the base of the core. WASP therefore does undergo lateral movements but no height (or intensity) fluctuations since it is only associated to the base of the core. When applying STICS to these image series, we observed significantly lower vector pair-correlation and significantly reduced wave velocity with respect to LifeAct STICS results. This indicates that core movement is picked up but that a significant fraction of the correlations is due to the contribution of height fluctuations.

Encouraged by the reviewer’s comments, we have made preliminary efforts to push our exploration of the ontology of these limitations, seeking to find ways to overcome or minimize them. Three main issues limit the application of feature-detection and tracking-based or image correlation-based approaches to selectively extracting podosome height waves directly from XY podosome displacement dynamics:

1. **Discontinuity:** the waves we are trying to measure are discontinuous in the experimentally available space of F-actin. Indeed, the decrease in F-actin intensity of a given podosome is correlated with an F-actin intensity increase at a neighbouring podosome, but these podosomes

are, in most cases, a couple of microns apart. This constitutes perhaps the greatest contribution to the risk of detecting and tracking individual or local collective podosome displacements instead of collective podosome height waves.

2. **Without a traceable pattern:** even if we were able to measure the continuous actin concentration gradient space, this would be composed of peaks, valleys, and smooth slopes, but not have a feature, “signature” or “pattern” that could be “tracked” as an object would. As the reviewer points out, it would be difficult to say with certainty that “this actin molecule went from here to there”. For this reason, feature detection and tracking would not be advantageous.
3. **Not a physical movement:** unlike particle tracking or flow determination, height wave tracking does not measure the displacement of a physical entity, but rather a signal that is relayed from a podosome to the next via local actin concentration fluctuations, without implying that an actin molecule is physically carried by the wave.

To our understanding, these limitations apply for all the image motion detection techniques we are familiar with, as they rely on detecting a pattern in the image and on detecting it in a different position at a later point in time, assuming it is the same object that moved from one place to the other.

We appreciate the video of the Kanade-Lucas-Tomasi feature tracker applied to our data that the reviewer provides and would be happy to test their code in future applications. In their video, we observe a significant improvement in podosome displacement tracking in the sense that it accurately tracks each podosome and locates each vector on the center of each podosome instead of using a regular grid, which makes podosome displacements clearer: we can observe how each arrow seems to “pull” from the podosome, predicting its position at future stages.

Importantly, to more accurately measure the podosome height waves, and to separate them from the lateral podosome displacement waves, we are currently developing improved STICS approaches. Yet, considering that the main goal of this paper is not to quantitatively measure the waves, but rather to provide a model to explain their emergence, we are strongly convinced that this method development is outside the scope of this paper and would be better suited in a separate publication. Nonetheless, since the reviewer expresses such explicit interest in the STICS method, we do confidentially share some of our preliminary findings below.

Improving the quantification of podosome height waves

Despite the aforementioned limitations and challenges, we have found a way to improve our ability to capture podosome height waves using STICS. As discussed above (see “Considerations regarding the use of STICS in the context of our work”), STICS features particular advantages that can be leveraged to minimize the contribution of podosome displacements in favor of podosome height waves. By modifying the way that we process our data before applying STICS, we have now significantly reduced the discontinuity limitation, further improving our ability to capture podosome height waves.

So far, we have presented results of STICS applied directly to the original podosome image series (only preprocessed by an image registration algorithm to reduce subsequent frame displacements), choosing the parameters to the best of our ability. However, to be able to capture the height waves while minimizing the influence of their displacements, we need to preserve the information about the podosome height, encoded in the F-actin *intensity*, while suppressing the “texture” or “patterns” in the

image. In other words, we need to get rid of the podosome structures, but keep their intensity. For that purpose, we have applied a Gaussian filter to our data before applying STICS (**Fig. R6**). By filtering the image, we kept the relative differences in actin intensity (i.e., height) between different parts of the cluster, but we removed the image patterns that can “trick” STICS into tracking individual podosome positions (compare **Video 1** with **Video 6**).

Of course, as podosomes do both change in position and oscillate in height, podosome reorganization events that lead to their apparent displacement result in a change of the intensity gradient in the blurred image. The opposite is also true: height waves are in some cases concomitant with podosome displacements. This means that we cannot completely untangle the podosome displacements from their height waves. Whatever measurement we make of one or the other will have bleed-through contributions of its counterpart. Bearing that in mind, we consider that using this strategy to reduce the discontinuity does allow us to better capture the height waves in our system than when applying STICS directly. By applying a Gaussian filter, we removed the structural component of the image while preserving the relative and local actin intensity. Podosomes lost their shape, but their intensity was combined locally between neighbours and extended spatially, resulting in a continuous map of F-actin intensity. This way, the correlations made by STICS were no longer done on a discontinuous map of distant podosomes (see “The ability of STICS to accurately measure height waves.”), but rather in a continuous map of F-actin intensity.

Fig. R6. Schematic representation of the new strategy to capture podosome height waves using a Gaussian filter and STICS. Five consecutive frames of a time series (top left) are processed using a Gaussian filter (top right), which are then used to calculate STICS (bottom right). To verify that the captured waves reflect podosome height waves and not their displacement, each vector field is overlaid over the original time series (bottom left). Vectors are coloured and sized according to their magnitude.

We then applied STICS to Gaussian-filtered time series and validated that STICS vectors were consistent in both magnitude and orientation with the observed propagation of Gaussian-filtered height waves (**Video 7**). Furthermore, these waves were traced back to the original image series, where the podosomes F-actin intensity, rather than their position, fluctuated and propagated following the directions and magnitudes of the STICS vectors (**Fig. R6** and **Video 8**). This is true even if only vectors generated with 3 or more points are shown (**Video 9**). Not only were we able to quantify the propagation of the waves that we were able to visually identify and track; applying STICS to Gaussian-filtered image series revealed several other waves that were harder to pick up visually (**Video 10**). This can be better visualized if instead of following the sequence of vector fields, we display the sequence of frames and overlay two of the 5

vector fields they contribute to (**Video 11**). After thorough inspection of our results, we have so far been able to trace all wave propagation detected by STICS in the fluorescence intensity of the original image series.

By applying STICS to the Gaussian-filtered time series, we were able to minimize the effect of podosome displacements (which have a short range and occur in different directions over short periods of time, contributing to variations between adjacent flow fields) and instead prioritize the effect of podosome intensity fluctuations. This resulted in smooth transitions between flow fields, both in terms of angle and of magnitude (see **Fig. R7**), revealing the lateral propagation of waves. F-actin intensity waves were frequent and abundant, persisted between 1:45 min to 2:30 min (3 - 6 TOIs) approximately, and traveled 1-5 μm within clusters before clashing with others or dissolving (**Video 10** and **Video 12**). The use of the Gaussian filtering strategy has therefore allowed us to better capture the podosome height waves and to significantly reduce the variations in angle and magnitude between subsequent vector fields.

Fig. R7. Gaussian filtering strategy considerably reduces the variations between subsequent frames. Comparison between subsequent vector fields calculated without (left) and with Gaussian filtering before calculating STICS (right). All vectors with two or more points used for the linear fit are shown.

Although we consider these results promising in the context of unentangling the different wave types in podosome clusters, we are still optimizing this approach in terms of size of the Gaussian filter and whether we for example should keep a constant filter size or adapt it dependent on the local nearest neighbour distance. We are in the process of preparing a future manuscript that includes these types of adaptations of our initial STICS analysis.

Sensitivity to the algorithm used.

“...the algorithm used [...] may influence greatly the results.”

Just as the window size, the choice of the algorithm will depend on the dynamics of interest. For instance, the reviewer provided a very good example of the KLT algorithm being able to detect the podosomes as features and track their displacements. To our humble understanding, the KLT algorithm is particularly

efficient because it detects the features to be tracked based on the same criteria it uses to track them (Shi & Tomasi, 1994), therefore finding the *most useful* regions of the image to be tracked. This contrasts with attempting correlation-based tracking across the whole image in a regular grid as done by STICS. We would be curious to borrow the reviewer’s codes for future comparisons to see how the KLT algorithm performs on the Gaussian-filtered time series. However, in this case in which we are trying to avoid picking up podosome displacements, we don’t think that feature tracking would be particularly advantageous, specially in the case of the Gaussian-filtered images which will contain smooth gradients, therefore small differentials and not very informative or useful features to be tracked. We find that STICS has been particularly useful in our case, as it can extract useful information from smooth Gaussian-filtered images as long as the ROI size encompasses sufficient area and that the gradients preserve some similarity in time, as is the case of propagating height waves seen in Gaussian-filtered time series.

Regarding the assumption requiring homogeneous image intensity discussed by Heitz *et.al.*, 2009, we note that the displacement field d obtained by the DFD model is equivalent to the one obtained through a correlation maximization (correlation peak match as in PIV) if equation 14 is held:

$$\forall d \sum_{r \in W(x)} I_2(r + d)^2 \approx constant$$

As mentioned in the paper, this is usually the case in PIV measurements where particle seeding is generally uniform and where the interrogation window is sufficiently large. We understand this section as a comparison between the DFD model and the correlation-based techniques. In our case, the image intensity is not constant in space, as the podosomes density is heterogeneous. In STICS, the correlation function is obtained by normalizing over the mean-squared intensity, such that patterns can be detected even if the absolute values of intensity vary from one region to another. This allows us to obtain STICS measurements that are insensitive to the absolute intensity values, provided the signal to noise is sufficient. This normalization is done to enable the extraction of the relative number of particles found in a given region, thus providing density information from the amplitude of the autocorrelation function at zero spatial and temporal lags. In that sense, STICS does not assume a constant or homogeneous intensity field as in fact, the normalization allows us to measure fluorescent particle density variations and does not require this assumption.

Steady state in simulations

“I saw that there seems to be a transient regime, and that the steady state takes time to be reached (is it true? Normal?)”

For completeness, the steady state observed in our simulations is to be expected, as the initial condition or height of every pixel is sampled randomly, and it takes time for the entrainment of adjacent regions to start to oscillate in a coordinated manner.

Conclusion

Altogether, we hope that we have now better explained the possibilities and limitations of STICS in picking up the different types of podosome waves. We are certainly aware of the fact that multiple types of

podosome dynamics are simultaneously detected, and, for this reason, we continuously put effort in optimizing our STICS approach. For one of these possible optimization strategies (preprocessing the raw image data with a Gaussian filter) we have provided the details above. We again emphasize that we would be happy to interact with the Reviewer on this topic but we also hope that the reviewer acknowledges that these discussions are better suited for future collaborations/investigations and do not add scientific value to the current manuscript.

Videos

Video	STICS on	Background	Quiver	Tau	Quiver size	Frames
1	Original	Original	None	NA	NA	5
2	Original	Original	Magnitude	> 3 (4-5)	Displacement	5
3	Original	Original	Magnitude	> 2 (3-5)	Displacement	5
4	Original	Original	Magnitude	> 1 (2-5)	Displacement	5
5	Original	Original	Magnitude	NA, >3, >2, >1	Displacement	20
6	Gaussian-filtered	Gaussian-filtered	None	NA	NA	5
7	Gaussian-filtered	Gaussian-filtered	Magnitude	> 1	30x	5
8	Gaussian-filtered	Original	Magnitude	> 1	30x	5
9	Gaussian-filtered	Original	Magnitude	> 2	30x	5
10	Gaussian-filtered	Original	Magnitude	> 1	15x	96
11	Gaussian-filtered	Original	Magnitude	> 1	15x	100
12	Gaussian-filtered	Gaussian-filtered	Magnitude	> 1	30x	96

Video 1

First five frames of the time series.

Video 2

First vector field showing only vectors generated with 4 or 5 points.

Video 3

First vector field showing only vectors generated with 3 or more points.

Video 4

First vector field showing only vectors generated with 2 or more points.

Video 5

This video is the concatenation of Videos 1-4.

Video 6

First five frames of the time series after Gaussian filtering.

Video 7

First vector field generated by using the first five frames of the Gaussian-filtered time series.

Video 8

First vector field generated by using the first five frames of the Gaussian-filtered time series over the first five frames before filtering.

Video 9

Same as video 8 but showing only vectors generated with 3 or more points.

Video 10

This video shows each vector field overlaid on top of the original time series before Gaussian filtering. Each vector field is plotted on top of the centred frame: vector field 1 is generated from frames 1-5 and is plotted over frame 3. Vector field 2 over frame 4 and so on.

Video 11

This video shows each frame in a sequence and plots two of the five vector fields it contributed to. For example, frame 5 is overlaid with vector fields 2 and 3, which are generated using frames 2-6 and 3-7, respectively. Since no vector fields are generated using only frames 1, 2, 99 and 100, the first and last vector fields are plotted with thinner vectors.

Video 12

This video shows each vector field overlaid atop the average of the five Gaussian-filtered frames used to calculate it. It shows the propagation of podosome height waves at longer times.

References

- Meddens, M. B. M., Pandzic, E., Slotman, J. A., Guillet, D., Joosten, B., Mennens, S., Paardekooper, L. M., Houtsmuller, A. B., Van Den Dries, K., Wiseman, P. W., & Cambi, A. (2016). Actomyosin-dependent dynamic spatial patterns of cytoskeletal components drive mesoscale podosome organization. *Nature Communications*, 7. <https://doi.org/10.1038/ncomms13127>
- Shi, J., & Tomasi, C. (1994). Good Features to Track. *Pattern Recognition*, June.